# Cleavage of mRNAs by a minority of pachytene piRNAs improves sperm fitness

Katharine Cecchini[1], Mina Zamani[2], Nandagopal Ajaykumar[2], Joel Vega-Badillo[1], Ayca Bagci[1], Shannon Bailey[1], Phillip D. Zamore[1✉] & Ildar Gainetdinov[2✉]

Animals use 18–33-nucleotide PIWI-interacting RNAs (piRNAs) to silence transposons in germ cells[1–3]. In addition to transposon-silencing piRNAs, placental mammals make pachytene piRNAs[4,5], an abundant class of testis-specific small RNAs derived from long noncoding RNA precursors. Although the sites of pachytene piRNA precursor transcription are often conserved among placental mammals, the sequences of the piRNAs themselves are rapidly diverging, even in the human population[6]. Consequently, the biological function and mechanism of action of pachytene piRNAs remain debated. Here we report that most mouse pachytene piRNAs have no biological function but instead 'selfishly' promote their own production. Our data suggest that pachytene piRNAs direct endonucleolytic cleavage of partially complementary targets and neither activate nor repress mRNA translation. Although many pachytene piRNAs guide cleavage of specific mRNAs, few alter the steady-state abundance of their targets. The minority of pachytene piRNAs that reduce target mRNA abundance enhance sperm fitness, thereby ensuring production of the entire pachytene piRNA repertoire. Together, our findings explain the lack of conservation of most pachytene piRNA sequences and suggest that these 'selfish' small RNAs persist in mammalian evolution because target cleavage by a tiny minority of piRNAs supports male fertility.

In animals, piRNAs silence transposons, regulate host genes and repress viral transcripts[1–3,7]. piRNAs direct PIWI proteins to cleave complementary RNAs in the cytoplasm or to initiate transcriptional repression in the nucleus. piRNA precursors are transcribed from dedicated genomic loci called piRNA clusters[8,9]. For example, in fly ovaries and mouse fetal testes, transposon-targeting piRNAs are produced from precursors that comprise sequences complementary to transposons[8,9].

At the onset of male meiosis, placental mammals produce pachytene piRNAs, a distinct piRNA class made from a subset of testis-specific long noncoding RNAs (lncRNAs) devoid of active transposon sequences[4–6,10–14]. Pachytene piRNAs are highly abundant; for example, a mouse primary spermatocyte contains around 10 million pachytene piRNAs but only about 1.4 million mRNAs[15]. The peculiar mechanism of pachytene piRNA biosynthesis involves a feedback amplification loop, whereby cleavage of precursor transcripts by pachytene piRNAs initiates the production of more pachytene piRNAs[16,17].

The loci that produce most pachytene piRNAs are present at syntenic locations in all placental mammals, yet their sequences are not conserved[6]. In fact, the sequences of pachytene piRNA-producing loci diverge among species and even among modern humans nearly as rapidly as the non-transcribed regions of the genome[6]. Notably, most pachytene piRNAs are extensively complementary only to the genomic loci from which they are transcribed. The targets of pachytene piRNAs are therefore not obvious, and several models have been proposed to explain their function. The following roles have been ascribed to

pachytene piRNAs: (1) they destabilize partially complementary transcripts through a miRNA-like mechanism[18]; (2) they cleave extensively complementary RNAs through a small interfering RNA (siRNA)-like mechanism[17,19–21]; or (3) they activate translation of partially complementary mRNAs[22,23]. Alternatively, they lack intrinsic function and instead are degradation products[24].

Of the six mouse pachytene piRNA clusters that have been genetically disrupted, just two are required for normal male fertility[15,17,21,25,26]. Here our genetic data and sperm functional assays demonstrate that all six major pachytene piRNA loci have a role in supporting spermatogenesis. We show that mouse pachytene piRNAs neither activate nor repress mRNA translation, but instead regulate partially complementary targets exclusively through endonucleolytic cleavage. Of the tens of thousands of pachytene piRNA species present in primary spermatocytes, only several hundred (around 1%) are sufficiently complementary to a target RNA to direct its slicing. Even among this minority of target-cleaving piRNAs, most have no effect on the steady-state levels of their targets. Our data show that the efficacy of cleavage by most pachytene piRNAs is low, whereas target transcription rates are high, which may explain why piRNA-directed cleavage usually has little impact on the steady-state abundance of targets. Our analyses suggest that cleavage of the few targets for which abundance is reduced by piRNAs is essential to produce fully functional sperm. Because pachytene piRNAs that change steady-state levels of targets are rare, we propose that most such piRNAs are deleterious and are removed through purifying selection.

[1]RNA Therapeutics Institute and Howard Hughes Medical Institute, University of Massachusetts Chan Medical School, Worcester, MA, USA. [2]Department of Biology, New York University, New York, NY, USA. ✉e-mail: Phillip.Zamore@umassmed.edu; Ildar.Gainetdinov@nyu.edu

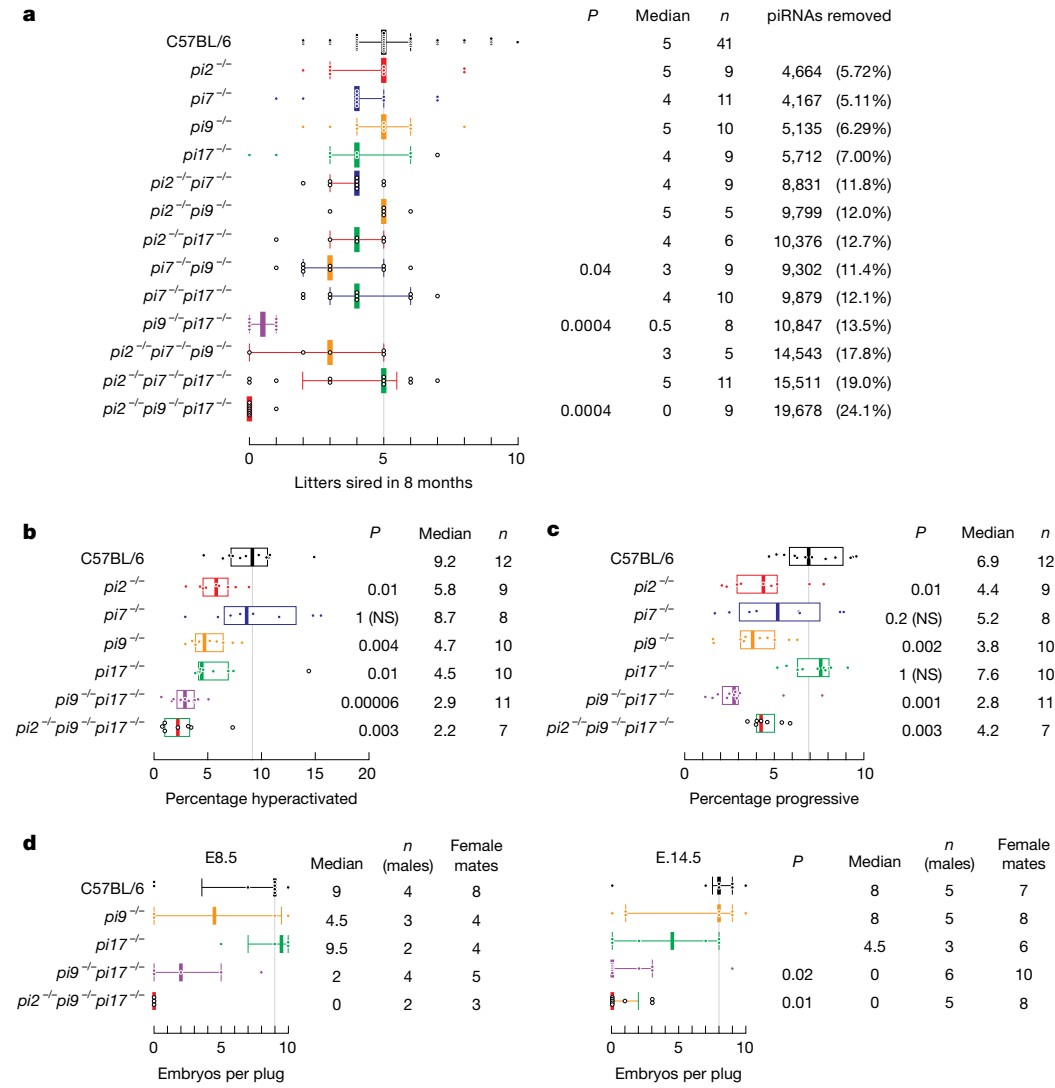

**Fig. 1 | Individual pachytene piRNA loci are required for the production of functional sperm. a**, Number of litters produced by control (C57BL/6) male mice and male mice with pachytene piRNA mutations in successive matings with C57BL/6 females over 8 months. Median and IQR values are shown. Kruskal–Wallis test (one-way analysis of variance (ANOVA) on ranks), $P = 7.3 \times 10^{-7}$. Benjamini–Hochberg-corrected $P$ values for post hoc pairwise Mann–Whitney tests are shown. **b**, Fraction of hyperactivated sperm from caudal epididymis of C57BL/6, $pi2^{-/-}$, $pi7^{-/-}$, $pi9^{-/-}$, $pi17^{-/-}$, $pi9^{-/-}pi17^{-/-}$ and $pi2^{-/-}pi9^{-/-}pi17^{-/-}$ male mice determined using CASAnova. Median and IQR values are shown. Kruskal–Wallis test (one-way ANOVA on ranks) $P = 7.9 \times 10^{-6}$. Benjamini–Hochberg-corrected $P$ values for post hoc pairwise two-tailed Mann–Whitney tests are shown. **c**, Fraction of progressive sperm from the caudal epididymis of C57BL/6, $pi2^{-/-}$, $pi7^{-/-}$, $pi9^{-/-}$, $pi17^{-/-}$, $pi9^{-/-}pi17^{-/-}$ and $pi2^{-/-}pi9^{-/-}pi17^{-/-}$ male mice determined using CASAnova. Median and IQR values are shown. Kruskal–Wallis test (one-way ANOVA on ranks) $P = 0.000048$. Benjamini–Hochberg-corrected $P$ values for post hoc pairwise two-tailed Mann–Whitney tests are shown. **d**, Number of embryos produced by males mated with C57BL/6 females at embryonic day 8.5 (E.8.5) or E14.5 after mating. Median and IQR values are shown. For E14.5, Kruskal–Wallis test (one-way ANOVA on ranks) $P = 0.044$; Benjamini–Hochberg-corrected $P$ values for post hoc pairwise two-tailed Mann–Whitney tests are shown. For E8.5, non-parametric Kruskal–Wallis and Mann–Whitney tests do not detect a difference owing to the lower number of replicates. NS, not significant.

Our data suggest that the fitness advantage provided by a minority of pachytene piRNAs ensures that the non-functional majority of pachytene piRNAs are retained in mammalian evolution.

## Redundancy among pachytene piRNA loci

The six largest sources of pachytene piRNAs ($pi2$, $pi6$, $pi7$, $pi9$, $pi17$ and $pi18$) in mice[15,17,21,25,26] (Supplementary Table 1) are syntenically conserved among placental mammals and produce around 40% of all pachytene piRNAs in mouse primary spermatocytes[6] (Supplementary Table 2). Yet genetic disruption of only two—$pi6$ and $pi18$ (refs. 17,21)—of the six loci leads to male infertility in mice. To determine whether the absence of a fertility phenotype for $pi2^{-/-}$, $pi7^{-/-}$, $pi9^{-/-}$ and $pi17^{-/-}$ mutations reflects apparent genetic redundancy, we generated male mice

with all possible double mutations and triple mutations and tested their fertility in successive matings with control C57BL/6 females (Fig. 1a and Extended Data Fig. 1). Two double-mutation combinations reduced male fertility. During eight months, $pi7^{-/-}pi9^{-/-}$ males sired a median of three litters and $pi9^{-/-}pi17^{-/-}$ males sired a median of only 0.5 litters compared with five for control C57BL/6 males. The $pi2^{-/-}pi9^{-/-}pi17^{-/-}$ male mice were nearly sterile.

The fertility defects observed in mice with double mutations or triple mutations suggest that $pi2$, $pi7$, $pi9$ and $pi17$ have a role in supporting spermatogenesis. Indeed, $pi2$, $pi9$ and $pi17$ are essential to produce fully mature spermatozoa. The fraction of hyperactivated sperm, a hallmark of $Ca^{2+}$-induced capacitation, was halved in mice with a $pi2^{-/-}$, $pi9^{-/-}$ or $pi17^{-/-}$ single mutation compared with C57BL/6 controls (C57BL/6, median 9.2%; $pi2^{-/-}$, median = 5.8%, unpaired, two-tailed Mann–Whitney

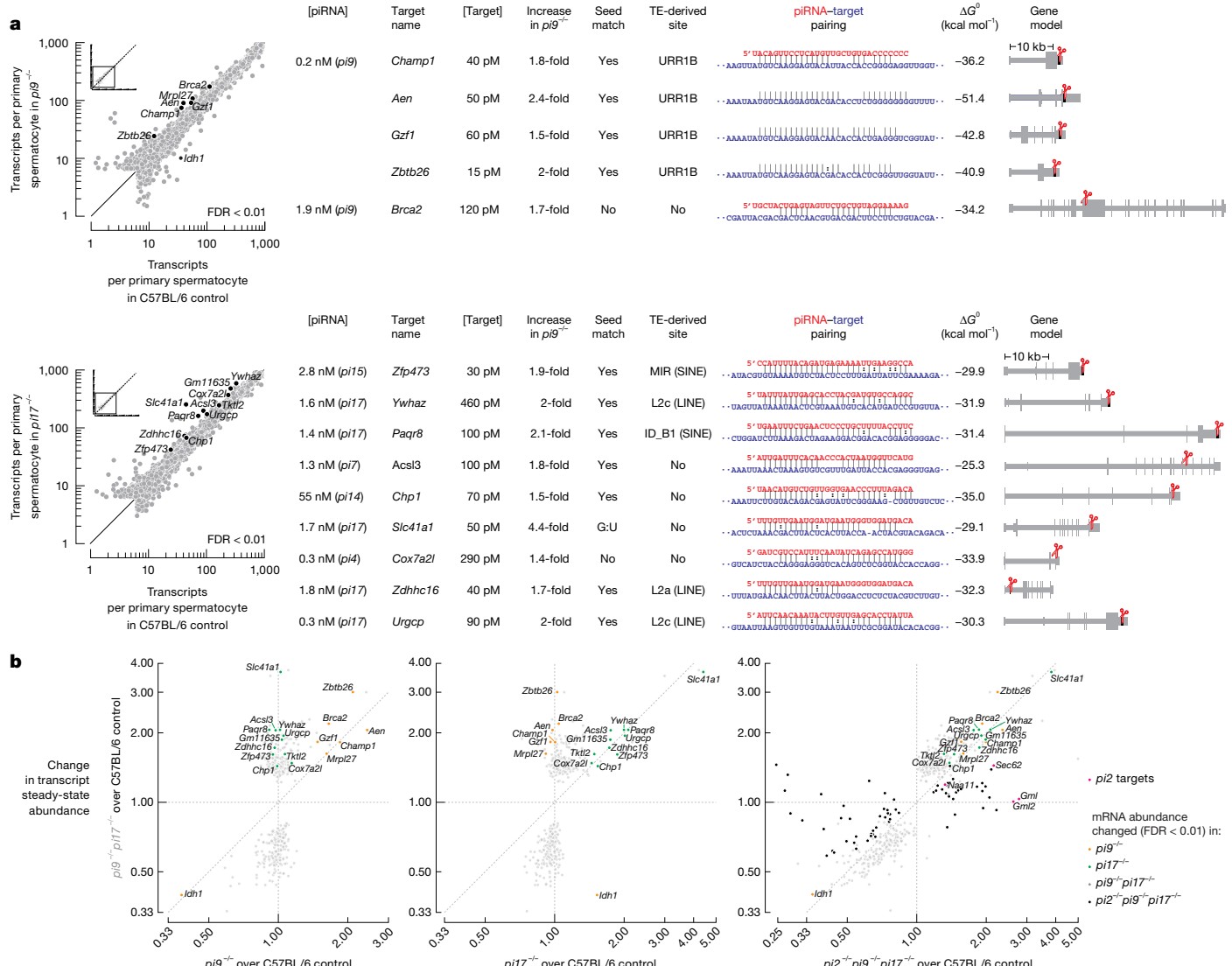

**Fig. 2 | Most mRNAs derepressed in *pi9*^−/−^ and *pi17*^−/−^ primary spermatocytes are direct targets of pachytene piRNAs. a**, Left, scatter plots of mean steady-state transcript abundance in FACS-sorted primary spermatocytes of C57BL/6 (*n* = 7), *pi9*^−/−^ (top; *n* = 9) and *pi17*^−/−^ (bottom; *n* = 8) male mice. Differentially expressed transcripts (FDR < 0.01) were identified using DESeq2 (Methods) and are indicated. Right, direct targets of *pi9* and *pi17* piRNAs. Mean piRNA (*n* = 12) and target (*n* = 7) concentrations in C57BL/6 primary spermatocytes, extent of target increase in *pi9*^−/−^ and *pi17*^−/−^ primary spermatocytes, target site location in transcript, piRNA–target pairing pattern and binding energy (computationally predicted Gibbs free energy, Δ*G*^0^) are shown. Presence of seed match (g2–g8) and whether a cleavage site is found in a transposon fragment is indicated.

Cleavage of *Cox7a2l*, *Urgcp*, *Acsl3*, *Chp1* and *Zfp473* mRNAs is guided by *4-qC5-17839*, *7-qD1-19431*, *7-qD2-24830*, *14-qA3-3095* and *15-qD1-17920* piRNAs, respectively. Biogenesis of these non-*pi17* piRNAs is initiated by *pi17* piRNA-directed cleavage. **b**, Scatter plots comparing the change in mean steady-state transcript abundance in FACS-sorted primary spermatocytes from *pi9*^−/−^ (*n* = 9), *pi17*^−/−^ (*n* = 8), *pi9*^−/−^*pi17*^−/−^ (*n* = 10) and *pi2*^−/−^*pi9*^−/−^*pi17*^−/−^ (*n* = 7) mice versus C57BL/6 controls (*n* = 7). Differentially expressed transcripts (FDR < 0.01) were identified using DESeq2 (Methods) and are shown in orange (changed in *pi9*^−/−^, *pi9*^−/−^*pi17*^−/−^ and *pi2*^−/−^*pi9*^−/−^*pi17*^−/−^), green (changed in *pi17*^−/−^, *pi9*^−/−^*pi17*^−/−^ and *pi2*^−/−^*pi9*^−/−^*pi17*^−/−^), grey (changed in *pi9*^−/−^*pi17*^−/−^ and *pi2*^−/−^*pi9*^−/−^*pi17*^−/−^) and black (changed in *pi2*^−/−^*pi9*^−/−^*pi17*^−/−^).

test, Benjamini–Hochberg (BH)-corrected *P* = 0.01 versus control; *pi9*^−/−^, median = 4.7%, *P* = 0.004; *pi17*^−/−^, median = 4.5%, *P* = 0.01; Fig. 1b). Moreover, mice with a *pi2*^−/−^ or *pi9*^−/−^ mutation had fewer progressively motile sperm (Fig. 1c). Our data therefore establish the importance of individual pachytene piRNA loci to produce fully functional sperm.

## Specific piRNAs explain mRNA repression

To investigate the molecular changes that underlie the defects observed in *pi9*^−/−^ and *pi17*^−/−^ sperm, we identified transcripts for which abundance was altered in *pi9*^−/−^ and *pi17*^−/−^ primary spermatocytes compared with C57BL/6 controls. Mice with a *pi9*^−/−^ or *pi17*^−/−^ mutation were generated using a pair of single guide RNAs (sgRNAs) to direct Cas9-mediated

deletion of promoter sequences (Supplementary Table 1). To exclude Cas9-induced off-target changes in gene expression, we used two different pairs of sgRNAs to generate two independent alleles for each locus and considered only molecular changes detected in both alleles. We report the smaller change in transcript abundance between the two alleles.

*pi9*^−/−^ and *pi17*^−/−^ mutations removed around 5,100 and 5,700 pachytene piRNA species, respectively (around 13.5% of all pachytene piRNAs in primary spermatocytes; Fig. 1a and Supplementary Table 2). Yet the steady-state abundance of just seven transcripts in *pi9*^−/−^ and 16 in *pi17*^−/−^ primary spermatocytes was significantly altered compared with C57BL/6 controls (false discovery rate (FDR) < 0.01; Fig. 2a, Methods and Supplementary Table 3). Among the 23 dysregulated transcripts

in *pi9*[-/-] and *pi17*[-/-] primary spermatocytes, 17 mRNAs and 5 lncRNAs were increased by 1.4–4.4-fold (median = 2-fold, interquartile range (IQR) = 1.7–2.4) and 1 mRNA was decreased by 2.5-fold (Fig. 2a and Supplementary Table 3).

Our analyses showed that of the 17 mRNAs derepressed in *pi9*[-/-] and *pi17*[-/-] primary spermatocytes, 14 were the targets of *pi9* or *pi17* piRNAs (Fig. 2a). First, we identified *pi9* and *pi17* piRNAs of sufficient intracellular abundance and complementarity to the 14 targets. No piRNA from outside *pi9* or *pi17* loci was predicted to cleave these 14 mRNAs[15] (Methods). Second, RNA slicing by PIWI proteins produces a 5′-monophosphate-bearing 3′ cleavage product. We sequenced 5′-monophosphorylated long RNAs from *pi9*[-/-], *pi17*[-/-] and *pi9*[-/-]*pi17*[-/-] primary spermatocytes. For each of the five mRNAs targeted by *pi9* piRNAs, the cleavage product was present in C57BL/6 and *pi17*[-/-] primary spermatocytes and was decreased by ≥8-fold in *pi9*[-/-] and *pi9*[-/-]*pi17*[-/-] primary spermatocytes (Extended Data Fig. 2a). For all nine targets of *pi17* piRNAs, cleavage products were detected in C57BL/6 and *pi9*[-/-] primary spermatocytes and were decreased by ≥8-fold in *pi17*[-/-] and *pi9*[-/-]*pi17*[-/-] primary spermatocytes (Extended Data Fig. 2b).

Our genetic data establish that removal of specific piRNAs reduces the abundance of the 3′ cleavage products and increases the levels of the predicted RNA targets. From these data, we infer that a piRNA directs the PIWI proteins MIWI (also known as PIWIL1) or MILI (also known as PIWIL2) to slice the predicted target. To directly test this inference, we incubated model target RNAs with recombinant MIWI that was programmed with four different synthetic piRNAs. A previously characterized *pi6* piRNA and its target site in the *Scpep1* mRNA served as a positive control[17,27]. We tested piRNA–target pairs with or without perfect pairing to the piRNA seed or nucleotides near the scissile phosphate (*Brca2*, *Zdhhc16*, *Urgcp* and *Cox7a2l*; Extended Data Fig. 3). Consistent with our in vivo data, we detected cleavage for all tested targets (Extended Data Fig. 3). Together, our data suggest that only a small number of mRNAs are regulated by *pi9* or *pi17* piRNA-directed slicing.

## *pi2*, *pi9* and *pi17* piRNA targets

Unlike *pi9*[-/-] and *pi17*[-/-], *pi9*[-/-]*pi17*[-/-] and *pi2*[-/-]*pi9*[-/-]*pi17*[-/-] male mice were infertile (Fig. 1a). The median number of embryos carried by females at day 14.5 after mating with *pi9*[-/-]*pi17*[-/-] or *pi2*[-/-]*pi9*[-/-]*pi17*[-/-] males was 0 compared with 8 embryos for females mated with control C57BL/6 males (Fig. 1d). *pi9*[-/-]*pi17*[-/-] and *pi2*[-/-]*pi9*[-/-]*pi17*[-/-] males did not exhibit changes in testicular germ cell composition or sperm gross morphology (Extended Data Fig. 4a). However, sperm from *pi9*[-/-]*pi17*[-/-] and *pi2*[-/-]*pi9*[-/-]*pi17*[-/-] mice were less plentiful, showed impaired motility and failed to penetrate the oocyte zona pellucida (Fig. 1b,c and Extended Data Fig. 1b–d). Caudal sperm from *pi9*[-/-]*pi17*[-/-] mice also contained deformed midpiece mitochondria (Extended Data Fig. 4b).

To identify the molecular cause of *pi9*[-/-]*pi17*[-/-] and *pi2*[-/-]*pi9*[-/-]*pi17*[-/-] fertility defects, we compared the transcriptomes of *pi9*[-/-], *pi17*[-/-], *pi9*[-/-]*pi17*[-/-] and *pi2*[-/-]*pi9*[-/-]*pi17*[-/-] primary spermatocytes. The abundance of *pi17* targets increased to the same extent in *pi17*[-/-], *pi9*[-/-]*pi17*[-/-] and *pi2*[-/-]*pi9*[-/-]*pi17*[-/-] primary spermatocytes (Fig. 2b). *pi9* targets were also derepressed to similar (±50%) extents in *pi9*[-/-], *pi9*[-/-]*pi17*[-/-] and *pi2*[-/-]*pi9*[-/-]*pi17*[-/-] males (Fig. 2b).

Moreover, the simultaneous removal of *pi9* and *pi17* piRNAs changed the abundance of many transcripts that are not direct piRNA targets (Fig. 2b). In addition to derepression of the 14 *pi9* and *pi17* target mRNAs, the abundance of 159 transcripts increased (1.2–2.2-fold, median = 1.5-fold, IQR = 1.4–1.6, FDR < 0.01) and 165 RNAs decreased (1.2–2.2-fold, median = 1.6-fold, IQR = 1.5–1.8, FDR < 0.01; Fig. 2b) in *pi9*[-/-]*pi17*[-/-] mice. Among these 324 transcripts, 323 lacked extensive complementarity to *pi9* or *pi17* piRNAs, and only a single mRNA, *Asb1*, was targeted by both a *pi9* and a *pi17* piRNA (Extended Data Fig. 5a).

The steady-state abundance of *Asb1* mRNA was unchanged in primary spermatocytes with a *pi9*[-/-] and *pi17*[-/-] single mutation but increased by 1.4-fold in *pi9*[-/-]*pi17*[-/-] primary spermatocytes. Because *Asb1* derepression was only detectable when *pi9* and *pi17* piRNAs were both removed, we conclude that *pi9* and *pi17* loci act redundantly to repress *Asb1* (Extended Data Fig. 5a).

Our data support the idea that the combined derepression of the 14 *pi9* and *pi17* targets explains the spermatogenic defects observed in *pi9*[-/-]*pi17*[-/-] male mice. *pi9* and *pi17* targets have distinct molecular functions yet act in the same pathways. Notably, three *pi9* and five *pi17* target mRNAs encode proteins implicated in the DNA damage response (*Brca2*), regulation of cell proliferation (*Gzf1*, *Ywhaz*, *Acsl3*, *Zdhhc16*, *Cox7a2l* and *Urgcp*) and apoptosis (*Aen*; Supplementary Table 4). Consistent with these molecular functions, testes from *pi9*[-/-]*pi17*[-/-] mice showed an increased incidence of double-stranded DNA breaks in testicular sperm (40 ± 10% tubules per section for *pi9*[-/-]*pi17*[-/-] versus 10 ± 9% tubules in controls; two-tailed *t*-test, Welch-corrected $P = 0.049$; Extended Data Fig. 6). The genomic instability observed in *pi9*[-/-]*pi17*[-/-] testicular sperm possibly reflects the previously reported consequences of supraphysiological levels of the proteins encoded by these target mRNAs. For example, overexpression of a RAD51-interacting domain of BRCA2 disrupts DNA repair by homologous recombination in human cultured cells[28,29]. Moreover, successful meiosis requires suppression of S phase during the second round of cell division[30], but overexpression of GZF1 (ref. 31), YWHAZ (ref. 32), ACSL3 (ref. 33), ZDHHC16 (ref. 34) or COX7A2L (ref. 35) stimulates mitosis in human immortalized cells and mouse cancer models. Furthermore, an increased abundance of URGCP stimulates the G1/S transition[36], and an increased level of AEN (apoptosis-enhancing nuclease) induces apoptosis and DNA fragmentation[37,38].

By contrast, the incidence of double-stranded DNA breaks in *pi6*[-/-] male mice was normal (20 ± 10% tubules for *p6*[-/-] versus 10 ± 9% for controls; two-tailed *t*-test, Welch-corrected $P = 0.45$; Extended Data Fig. 6). None of the eight mRNAs for which steady-state abundance increased in *pi6*[-/-] primary spermatocytes encode proteins expected to promote double-stranded DNA breaks[17].

As in *pi6*[-/-] and *pi18*[-/-] mice[17,21], transcripts derived from actively transposing repeat families with intact open-reading frames remained repressed in *pi9*[-/-], *pi17*[-/-], *pi9*[-/-]*pi17*[-/-] and *pi2*[-/-]*pi9*[-/-]*pi17*[-/-] males[39–41] (Supplementary Table 5). Together, these data show that cleavage of mRNAs by *pi9* or *pi17* piRNAs is required for normal spermatogenesis in mice.

## piRNA loci rarely have common targets

*Asb1* was the only mRNA we could identify that was co-regulated by both a *pi9* and a *pi17* piRNA. To find other transcripts co-regulated by piRNAs from different piRNA-producing loci, we examined data from *pi2*[-/-]*pi9*[-/-]*pi17*[-/-] males. Transcripts for which levels were altered in *pi9*[-/-]*pi17*[-/-] changed to a similar degree in *pi2*[-/-]*pi9*[-/-]*pi17*[-/-] primary spermatocytes (Fig. 2b, right). Consistent with the stronger fertility defects observed in *pi2*[-/-]*pi9*[-/-]*pi17*[-/-] mice than in *pi9*[-/-]*pi17*[-/-] mice (Fig. 1 and Extended Data Fig. 1), the abundance of an additional 59 transcripts was altered in mice with the triple mutation compared mice with the double mutation: 26 RNAs were increased (1.2–2.8-fold, median = 1.4-fold, IQR = 1.3–1.6, FDR < 0.01) and 33 transcripts were decreased (1.2–3.7-fold, median = 1.5-fold, IQR = 1.3–1.8, FDR < 0.01; Fig. 2b and Supplementary Table 3). Among the 26 RNAs derepressed in *pi2*[-/-]*pi9*[-/-]*pi17*[-/-] but not *pi9*[-/-]*pi17*[-/-] males, we identified 4 mRNAs cleaved exclusively by *pi2* piRNAs but could not find any transcripts repressed by piRNAs from *pi2* and *pi9* or *pi2* and *pi17*. The four *pi2* targets paired extensively and exclusively to *pi2* piRNAs, and the predicted 3′ cleavage products were detected in controls and were decreased by ≥8-fold in *pi2*[-/-]*pi9*[-/-]*pi17*[-/-] primary spermatocytes (Extended Data Fig. 5b). Together, these results suggest that

targets co-regulated by piRNAs from different piRNA-producing loci are rare.

## piRNAs rarely alter target abundance

Mouse primary spermatocytes contain about 81,600 distinct pachytene piRNA species for which concentrations range from 0.01 to 10 nM, of which *pi9* and *pi17* together produce about 11,000 (around 13.5%; Supplementary Table 2). Yet the abundance of just 17 mRNAs was significantly increased in *pi9*[−/−] and *pi17*[−/−] primary spermatocytes (FDR < 0.01; Fig. 2). Our analyses suggest that *pi9* and *pi17* piRNAs cleave >100 transcripts, but the abundance of only a small fraction of targets is influenced by piRNA cleavage.

PIWI protein-catalysed slicing produces 5′-monophosphorylated 3′ cleavage products. We searched our 5′-monophosphate RNA sequencing (RNA-seq) data for products of piRNA-directed cleavage. Putative cleavage targets of *pi9* or *pi17* piRNAs were required to be sufficiently complementary to *pi9* or *pi17* piRNAs[15] (Methods), but not to piRNAs from outside *pi9* or *pi17* loci. The predicted 3′ cleavage products were also required to be detectable in the C57BL/6 controls and to be decreased by ≥8-fold in *pi9*[−/−]*pi17*[−/−] primary spermatocytes.

A median of 112 putative *pi9* and *pi17* cleavage targets met these requirements (IQR = 86–123 targets for 16 permutations of 4 control and 4 *pi9*[−/−]*pi17*[−/−] replicates of 5′-monophosphorylated RNAs; Supplementary Table 6a). Using the same strategy, we identified putative targets of piRNAs not from *pi9* or *pi17*, requiring that the 3′ cleavage products be detectable in both the C57BL/6 control and *pi9*[−/−]*pi17*[−/−] primary spermatocytes (Supplementary Table 6b). Compared with the targets of non-*pi9* and non-*pi17* piRNAs, the steady-state levels of *pi9* and *pi17* target transcripts were significantly increased in *pi9*[−/−] *pi17*[−/−] primary spermatocytes (two-tailed Kolmogorov–Smirnov (KS) test median *P* = 0.002, IQR = 0.001–0.004; Fig. 3a, left). These results support the idea that pachytene piRNAs regulate gene expression through the cleavage of extensively complementary transcripts.

To test whether cleavage is necessary for piRNA-directed gene regulation, we repeated these analyses but selected putative piRNA targets without the requirement for a detectable 3′ cleavage product. Consistent with the idea that pachytene piRNAs regulate targets by cleavage, these analyses did not show differences in changes in steady-state levels in *pi9*[−/−]*pi17*[−/−] primary spermatocytes between *pi9* or *pi17* targets and non-*pi9* and non-*pi17* control targets (two-tailed KS test *P* = 0.21; Fig. 3a, right).

Thus, only approximately 1% (that is, 112 out of 11,000) of *pi9* and *pi17* piRNAs are sufficiently complementary to transcripts to direct their cleavage. Notably, for the 112 transcripts cleaved by *pi9* and *pi17* piRNAs, the median increase in their steady-state levels in *pi9*[−/−]*pi17*[−/−] primary spermatocytes was only around 4.9% (IQR = 3.2–7.3%, 16 permutations of 5′-monophosphorylated RNA data; Supplementary Table 6a). We conclude that most pachytene piRNA-directed cleavage events have a modest effect on target levels. Extrapolating from the number of transcripts targeted by *pi9* and *pi17* piRNAs (13.5% of all pachytene piRNAs), we estimate that about 830 (that is, 112/0.135) transcripts are pachytene piRNA cleavage targets in mouse primary spermatocytes.

## Target transcription and cleavage rates

The combined rates of transcription and decay determine steady-state transcript abundance. Our analyses suggest that PIWI cleavage has little effect on the majority of target RNAs because the efficiency of cleavage by most piRNAs is low and the transcription rates of targets are high. To compare slicing efficiency in vivo, we estimated the fraction cleaved for each target by calculating the ratio of 3′ cleavage product abundance to the steady-state level of the target. To compare the RNA polymerase II (PolII) density—a measure of transcription rate—of piRNA targets, we performed global run-on sequencing (GRO-seq) using nuclei from mouse primary spermatocytes purified by fluorescence-activated cell sorting (FACS).

We divided the targets of *pi9* and *pi17* into two classes: those that increased by ≤1.25-fold versus >1.25-fold in *pi9*[−/−]*pi17*[−/−] primary spermatocytes. Notably, the median cleavage efficiency for the ≤1.25-fold class was 3.5-fold lower than that of the >1.25-fold targets (two-tailed, unpaired Mann–Whitney test *P* = 10[−8]; Fig. 3b). Consistent with piRNA abundance being the major determinant of slicing efficacy[15], the ≤1.25-fold class of targets was cleaved by piRNAs for which the median concentration was half that of the piRNAs that cleaved the >1.25-fold targets (two-tailed, unpaired Mann–Whitney test *P* = 6 × 10[−6]; Fig. 3b).

High target transcription rates also precluded piRNAs from changing target levels. The median RNA PolII density for the ≤1.25-fold class was twice that of the >1.25-fold targets (changed ≤1.25-fold, median = 470 ppm, IQR = 390–530 ppm; changed >1.25-fold, median = 250 ppm, IQR = 230–270 ppm; two-tailed, unpaired Mann–Whitney test *P* = 4 × 10[−5]; Fig. 3b). Moreover, transcripts cleaved by pachytene piRNAs generally had higher than typical transcription rates. We compared the RNA PolII density for genes with no evidence of piRNA-directed transcript slicing to that of the putative targets of all mouse pachytene piRNAs. All transcripts present in primary spermatocytes at >5 copies per cell were searched for pairing to pachytene piRNAs. Transcripts were considered pachytene piRNA targets if piRNA complementarity was predicted to be sufficient to trigger cleavage[15] and the putative cleavage product was detected in control C67BL/6 primary spermatocytes. Transcripts with <15 total nucleotides of complementarity to pachytene piRNAs were defined as non-target controls. Genes targeted by pachytene piRNAs had around 3-fold higher RNA PolII signal (median = 370 ppm, IQR = 330–430 ppm; two-tailed, unpaired Mann–Whitney test *P* = 7 × 10[−7]) than controls (median = around 123 ppm, IQR = 122–124 ppm; Fig. 3c).

We conclude that cleavage by pachytene piRNAs has a limited impact on transcript steady-state levels owing to low piRNA-directed slicing efficiency and high target transcription rates. Because pachytene piRNAs that change transcript abundance are rare, we propose that most such piRNAs are removed through purifying selection because their appearance during evolution is more frequently detrimental than advantageous for sperm fitness.

## Cleavage explains repression by piRNAs

The *pi6*[−/−] mutation removes around 4,000 piRNAs but increases the steady-state abundance of just 24 mRNAs in primary spermatocytes, secondary spermatocytes and spermatids. A previous study showed that the altered abundance of just 6 out of the 24 mRNAs could be explained by piRNA-directed target cleavage[17]. To identify additional regulatory targets of *pi6* piRNAs, we re-examined the *pi6* data using recently discovered piRNA targeting requirements for extent of complementarity and piRNA abundance[15]. Because 5′-monophosphate-bearing cleavage products are short lived in cells[42], we sequenced 5′ monophosphorylated RNAs from control C57BL/6 and *pi6*[−/−] primary spermatocytes 20-fold more deeply than previously[17]. For *pi6* targets, the putative cleavage products were required to be present in C57BL/6 and undetectable in *pi6*[−/−] primary spermatocytes (Extended Data Fig. 7).

These analyses identified an additional ten piRNAs that explained the increased abundance of eight mRNAs and two lncRNAs in *pi6*[−/−] spermatocytes and spermatids (Extended Data Fig. 7). The remaining mRNAs for which derepression in *pi6*[−/−] mice is not explained by cleavage directed by a piRNA from *pi6* had a median steady-state abundance of ten molecules per primary spermatocyte (Supplementary Table 2), an abundance that is probably too low to permit detection of the short-lived cleavage products derived from these transcripts (Supplementary Discussion). Thus, piRNA-directed cleavage explains at least 14 out of the 24 mRNAs derepressed in *pi6*[−/−] mice.

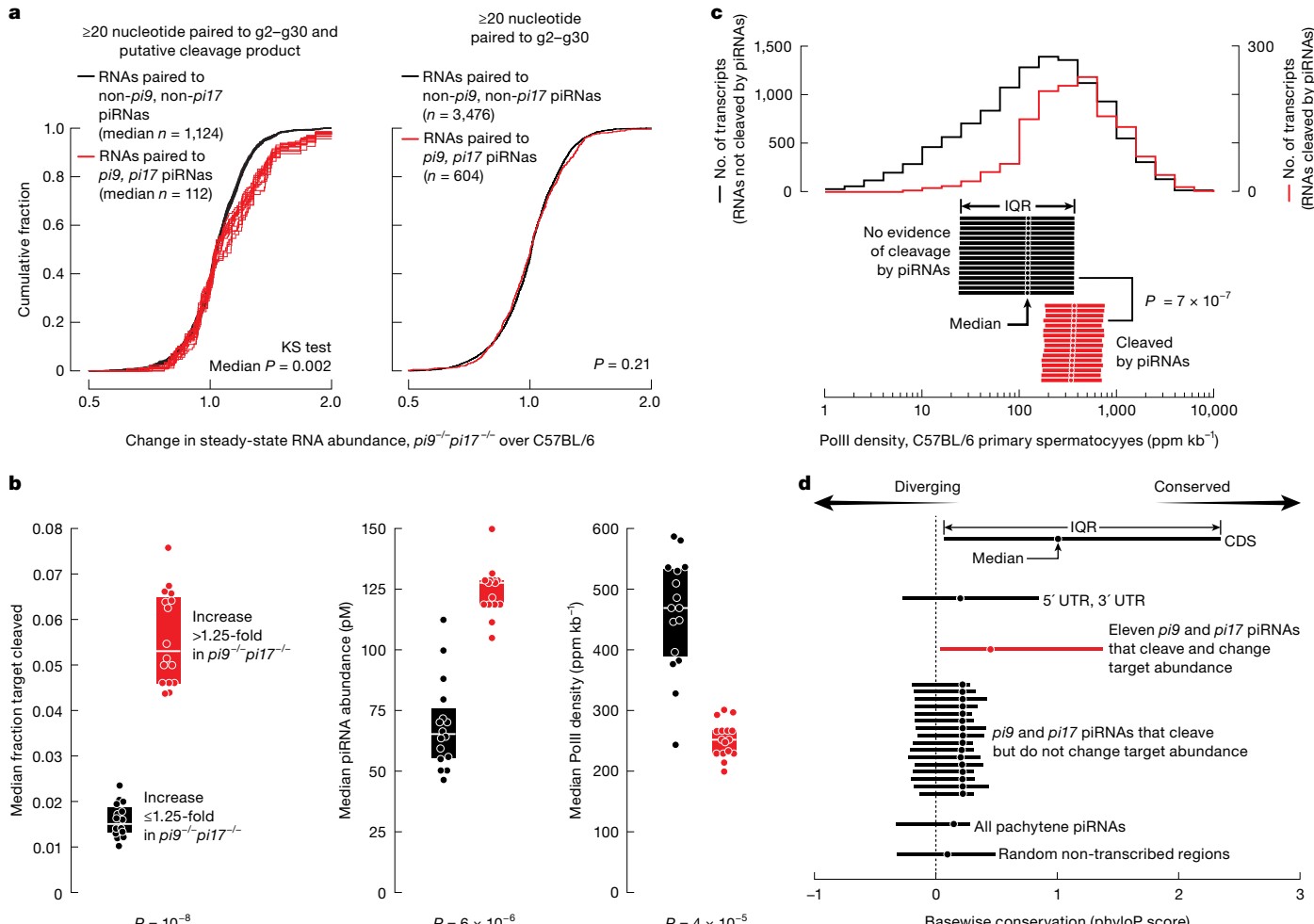

**Fig. 3 | Pachytene piRNAs regulate targets through endonucleolytic cleavage. a**, Change in mean steady-state abundance in *pi9⁻/⁻ pi17⁻/⁻* (*n* = 10) versus C57BL/6 (*n* = 7) primary spermatocytes for transcripts that pair with ≥20 nucleotides to *pi9* and *pi17* piRNAs or to control piRNAs (that is, piRNAs for which abundance does not change in *pi9⁻/⁻ pi17⁻/⁻* primary spermatocytes). Left, data are only for transcripts with detectable cleavage products; *n* = 16 permutations of 4 control and 4 *pi9⁻/⁻ pi17⁻/⁻* replicates of 5′-monophosphorylated RNAs. Two-tailed KS test *P* values are shown. **b**, Median fraction of target cleaved, piRNAs abundance and RNA PolII density (GRO-seq) in C57BL/6 primary spermatocytes for targets of *pi9* and *pi17* piRNAs that increased by ≥1.25-fold or by ≤1.25-fold in *pi9⁻/⁻ pi17⁻/⁻* versus C57BL/6 primary spermatocytes; *n* = 16 permutations of 4 C57BL/6 and 4 *pi9⁻/⁻ pi17⁻/⁻* 5′-monophosphate sequencing replicates. Unpaired, two-tailed Mann–Whitney test *P* value is shown. piRNA abundance (*n* = 12) and GRO-seq data (*n* = 3) are the mean. Boxplots show median

and IQR values. **c**, RNA PolII density in C57BL/6 primary spermatocytes for transcripts with and without detectable pachytene piRNA-directed cleavage. Top, representative distribution of RNA PolII density for one pair of C57BL/6 and *pi9⁻/⁻ pi17⁻/⁻* 5′-monophosphate sequencing replicates. Bottom, median and IQR values for all 16 permutations of 4 C57BL/6 and 4 *pi9⁻/⁻ pi17⁻/⁻* 5′-monophosphate sequencing replicates. Unpaired, two-tailed Mann–Whitney test *P* value is shown. GRO-seq data are the mean (*n* = 3) ppm kb⁻¹. **d**, Base-wise conservation (PhyloP score) for genomic origins of all pachytene piRNAs, *pi9* and *pi17* piRNAs with identifiable cleavage targets (16 permutations of 4 C57BL/6 and 4 *pi9⁻/⁻ pi17⁻/⁻* 5′-monophosphate sequencing replicates) and 11 *pi9* and *pi17* piRNAs that change target abundance shown in Fig. 2. Median and IQR values for piRNA nucleotides g2–g30 are shown. For non-transcribed regions, 5′ UTRs, 3′ UTRs and coding sequences, 80,000 random 29-nucleotide-long segments were sampled from each category.

We conclude that the main mechanism by which pachytene piRNAs regulate gene expression is PIWI-protein-catalysed endonucleolytic cleavage.

## Evolutionary drift of regulatory piRNAs

Our data suggest that around 99% of pachytene piRNAs have no cleavage target, which may explain the rapid divergence of most pachytene piRNA sequences among mammals[6] (Fig. 3d). Consistent with random genetic drift in the absence of selective pressure, both piRNAs without cleavage targets and piRNAs that cleave transcripts but do not change their steady-state levels displayed a low degree of conservation. That is, the median base-wise conservation score was 0.22 (IQR = −0.18 + 0.34) for piRNAs with cleavage targets versus 0.15 (IQR = −0.33 + 0.28) for piRNAs without targets (Fig. 3d).

Pachytene piRNAs that changed target abundance also showed rapid evolutionary turnover. Of the 14 *pi9* or *pi17* piRNA–target pairs (Fig. 2), the sequences of just 5 piRNAs and their corresponding mRNA target sites (*Brca2*, *Slc41a1*, *Urgcp*, *Ywhaz* and *Acsl3*) are conserved among placental mammals (87 million years ago)[43]. The other nine piRNA–target pairs are found only in the Murinae subfamily of rodents (13 million years ago)[43]. Thus, pachytene piRNA-mediated regulation of most mRNA targets in mice and rats emerged recently.

Only around 17% of pachytene piRNAs arise from sequences derived from inactive transposons[16], yet more than half of the *pi9* and *pi17* piRNAs that change target abundance derive from repeat elements (Fig. 2). Notably, a single abundant piRNA derived from a URR1B DNA transposon insertion in the *pi9* locus regulates the steady-state levels of four mRNAs (*Champ1*, *Aen*, *Gzf1* and *Zbtb26*) via imperfect complementarity to insertions of the same transposon family in the 3′ untranslated

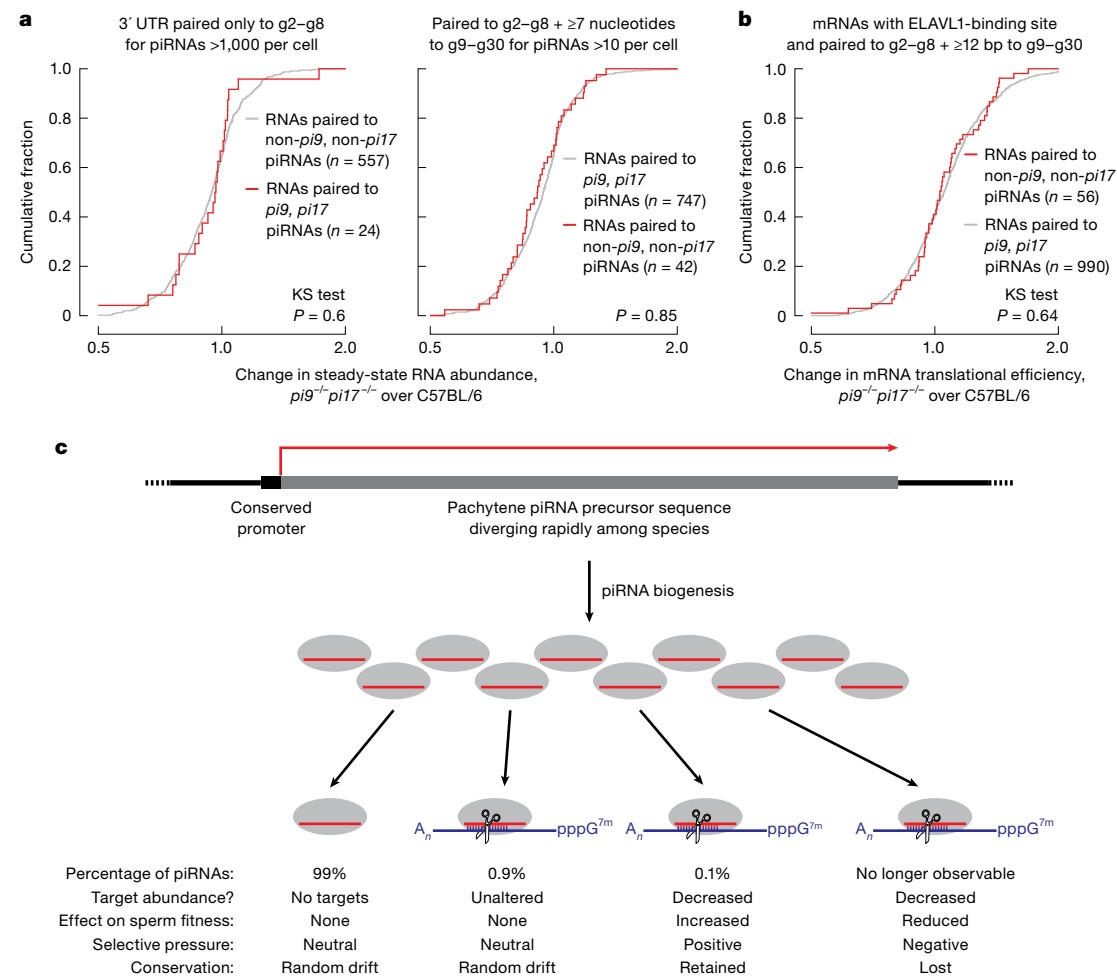

**Fig. 4 | Pachytene piRNAs do not regulate mRNA translation and are dispensable for germ granule formation. a**, Change in mean steady-state abundance in FACS-purified $pi9^{-/-}pi17^{-/-}$ ($n = 10$) versus C57BL/6 ($n = 7$) primary spermatocytes for mRNAs for which 3' UTRs pair to $pi9$ and $pi17$ piRNAs or to control piRNAs (that is, piRNAs for which abundance does not change in $pi9^{-/-}pi17^{-/-}$ primary spermatocytes). Two-tailed KS test $P$ values are shown. **b**, Change in mean translational efficiency in FACS-purified $pi9^{-/-}pi17^{-/-}$ ($n = 4$) versus C57BL/6 ($n = 4$) primary spermatocytes for mRNAs for which 3' UTRs pair to $pi9$ and $pi17$ piRNAs or to control piRNAs (that is, piRNAs for which abundance does not change in $pi9^{-/-}pi17^{-/-}$ germ cells). Two-tailed KS test $P$ values are shown. **c**, A model for the origins of 'selfish' pachytene piRNAs. The model seeks to account for our observation that 99.9% of pachytene piRNAs probably have no detectable biological function.

regions (UTRs) of the targets (Fig. 2a). Moreover, insertion of URR1B into the target 3' UTRs and the $pi9$ locus are found only in mice and rats.

In fact, repeat-derived sequences occurred 2.5-fold more frequently among the $pi9$ and $pi17$ piRNAs that cleave targets without changing their abundance than in piRNAs with no identifiable cleavage target (median = 41%, IQR = 37–46% for pachytene piRNAs with cleavage targets versus 17% for piRNAs without detectable targets; one-sample, two-tailed $t$-test $P = 10^{-5}$; Supplementary Table 6a). All transposon-derived pachytene piRNAs with cleavage targets were derived from repeats that are now transpositionally inactive[44] (Supplementary Table 6a). Together, these results demonstrate that piRNA-mediated gene regulation exhibits rapid turnover during mammalian evolution and suggest that repeat-derived pachytene piRNAs have an important role in the acquisition of new targets.

## Pachytene piRNAs do not act like miRNAs

Pachytene piRNAs are proposed to bind mRNAs via seed complementarity (piRNA nucleotides g2–g8) and recruit the deadenylase CAF1 to destabilize the targets, a mechanism used by miRNAs[18,45]. To test this model, we identified the potential miRNA-like targets of $pi9$ and $pi17$ piRNAs. That is, mRNAs with seed complementarity to abundant

(>1,000 molecules per cell) $pi9$ and $pi17$ piRNAs. Among the 24 putative miRNA-like targets, none were derepressed in $pi9^{-/-}pi17^{-/-}$ primary spermatocytes, secondary spermatocytes, or round or elongating spermatids (Fig. 4a and Extended Data Fig. 8a). A $pi17$ piRNA was reported to destabilize the *Grk4* mRNA in elongating spermatids[18], yet *Grk4* steady-state level did not change in $pi9^{-/-}pi17^{-/-}$ male mice (Extended Data Fig. 8b). Our data argue that pachytene piRNAs do not regulate mRNA expression through a miRNA-like mechanism.

## piRNAs do not regulate translation

The protein ELAVL1 has been proposed to collaborate with pachytene piRNAs to activate translation of partially complementary mRNAs[22,23]. To test this mechanism for pachytene piRNA-mediated gene regulation, we sequenced polyadenylated RNAs and ribosome footprints from FACS-purified primary spermatocytes, secondary spermatocytes and round spermatids from $pi9^{-/-}pi17^{-/-}$ males and controls (Supplementary Table 7 and Supplementary Fig. 2). Translation is delayed for many mRNAs involved in late spermatogenesis[46], and, as expected, mRNAs for which translational efficiency increased in round spermatids were enriched for transcripts required for the development of mature sperm (Supplementary Table 8 and Supplementary Fig. 3).

By contrast, our analyses do not support the model for pachytene piRNA-directed activation of mRNA translation. Loss of translation-activating piRNAs is predicted to decrease ribosome occupancy on their mRNA targets. We identified mRNAs for which 3′ UTRs contained both an ELAVL1-binding motif and a piRNA target site complementary to the piRNA seed (nucleotides g2–g8) and an additional ≥12 nucleotides in the piRNA region g9–g30 (refs. 22,23). We divided these putative targets into two types: those predicted to be regulated by piRNAs from *pi9* or *pi17* and piRNAs from neither *pi9* nor *pi17*. For each target type, we calculated the change in translational efficiency in $pi9^{-/-} pi17^{-/-}$ mice compared with controls. The change in translational efficiency was indistinguishable for the two target types in primary spermatocytes (two-tailed KS test $P = 0.64$), secondary spermatocytes ($P = 0.09$) and round spermatids ($P = 0.37$; Fig. 4b and Extended Data Fig. 9a). We conclude that piRNAs from *pi9* or *pi17* do not activate translation in collaboration with ELAVL1.

*Tbpl1*, *Cnot4* and *Spesp1* have been reported to be translationally activated in round spermatids by piRNAs from *pi17* (refs. 22,23), yet we did not detect changes in ribosome occupancy or mRNA abundance for any of these three mRNAs in $pi9^{-/-} pi17^{-/-}$ primary spermatocytes, secondary spermatocytes or round spermatids (Extended Data Fig. 9b, Supplementary Table 7 and Supplementary Figs. 4–7). Finally, translational efficiency was increased for 36 mRNAs in primary spermatocytes (1.3–2.0-fold, median = 1.35-fold), 44 mRNAs in secondary spermatocytes (1.3–1.6-fold, median = 1.35-fold) and 81 mRNAs in round spermatids (1.3–1.9-fold, median = 1.4-fold) in $pi9^{-/-} pi17^{-/-}$ mice compared with controls (FDR <0.01 and ≥10 transcripts per million (TPM) for ribosome occupancy in C57BL/6 controls; Extended Data Fig. 9b, Supplementary Table 7 and Supplementary Fig. 7). Yet we were unable to identify *pi9* or *pi17* piRNAs with ≥15 total nucleotides of complementary to any of these mRNAs. We conclude that pachytene piRNAs neither activate nor repress translation of partially complementary mRNAs.

## Discussion

Our data suggest that pachytene piRNAs mainly—perhaps exclusively—regulate their targets through siRNA-like endonucleolytic cleavage. We did not find support for pachytene piRNAs directing translational activation or repression or guiding miRNA-like regulation of mRNA stability. Major pachytene piRNA-producing loci are also dispensable for germ cell granule formation (Extended Data Fig. 10 and Supplementary Discussion).

Among the tens of thousands of distinct pachytene piRNA species present at ≥10 pM in mouse primary spermatocytes, only around 1% are sufficiently complementary to a transcript to direct its cleavage. Notably, pachytene piRNA-directed cleavage rarely changes the steady-state levels of target RNAs, which may be because of the low efficiency of slicing by most pachytene piRNAs and the high transcription rates of their targets (Supplementary Discussion). piRNA-directed cleavage of the few targets for which abundance does decrease seems to be essential for spermatogenesis (Fig. 4c).

Our data support the idea that most pachytene piRNAs do not have a biological target[47]. This is because either they are sufficiently complementary to cleave an RNA but cleavage does not alter its steady-state abundance or are not complementary to any transcript. These results explain the poor sequence conservation of most pachytene piRNAs[6]. Pachytene piRNA biogenesis is highly interdependent, which makes the few functional piRNAs require production of the apparently dispensable majority[16,17]. We propose that most pachytene piRNAs are retained because the regulation of a small number of targets by a few functional piRNAs improves spermatogenesis. Future experiments, such as deletion of individual piRNA target sites or piRNA sequences, should help test this 'piRNA addiction' model (Fig. 4c).

Rapid genetic drift of the majority of pachytene piRNA sequences among species may have important implications for mammalian

evolution. For example, rapidly diverging pachytene piRNAs may create reproductive barriers between emerging mammalian lineages and therefore drive speciation. Together, our findings uncover how a pathway designed to protect genomes from invading nucleic acids created a population of 'selfish' small RNAs that have perpetuated themselves for the past hundred million years of mammalian evolution.

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

# Methods

## Mouse strains and mutants

Mice (wild-type C57BL/6J, IMSR, JAX: 000664, RRID: IMSR_JAX:000664; *Miwi*[−/−] mutation, MGI: 2182488; and pachytene piRNA mutations listed in Supplementary Table 1) were housed and euthanized according to the guidelines of the Institutional Animal Care and Use Committee of the University of Massachusetts Chan Medical School in an Association for Assessment and Accreditation of Laboratory Animal Care International-accredited barrier facility at controlled temperature ($22 \pm 2\,°C$), relative humidity ($40 \pm 15\%$) and a 12-h day–light cycle. All experimental animals were 2–6 months old[6,15,48]. sgRNAs (Supplementary Table 1) were designed using a CRISPR design tool (https://www.idtdna.com/site/order/designtool/index/CRISPR_SEQUENCE). sgRNAs were transcribed with T7 RNA polymerase and then purified by electrophoresis on 10% denaturing polyacrylamide gels. gRNA (20 ng μl$^{-1}$) and *Cas9* mRNA (50 ng μl$^{-1}$, TriLink Biotechnologies, L-7206) were injected together into the pronucleus of one-cell C57BL/6 zygotes in M2 medium (Sigma, M7167). After injection, the zygotes were cultured in EmbryoMax Advanced KSOM medium (Sigma, MR-106-D) at 37 °C under 5% $CO_2$ until the blastocyst stage (3.5 days), then transferred into the uterus of pseudopregnant ICR females 2.5 days post coitum. To screen for founders with the mutation, gDNA extracted from tail tissues was analysed by PCR using the primers listed in Supplementary Table 1. All mutant strains were maintained in a C57BL/6 background; all experimental animals were the progeny of at least two backcrosses.

## Mouse fertility

Fertility was measured as previously described[6,15,48]. In brief, each 2–6-month-old male mouse was continuously housed with one 2–4-month-old C57BL/6 female. For male mice that did not produce pups after 3 months (around 3 cycles), the original female was replaced with a new female and the fertility test continued.

To generate E8.5 or E14.5 embryos, one male mouse was housed with two C57BL/6 females. When a copulatory plug was observed, the female was housed separately until the experiment was completed.

## Epididymal sperm count

Sperm counts were obtained as previously described[6,15,48]. In brief, to quantify sperm abundance, cauda epididymides were collected from mice and placed in PBS. A few incisions were made in the epididymides with scissors to release the sperm, followed by incubation at 37 °C and 5% $CO_2$ for 20 min. A 20 μl aliquot of sperm suspension was diluted in 480 μl of 1% (w/v) paraformaldehyde (PFA) and sperm cells were counted using a Leica DMi8 bright-field microscope equipped with a ×10, NA 0.4 objective.

## TUNEL immunohistochemistry

Mouse testes were fixed in Bouin's solution overnight, washed with 70% ethanol, embedded in paraffin and sectioned at 5 μm thickness. A Click-iT TUNEL Colorimetric IHC Detection kit (Thermo Fisher, C10625) was used to detect DNA breaks according to the manufacturer's protocol. In brief, testes were fixed and embedded as described above, then were de-paraffinized in three changes of xylene for 5 min each, gradually re-hydrated in 100% (v/v), 95% (v/v) and 70% (v/v) ethanol for 5 min each, and then washed in 1× PBS for 5 min. After pretreating the slides with 20 μg ml$^{-1}$ proteinase K at room temperature for 15 min, slides were washed with water twice (2 min each). Positive-control slides were treated with 1.0 U Turbo DNase (Thermo Fisher, AM2238) at room temperature for 30 min. Slides were then incubated with TdT reaction buffer containing terminal deoxynucleotidyl transferase in a humidified chamber at 37 °C for 1 h. The reaction was quenched with 2× SSC for 15 min, then washed twice in PBS. Peroxidase activity was quenched in 3% (v/v) $H_2O_2$ at room temperature for 5 min. Slides were incubated with biotin azide and copper sulfate in a humidified chamber at 37 °C for 30 min, then stained with peroxidase substrate at room temperature for 10 min. Nuclei were counterstained with haematoxylin I, and the slides were sealed with EcoMount (Biocare Medical, EM897L). Images were captured using a Leica DMi8 bright-field microscope equipped with a ×20 objective with 0.4 NA (HC PL FL L ×20/0.40 CORR PH1, Leica Microbiosystems).

## In vitro fertilization and embryo transfer

In vitro fertilization (IVF) was performed as previously described[6,15,48,49]. In brief, using spermatozoa from caudal epididymis of C57BL/6 or *pi9*[−/−]*pi17*[−/−] mice, spermatozoa were incubated in complete human tubal fluid medium (101.6 mM NaCl, 4.69 mM KCl, 0.37 mM $KH_2PO_4$, 0.2 mM $MgSO_4 \cdot 7H_2O$, 21.4 mM sodium lactate, 0.33 mM sodium pyruvate, 2.78 mM glucose, 25 mM $NaHCO_3$, 2.04 mM $CaCl_2 \cdot 2H_2O$, 0.075 mg ml$^{-1}$ penicillin-G, 0.05 mg ml$^{-1}$ streptomycin sulfate, 0.02% (v/v) phenol red and 4 mg ml$^{-1}$ BSA) with oocytes from B6SJLF1/J mice for 3–4 h at 37 °C with constant 5% $O_2$, 90% $N_2$ and 5% $CO_2$. Oocyte viability and the presence of pronuclei were assessed using a Nikon SMZ-2B (Nikon) dissecting microscope with a ×5, NA 0.6 objective. To observe embryo development, embryos were moved into potassium-supplemented simplex optimized medium (KSOM; 95 mM NaCl, 2.5 mM KCl, 0.35 mM $KH_2PO_4$, 0.2 mM $MgSO_4 \cdot 7H_2O$, 10 mM sodium lactate, 0.2 mM sodium pyruvate, 0.2 mM glucose, 25 mM $NaHCO_3$, 1.71 mM $CaCl_2 \cdot 2H_2O$, 1 mM L-glutamine, 0.01 mM EDTA, 0.075 mg ml$^{-1}$ penicillin-G, 0.05 mg ml$^{-1}$ streptomycin sulfate, 0.02% (v/v) phenol red and 1 mg ml$^{-1}$ BSA; Millipore Sigma) after IVF and assessed every 24 h. To measure birth rates, two-cell embryos were transferred to Swiss Webster pseudopregnant females, and fetuses were isolated by caesarean section 18.5 days after embryo transfer. For zona-free IVF, the zona pellucida of oocytes was removed with acid Tyrode's solution as previously described[50,51].

## Sperm motility

Cauda epidydimal sperm motility was measured as previously described[6,15,48]. In brief, sperm were collected from mice and placed in warm human tubal fluid medium in a 37 °C incubator with 5% $CO_2$. A drop of sperm was removed from the suspension and pipetted into a sperm counting glass chamber, then assayed by CASA or video acquisition. CASA was conducted using an IVOS II instrument (Hamilton Thorne) with the following settings: 100 frames acquired at 60 Hz; minimal contrast, 50; 4-pixel minimal cell size; minimal static contrast, 5; 0% straightness (STR) threshold; 10 μm s$^{-1}$ VAP cutoff; prog. min VAP, 20 μm s$^{-1}$; 10 μm s$^{-1}$ VSL cutoff; 5-pixel cell size; cell intensity, 90; static head size, 0.30–2.69; static head intensity, 0.10–1.75; static elongation, 10–94; slow cells motile, yes; ×0.68 magnification; LED illumination intensity, 3,000; IDENT illumination intensity, 3,603; 37 °C. The raw data files (that is, .dbt files for motile sperm and .dbx files for static sperm) were used for sperm motility analyses. For motile sperm, only those for which movement was captured with ≥45 consecutive frames were analysed. For progressive or hyperactivated motility analyses, .dbt files of motile sperm were used as input for CASAnova, as previously described[52].

## Transmission electron microscopy

Mouse testis and caudal epididymides were dissected and immediately fixed by immersion in Karnovsky's fixative (2% formaldehyde (v/v) and 3% glutaraldehyde (v/v) in 0.1 M sodium phosphate buffer, pH 7.4; Electron Microscopy Sciences) overnight at 4 °C and washed 3 times in 0.1 M phosphate buffer. Following the third wash, the tissues were post-fixed in 1% osmium tetroxide (w/v; Electron Microscopy Sciences) for 1 h at room temperature, washed 3 more times with water for 10 min each and dehydrated using a graded series of 30%, 50%, 70%, 85%, 95% and 100% (three changes) ethanol and 100% propylene oxide (two changes) and a mixture of 50% propylene oxide (v/v) and 50% SPI-Pon 812 resin mixture (v/v; SPI Supplies). The sample was incubated in seven successive changes of SPI-Pon 812 resin over 3 days, polymerized at 68 °C in

flat moulds and reoriented to allow cross-sectioning of spermatozoa in the lumen of epididymis. Sections measuring 70 nm were cut on a Leica EM UC7 ultramicrotome (Leica Microsystems) using a diamond knife, collected on copper mesh grids and stained with 3% lead citrate (w/v) and 0.1% uranyl acetate (w/v) to increase contrast. Finally, sections were examined using a Philips CM10 transmission electron microscope (Philips Electron Optics) at 100 kV. Images were recorded using an Erlangshen digital camera system (Gatan).

### piRNA loading and recombinant piRISC purification for MIWI

Recombinant MIWI loading was done as previously described[6,15,48]. In brief, synthetic piRNA guides (Extended Data Fig. 3) were ordered from IDT and purified by electrophoresis through a 15% denaturing polyacrylamide gel. HEK293T cells (American Type Culture Collection) expressing SNAP-tagged, 3×Flag-tagged MIWI were generated as previously described[27]. Cells were collected at around 70% confluency using a TC Cell Scraper (ThermoFisher, 50809263) into ice-cold PBS and collected by centrifugation at 500$g$. Supernatant was removed, and the pellet was stored at −80 °C until lysed in 10 ml of 30 mM HEPES-KOH, pH 7.5, 100 mM potassium acetate, 3.5 mM magnesium acetate, 2 mM DTT, 0.1% (v/v) Triton X-100, 15% (v/v) glycerol and 1× protease inhibitor cocktail (1 mM 4-(2-aminoethyl)benzenesulfonyl fluoride hydrochloride (Sigma; A8456), 0.3 μM aprotinin, 40 μM betanin hydrochloride, 10 μM E-64 (Sigma; E3132) and 10 μM leupeptin hemisulfate) per gram of frozen cells. Cell lysis was monitored by staining with trypan blue. Crude cytoplasmic lysate was clarified at 20,000$g$, flash-frozen in liquid nitrogen and stored at −80 °C.

To capture MIWI, 1 ml of clarified lysate was incubated with 20 μl anti-Flag M2 paramagnetic beads (Sigma, M8823) for 4 h or overnight rotating at 4 °C. Beads were washed 4 times with extract buffer (30 mM HEPES-KOH, pH 7.5, 3.5 mM magnesium acetate, 2 mM DTT, 15% (v/v) glycerol and 0.01% (v/v) Triton X-100) containing 2 M potassium acetate and 4 times with extract buffer containing 100 mM potassium acetate. To assemble MIWI piRISC, beads were resuspended in extract buffer containing 100 mM potassium acetate and 100 nM synthetic piRNA guide (Fig. 2a) and incubated with rotation for 30 min at 37 °C or room temperature. After 5 washes in 2 M potassium acetate extract buffer and 5 washes in 100 mM potassium acetate extract buffer, MIWI piRISC was eluted from the beads twice with 200 ng μl⁻¹ 3×Flag peptide in 100 μl of 100 mM potassium extract buffer with rotation for 1 h at room temperature. The combined 200 μl eluate was flash-frozen in liquid nitrogen and stored at −80 °C.

### Recombinant mouse GTSF1 purification

Recombinant mouse GTSF1 was purified as previously described[6,15,48]. In brief, pCold-GST(glutathione S-transferase) GTSF-expression vectors were transformed into Rosetta-Gami 2 competent cells (Sigma, 71351). Cells were grown to an $OD_{600}$ of about 0.6–0.8 in the presence of 1 μM $ZnSO_4$ at 37 °C, then chilled on ice for 30 min to initiate cold shock. Protein expression was induced with 0.5 mM IPTG for 18 h at 15 °C. Cells were collected by centrifugation, washed twice with PBS, and cell pellets were flash frozen and stored at −80 °C. Cell pellets were resuspended in lysis/GST column buffer containing 20 mM Tris-HCl pH 7.5, 500 mM NaCl, 1 mM DTT, 5% (v/v) glycerol and 1× protease inhibitor cocktail (1 mM 4-(2-Aminoethyl)benzenesulfonyl fluoride hydrochloride (Sigma; A8456), 0.3 μM aprotinin, 40 μM betanin hydrochloride, 10 μM E-64 (Sigma; E3132) and 10 μM leupeptin hemisulfate). Cells were lysed by a single pass at 18,000 psi through a high-pressure microfluidizer (Microfluidics M110P), and the resulting lysate clarified at 30,000$g$ for 1 h at 4 °C. Clarified lysate was filtered through a 0.22 μm Millex Durapore low-protein-binding syringe filter (EMD Millipore) and applied to glutathione Sepharose 4b resin (Cytiva, 17075604) equilibrated with GST column buffer. After draining the flow through, the resin was washed with 50 column volumes GST column buffer. To elute the bound protein and cleave the GST tag in a single step, 50 U

HRV3C protease (Millipore, 71493) in 2.5 ml 20 mM Tris-HCl, pH 7.5, 50 mM NaCl, 1 mM DTT and 5% (v/v) glycerol was added to the column, and the column sealed and incubated for 3 h at 4 °C. Next, the column was drained to collect the cleaved protein. The eluate was diluted to 50 mM NaCl and further purified using a HiTrap Q (Cytiva, 29051325) anion-exchange column equilibrated with 20 mM Tris-HCl, pH 7.5, 50 mM NaCl, 1 mM DTT and 5% (v/v) glycerol. The bound protein was eluted using a 100–500 mM NaCl gradient in the same buffer. Peak fractions were analysed for purity by SDS–PAGE and the purest were pooled and dialysed into storage buffer containing 30 mM HEPES-KOH, pH 7.5, 100 mM potassium acetate, 3.5 mM magnesium acetate, 1 mM DTT and 20% (v/v) glycerol. Aliquots of the pooled fractions were flash-frozen in liquid nitrogen and stored at −80 °C.

### In vitro cleavage assays

Cleavage assays were conducted as previously described[6,15,48]. In brief, target RNA substrates (Extended Data Fig. 3) were ordered from IDT and labelled using [γ-³²P]ATP (Perkin Elmer) and polynucleotide kinase (NEB, M0201). Unincorporated [γ-³²P]ATP was removed using a G-25 spin column (Cytiva, 27532501), and target RNA was purified using a 15% denaturing polyacrylamide gel, eluted overnight with rotation in 0.4 M NaCl at 4 °C and collected by ethanol precipitation. Radiolabelled target (0.1 nM final concentration (f.c.)) was added to a mix of purified piRISC (0.5 nM f.c.) and GTSF1 (100 nM f.c.) to assemble a 30 μl cleavage reaction. At 0, 1, 5, 10, 30 and 60 min (0.5, 1, 2, 4 and 6 h for *Cox7a2l*), a 5 μl sample was quenched in 280 μl 50 mM Tris-HCl, pH 7.5, 100 mM NaCl, 25 mM EDTA and 1% (w/v) SDS, then proteinase K (1 mg ml⁻¹ f.c.) was added and the mix incubated at 45 °C for 15 min, followed by extraction with phenol–chloroform–isoamyl alcohol (25:24:1, pH 6.7) and ethanol precipitation. RNA was resuspended in 10 μl 95% (v/v) formamide, 5 mM EDTA, 0.025% (w/v) bromophenol blue and 0.025% (w/v) xylene cyanol, heated at 95 °C for 2 min and resolved in a 7% denaturing polyacrylamide gel. Gels were dried, exposed to a storage phosphor screen and imaged on a Typhoon FLA 7000 (GE). The raw image file was used to quantify the substrate and product bands, corrected for background.

Data were used to fit the burst-and-steady-state scheme

$$E + S \underset{k_{-1}}{\overset{k_1}{\rightleftharpoons}} ES \overset{k_2}{\rightarrow} EP \overset{k_3}{\rightarrow} E + P, \text{ using the equation:}$$

$$
\begin{aligned}
[P_{relative}] = f(t) \\
= [E_{relative}]\Big( [k_2/(k_2 + k_3)]^2 \times (1 - e^{-[k_2+k_3]t}) \\
+ [k_2 k_3/(k_2 + k_3)]t \Big)([E_{relative}]).
\end{aligned}
$$

### FACS isolation and immunostaining of mouse germ cells

Mouse germ cells were sorted as previously described[6,15,48]. In brief, testes of 2–7-month-old mice were isolated, decapsulated and incubated for 15 min at 33 °C in 1× Gey's balanced salt solution (GBSS, Sigma, G9779) containing 0.4 mg ml⁻¹ collagenase type 4 (Worthington, LS004188) rotating at 150 rpm. Seminiferous tubules were then washed twice with 1× GBSS and incubated for 15 min at 33 °C in 1× GBSS with 0.5 mg ml⁻¹ trypsin and 1 μg ml⁻¹ DNase I, rotating at 150 rpm. Next, tubules were homogenized by pipetting through a glass Pasteur pipette for 3 min at 4 °C. Fetal bovine serum (FBS; 7.5% f.c., v/v) was added to inactivate trypsin, and the cell suspension was then strained through a pre-wetted 70 μm cell strainer (ThermoFisher, 22363548). Cells were collected by centrifugation at 300$g$ for 10 min. The supernatant was removed, cells were resuspended in 1× GBSS containing 5% (v/v) FBS, 1 μg ml⁻¹ DNase I and 5 μg ml⁻¹ Hoechst 33342 (ThermoFisher, 62249) and rotated at 150 rpm for 45 min at 33 °C. Propidium iodide (0.2 μg ml⁻¹, f.c.; ThermoFisher, P3566) was added, and cells were strained through a pre-wetted 40 μm cell strainer (ThermoFisher, 22363547).

Spermatogonia, primary spermatocytes, secondary spermatocytes and round spermatids were purified using a BD FACSDiscover S8 Cell Sorter (Genomics Core at NYU Center for Genomics and Systems Biology) and a FACSAria II Cell Sorter (BD Biosciences; UMass Medical School FACS Core) as previously described[48,53]. In brief, the 355-nm laser was used to excite Hoechst 33342, whereas the 488-nm laser was used to record forward and side scatter and to excite propidium iodide. Propidium iodide emission was detected using a 610/20 bandpass filter. Hoechst 33342 emission was recorded using 450/50 and 670/50 band pass filters (Supplementary Fig. 8). Cells were collected by centrifugation at 900$g$ for 10 min. The supernatant was removed and the cell pellets were flash-frozen in liquid nitrogen and stored at −80 °C.

Germ cell stages in the unsorted population and the purity of sorted fractions were assessed by immunostaining aliquots of cells. Cells were incubated for 20 min in 25 mM sucrose and then fixed on a slide with 1% (w/v) PFA containing 0.15% (v/v) Triton X−100 for 2 h at room temperature in a humidifying chamber. Slides were washed sequentially for 10 min as follows: (1) PBS containing 0.4% (v/v) Photo-Flo 200 (Kodak, 1464510); (2) PBS containing 0.1% (v/v) Triton X-100; and (3) PBS containing 0.3% (w/v) BSA, 1% (v/v) donkey serum (Sigma, D9663) and 0.05% (v/v) Triton X-100. After washing, slides were incubated with primary antibodies in PBS containing 3% (w/v) BSA, 10% (v/v) donkey serum and 0.5% (v/v) Triton X-100 overnight at room temperature in a humidified chamber. Rabbit polyclonal anti-SYCP3 (Abcam, ab15093, RRID:AB_301639, 1:1,000 dilution) and mouse monoclonal anti-γH2AX (Millipore, 05-636, RRID:AB_309864, 1:1,000 dilution) were used as primary antibodies. Slides were washed again as described and then incubated with secondary donkey anti-mouse IgG (H+L) Alexa Fluor 594 (ThermoFisher, A-21203, RRID:AB_2535789, 1:2,000 dilution) or donkey anti-rabbit IgG (H+L) Alexa Fluor 488 (ThermoFisher, A-21206, RRID:AB_2535792, 1:2,000 dilution) for 1 h at room temperature in a humidified chamber. After incubation, slides were washed 3 times (10 min each) in PBS containing 0.4% (v/v) Photo-Flo 200 and once for 10 min in 0.4% (v/v) Photo-Flo 200. Finally, slides were dried and mounted in ProLong Gold Antifade mountant with DAPI (ThermoFisher, P36931). To assess the purity of sorted fractions, 50–100 cells were staged by DNA, γH2AX and SYCP3 staining[53]. All samples used here met the following criteria: spermatogonia, around 95–100% pure with ≤5% pre-leptotene spermatocytes; primary spermatocytes, about 10–15% leptotene/zygotene spermatocytes, around 45–50% pachytene spermatocytes and about 35–40% diplotene spermatocytes; secondary spermatocytes, around 100%; round spermatids, about 95–100%, with ≤5% elongated spermatids.

### Small RNA-seq library preparation

Total RNA from sorted mouse germ cells was extracted using a mirVana miRNA isolation kit (ThermoFisher, AM1560). Small RNA libraries were constructed as previously described[6,15,48] with modifications. In brief, before library preparation, an equimolar mix of nine synthetic spike-in RNA oligonucleotides (Supplementary Table 9) was added to each RNA sample to enable absolute quantification of small RNAs (Supplementary Table 10). The median volume of primary spermatocytes (1,800 μm³) from ref. 16 was used to calculate the intracellular concentration: 1 molecule per primary spermatocyte corresponds to around 1 pM. To reduce ligation bias and to eliminate PCR duplicates, the 3′ and 5′ adaptors both contained nine random nucleotides at their 5′ and 3′ ends, respectively[54] (Supplementary Table 9) and 3′ adaptor ligation reactions contained 25% (w/v) PEG-8000 (f.c.). In brief, 500–1,000 ng total RNA was first ligated to 25 pmol of 3′ DNA adapter (Supplementary Table 9) with adenylated 5′ and dideoxycytosine-blocked 3′ ends in 30 μl of 50 mM Tris-HCl (pH 7.5), 10 mM MgCl₂, 10 mM DTT and 25% (w/v) PEG-8000 (NEB) with 600 U of homemade T4 Rnl2tr K227Q at 16 °C overnight. After ethanol precipitation, the 50–90 nucleotide (14–54 nucleotide small RNA + 36 nucleotide 3′ UMI adapter) 3′ ligated product was purified from a 15% denaturing urea–polyacrylamide

gel (National Diagnostics). After overnight elution in 0.4 M NaCl followed by ethanol precipitation, the 3′ ligated product was denatured in 14 μl water at 90 °C for 60 s, 1 μl of 50 μM RT primer (Supplementary Table 9) was added and annealed at 65 °C for 5 min to suppress the formation of 5′-adapter–3′-adapter dimers during the next step. The resulting mix was then ligated to a mixed pool of equimolar amount of two 5′ RNA adapters (to increase nucleotide diversity at the 5′ end of the sequencing read; Supplementary Table 9) in 20 μl of 50 mM Tris-HCl (pH 7.8), 10 mM MgCl₂, 10 mM DTT and 1 mM ATP with 20 U of T4 RNA ligase (ThermoFisher, EL0021) at 25 °C for 2 h. The ligated product was precipitated with ethanol, cDNA synthesis was performed in 20 μl at 42 °C for 1 h using AMV reverse transcriptase (NEB, M0277), and 5 μl of the RT reaction was amplified in 25 μl using AccuPrime *Pfx* DNA polymerase (ThermoFisher, 12344024; 95 °C for 2 min, 15 cycles of 95 °C for 15 s, 65 °C for 30 s and 68 °C for 15 s; primers are listed in Supplementary Table 9). Finally, the PCR product was purified in a 2% agarose gel. Small RNA-seq libraries samples were sequenced using a NextSeq 550 (Illumina) to obtain 79-nucleotide, single-end reads.

### RNA-seq library preparation

Total RNA from sorted germ cells was extracted using a mirVana miRNA isolation kit (ThermoFisher, AM1560). RNA-seq of rRNA-depleted total RNAs was performed as previously described[6,15,48,55] with modifications, including the addition of the ERCC spike-in mix to enable absolute quantification of RNAs and the use of unique molecular identifiers in adapters (Supplementary Table 9) to eliminate PCR duplicates[54]. In brief, before library preparation, 1 μl of 1:100 diluted ERCC spike-in mix 1 (ThermoFisher, 4456740) was added to 1 μg total RNA. To remove rRNA, 1 μg total RNA was hybridized in 10 μl to a pool of 186 rRNA antisense oligos (0.05 μM f.c. each) in 10 mM Tris-HCl (pH 7.4), 20 mM NaCl by heating the mixture to 95 °C, cooling at −0.1 °C s⁻¹ to 22 °C, and incubating at 22 °C for 5 min. RNase H (10 U; Lucigen, H39500) was added and the mixture incubated at 45 °C for 30 min in 20 μl containing 50 mM Tris-HCl (pH 7.4), 100 mM NaCl, 20 mM MgCl₂. The reaction volume was adjusted to 50 μl with 1× TURBO DNase buffer (ThermoFisher, AM2238) and then incubated with 4 U TURBO DNase (ThermoFisher, AM2238) for 20 min at 37 °C. Next, RNA was purified using RNA Clean & Concentrator-5 (Zymo Research, R1016) to retain ≥200 nucleotide RNAs, followed by the stranded, dUTP-based RNA-seq protocol described in ref. 55. RNA-seq libraries were sequenced using a NextSeq 550 (Illumina) to obtain 79 + 79-nucleotide, paired-end reads. The median number of all non-rRNA transcripts was around 3,400,000 in primary spermatocytes, about 1,700,000 in secondary spermatocytes, about 770,000 in round spermatids and around 50,000 in elongating spermatids. The median volume of primary spermatocytes (1,800 μm³) from a previous study[16] was used to calculate intracellular concentration: 1 molecule per primary spermatocyte corresponds to around 1 pM.

For sequencing of polyadenylated RNAs, NEBNext Poly(A) mRNA Magnetic Isolation Module (NEB, E7490S) was used to purify poly(A)+ transcripts from 1–2 μg total RNA according to manufacturer's instructions. Poly(A)+ RNAs were used to prepare RNA-seq libraries with NEBNext UltraExpress RNA Library Prep Kit (E3330S) except that UMI-containing adaptors (Supplementary Table 9) were used. RNA-seq libraries were sequenced using an AVITI benchtop sequencer (Element Biosciences) to obtain 150 + 150-nucleotide, paired-end reads.

### Sequencing of 5′ monophosphorylated long RNAs

Total RNA from FACS-purified primary spermatocytes was extracted using mirVana miRNA isolation kit (ThermoFisher, AM1560) and used to prepare a library of 5′ monophosphorylated long RNAs as previously described[6,15,16,48,56] with modifications. Briefly, rRNA was depleted as described above for RNA-seq libraries. RNA was ligated to a mixed pool of equimolar amount of two 5′ RNA adapters (to increase nucleotide diversity at the 5′ end of the sequencing read, Supplementary Table 9)

in 20 μl of 50 mM Tris-HCl (pH 7.8), 10 mM MgCl$_2$, 10 mM DTT, 1 mM ATP with 60 U of High Concentration T4 RNA ligase (NEB, M0437M) at 16 °C overnight. The ligated product was isolated using RNA Clean & Concentrator-5 (Zymo Research, R1016) to retain ≥200 nucleotide RNAs and reverse transcribed in 25 μl with 50 pmol RT primer (Supplementary Table 9) using SuperScript III (ThermoFisher, 18080093). After purification with 50 μl Ampure XP beads (Beckman Coulter, A63880), cDNA was PCR amplified using NEBNext High-Fidelity (NEB, M0541; 98 °C for 30 s; 4 cycles of: 98 °C for 10 s, 59 °C for 30 s, 72 °C for 12 s; 6 cycles of: 98 °C for 10 s, 68 °C for 10 s, 72 °C for 12 s; 72 °C for 3 min; primers listed in Supplementary Table 9). PCR products between 200–400 bp were isolated from a 1% agarose gel, purified with QIAquick Gel Extraction Kit (Qiagen, 28706), and amplified again with NEBNext High-Fidelity (NEB, M0541; 98 °C for 30 s; 3 cycles of: 98 °C for 10 s, 68 °C for 30 s, 72 °C for 14 s; 6 cycles of: 98 °C for 10 s, 72 °C for 14 s; 72 °C for 3 min; primers listed in Supplementary Table 9). The PCR product was purified from a 1% agarose gel and sequenced using a NextSeq 550 or NovaSeq (Illumina) to obtain 79 + 79-nucleotide or 150 + 150-nucleotide, paired-end reads.

## Sequencing of ribosome footprints

Ribosome footprint profiling was performed as described previously[57]. All steps were performed on ice, unless otherwise indicated. FACS-purified primary spermatocytes, secondary spermatocytes, or round spermatids (1–2 million cells) were lysed in 0.5 ml of 10 mM Tris-HCl (pH 7.5), 100 mM KCl, 5 mM MgCl$_2$, 2 mM DTT, 1% (v/v) Triton X-100, 100 μg ml$^{-1}$ cycloheximide (Sigma, C4859), and 1× protease inhibitor cocktail (1 mM 4-(2-Aminoethyl)benzenesulfonyl fluoride hydrochloride [Sigma; A8456], 0.3 μM Aprotinin, 40 μM betanin hydrochloride, 10 μM E-64 (Sigma; E3132), 10 μM leupeptin hemisulfate). Cell debris were removed by centrifugation at 20,000g for 10 min at 4 °C. RNase I (Ambion, AM2294) was added to the supernatant (0.2 U μl$^{-1}$ f.c.) and the sample was incubated at 25 °C for 30 min and then moved to a polycarbonate ultracentrifuge tube (Beckman Coulter, 362305). A 3 ml sucrose cushion (10 mM Tris-HCl pH 7.5, 100 mM KCl, 5 mM MgCl$_2$, 2 mM DTT, 100 μg ml$^{-1}$ cycloheximide (Sigma, C4859), 20 U ml$^{-1}$ SUPERaseIn RNase Inhibitor (Fisher Scientific, AM2694) in 1 M sucrose) was placed under the sample using a 21 G needle (BD, 305167) on a 5 ml syringe (Fisher Scientific, 14955458). Ribosomes were precipitated by centrifugation at about 400,000g for 90 min at 4 °C (100,000 rpm in TLA-110 rotor in Optima MAX-XP Benchtop Ultracentrifuge). RNA was extracted from the ribosome pellet using mirVana miRNA isolation kit (ThermoFisher, AM1560). After ethanol precipitation, the 27–33-nucleotide ribosome footprints were purified from a 15% denaturing urea-polyacrylamide gel (National Diagnostics). After overnight elution in 0.4 M NaCl followed by ethanol precipitation, the 3′ ends of ribosome footprints we dephosphorylated at 37 °C for 4 h in 50 μl of 100 mM MES-NaOH (pH 5.5), 300 mM NaCl, 10 mM MgCl$_2$, 1 U μl$^{-1}$ SUPERaseIn RNase Inhibitor (Fisher Scientific, AM2694), 15 mM 2-mercaptoethanol, 0.8 U μl$^{-1}$ T4 PNK (NEB, M0201). After ethanol precipitation, ribosome footprints were ligated to 25 pmol of 3′ DNA adapter for small RNA sequencing (Supplementary Table 9) with adenylated 5′ and dideoxycytosine-blocked 3′ ends in 30 μl of 50 mM Tris-HCl (pH 7.5), 10 mM MgCl$_2$, 10 mM DTT, and 25% (w/v) PEG-8000 (NEB) with 600 U of homemade T4 Rnl2tr K227Q at 16 °C overnight. After ethanol precipitation, the 5′ ends of 63–69 nucleotide 3′ ligated product (27–33-nucleotide footprints plus 36-nucleotide 3′ unique molecular identifier (UMI) adapter) were phosphorylated in 20 μl of 70 mM Tris-HCl (pH 7.6), 10 mM MgCl$_2$, 5 mM DTT, 1 mM ATP with 20 U of T4 PNK (NEB, M0201). Following an ethanol precipitation, RNAs were denatured in 14 μl water at 90 °C for 60 s, 1 μl of 50 μM RT primer (Supplementary Table 9) was added and annealed at 65 °C for 5 min to suppress the formation of 5′-adapter:3′-adapter dimers during the next step. The resulting mix was then ligated to a mixed pool of equimolar amount of two 5′ small RNA-seq adapters (to increase nucleotide

diversity at the 5′ end of the sequencing read, Supplementary Table 9) in 20 μl of 50 mM Tris-HCl (pH 7.8), 10 mM MgCl$_2$, 10 mM DTT, 1 mM ATP with 20 U of T4 RNA ligase (ThermoFisher, EL0021) at 25 °C for 2 h. The ligated product was precipitated with ethanol, cDNA synthesis was performed in 20 μl at 42 °C for 1 h using AMV reverse transcriptase (NEB, M0277), and 5 μl of the RT reaction was amplified in 25 μl using AccuPrime Pfx DNA polymerase (ThermoFisher, 12344024; 95 °C for 2 min, 16 cycles of: 95 °C for 15 s, 65 °C for 30 s, 68 °C for 15 s; primers listed in Supplementary Table 9). Finally, the PCR product was purified in a 2% agarose gel. Ribosome footprint libraries were sequenced using a NextSeq 550 (Illumina) to obtain 79-nucleotide, single-end reads.

## GRO-seq

All steps were performed on ice, unless otherwise indicated. FACS-purified primary spermatocytes (1–3 million cells) were collected by centrifugation at 400g for 10 min at 4 °C. Supernatant was removed and cells were carefully resuspended by pipetting in 1 ml swelling buffer (10 mM Tris-HCl (pH 7.5), 3 mM CaCl$_2$, 2 mM MgCl$_2$). Additional 9 ml of swelling buffer was added, then the cells were mixed by swirling and incubated on ice for 5 min. After collecting swollen cells by centrifugation at 400g for 10 min at 4 °C, supernatant was removed and cells were resuspended in 500 μl of Lysis buffer: 10 mM Tris-HCl (pH 7.5), 3 mM CaCl$_2$, 2 mM MgCl$_2$, 10% glycerol, 0.04 U μl$^{-1}$ RNasin PLUS (Promega, N2615), and 1× protease inhibitor cocktail (1 mM 4-(2-Aminoethyl)benzenesulfonyl fluoride hydrochloride (Sigma; A8456), 0.3 μM Aprotinin, 40 μM betanin hydrochloride, 10 μM E-64 (Sigma; E3132), 10 μM leupeptin hemisulfate). While carefully swirling the tube, 500 μl of Lysis buffer containing 1% Igepal CA-630 was added by drop and cells were lysed for 5 min on ice. Additional 9 ml of lysis buffer containing 0.5% Igepal CA-630 was added, lysate was mixed by swirling, nuclei were collected by centrifugation at 600g for 5 min at 4 °C, supernatant was removed, and nuclei were resuspended by pipetting in 1 ml of Lysis buffer containing 0.5% Igepal CA-630. Additional 9 ml of Lysis buffer containing 0.5% Igepal CA-630 was added, nuclei were mixed by swirling and collected by centrifugation at 600 × g for 5 min at 4 °C, supernatant was removed, and nuclei were resuspended in 1 ml of Freezing buffer (50 mM Tris-HCl pH 8.0, 5 mM MgCl$_2$, 5 mM EDTA, 40% glycerol, 0.4 U μl$^{-1}$ RNasin PLUS (Promega, N2615), 1× protease inhibitor cocktail). Nuclei were collected by centrifugation at 900g for 5 min at 4 °C, supernatant was removed, and nuclei were resuspended in 100 μl of Freezing buffer, flash frozen in liquid nitrogen, and stored at −80 °C.

For nuclear run-on reaction, 100 μl of frozen nuclei was thawed on ice for 5 min and then mixed with 100 μl of 10 mM Tris-HCl pH 8.0, 300 mM KCl, 5 mM MgCl$_2$, 10 mM DTT, 0.5 mM of each ATP, GTP, CTP, and BrdUTP (Sigma, B7166), 1% N-Lauroylsarcosine (Sigma L7414), 1 U μl$^{-1}$ RNasin PLUS (Promega, N2615), and 1× protease inhibitor cocktail. Reaction was mixed with P200 tip with its end cut off and incubated at 30 °C for 30 min, then 24 μl of 10× TURBO DNase buffer and 10 μl of TURBO DNase (2 U μl$^{-1}$, Fisher Scientific, AM2238) were added, reaction was incubated at 37 °C for 20 min and RNA was extracted with Trizol, resuspended in 30 μl water and stored at −80 °C.

To capture BrdU-labelled nascent transcripts, 30 μl of the sample from previous step was incubated at 65 °C for 5 min, chilled on ice, and mixed with 270 μl of IP buffer: 50 mM Tris-HCl pH 8.0, 150 mM NaCl, 1 mM DTT, 1 mM EDTA, 0.05% Tween-20, 1 U μl$^{-1}$ RNasin PLUS (Promega, N2615), and 1× protease inhibitor cocktail. Anti-BrdU mouse biotin-conjugated antibody (1 μg, 5 μl of 0.2 μg μl$^{-1}$ of Clone PRB-1, MilliporeSigma, MAB3262BMI) was added to the RNAs in IP buffer and RNAs were incubated at 4 °C for 1 h with rotation. In a separate tube, 50 μl of Dynabeads MyOne Streptavidin T1 (Fisher Scientific, 6560) were washed at room temperature for 5 min in 1 ml of IP buffer, and beads were then blocked at room temperature for 1 h with rotation in 300 μl of IP buffer containing 0.1% polyvinylpyrrolidone and 1 mg ml$^{-1}$ Ultrapure BSA (Fisher Scientific, AM2618). After blocking, supernatant was removed, and beads were resuspended in solution containing

RNAs and antibody from previous step. Biotin-conjugated antibody was allowed to bind streptavidin beads at 4 °C for 30 min with rotation. Beads were then washed five times with IP buffer at 4 °C for 5 min with rotation, and RNAs were extracted with Trizol. GRO-seq libraries were constructed using the method described for rRNA-depleted RNA-seq libraries and sequenced with a NextSeq 550 (Illumina) to obtain 79 + 79-nucleotide, paired-end reads.

### Analysis of small RNA sequencing data

Smal RNA data were analysed as previously described[6,15,48]. Briefly, the 3′ adapter (5′-TGGAATTCTCGGGTGCCAAGG-3′) was removed with fastx toolkit (v.0.0.14), PCR duplicates were eliminated as described[54], and rRNA matching reads were removed with bowtie (parameter -v 1; v.1.0.0; ref. 58) against Mus musculus set in SILVA rRNA database[59]. Deduplicated and filtered data were analysed with Tailor (v.1.1; ref. 60) to account for non-templated tailing of small RNAs. Sequences of synthetic spike-in oligonucleotides (Supplementary Table 9) were identified allowing no mismatches with bowtie (parameter -v 0; v1.0.0; ref. 58), and the absolute abundance of small RNAs calculated (Supplementary Table 10). Because piRNA 3′ trimming by PNLDC1 results in heterogeneous 3′ ends, sequencing reads were next grouped by their 5′, 25-nucleotide prefix. For further analyses, we kept only prefix groups that met two criteria. First, the prefix group total abundance was ≥1 ppm, that is, ≥10 piRNAs per mouse primary spermatocyte. Assuming a Poisson or a Negative Binomial distribution for piRNA concentration in different cells, this threshold ensures that ≥99.99% of primary spermatocytes contained at least one molecule of the piRNA 25 nucleotide prefix. Second, total abundance of the prefix group was required to be ≥1 ppm in all 12 replicates of the C57BL/6 control samples (Supplementary Table 2). piRNAs were considered undetectable in $pi6^{-/-}$, $pi9^{-/-}$, $pi17^{-/-}$, $pi9^{-/-}pi17^{-/-}$ or $pi2^{-/-}pi9^{-/-}pi17^{-/-}$ primary spermatocytes if their mean abundance in mutants was ≤0.1 ppm.

### Analysis of RNA-seq data

RNA-seq data were analysed as previously described[6,15,48]. Briefly, analysis was performed using piPipes for genomic alignment (v.1.5.0; ref. 61). Briefly, before starting piPipes, sequences were reformatted to extract UMIs[54]. The reformatted reads were then aligned to rRNA using bowtie2 (v.2.2.0)[62]. Unaligned reads were mapped to mouse genome mm10 using STAR (v.2.3.1)[63], and PCR were duplicates removed[54]. Transcript abundance was calculated with StringTie (v1.3.4)[64] using mm10/rmsk and gene annotation from Ensembl. Differential expression analysis was performed using DESeq2 (v.1.18.1)[65]. To exclude Cas9-induced off-target changes, only significant (FDR < 0.01) gene expression changes observed in both alleles (em1 and em2) for mice with a $pi6^{-/-}$, $pi9^{-/-}$ or $pi17^{-/-}$ single mutation and for mice with the $pi9^{-/-}pi17^{-/-}$ mutation were considered (Supplementary Table 3). Thus, all changes in transcript abundance reported for $pi9^{-/-}$, $pi17^{-/-}$ and $pi9^{-/-}pi17^{-/-}$ mice are the absolute minimums of the two alleles. This approach was not possible for $pi2^{-/-}$ and $pi2^{-/-}pi9^{-/-}pi17^{-/-}$ mice because only one allele of $pi2^{-/-}$ was generated. We considered only transcripts for which abundance was ≥3 TPM (around 10 molecules per primary spermatocytes[16]), which ensured that, assuming a Poisson or a negative binomial distribution for transcript concentration in different cells, ≥99.99% of primary spermatocytes contained at least 1 molecule of transcript.

### Analysis of 5′-monophosphorylated long RNA-seq data

The 5′-monophosphate RNA-seq data were analysed as previously described[6,15,48]. In brief, data for 5′-monophosphorylated long RNAs was aligned to the mouse genome with piPipes[61]. In brief, before starting piPipes, the degenerate portion of the 5′ adapter sequences were removed the (nucleotides 1–15 of read1). Because each library was sequenced at least twice to increase the sequencing depth, to harmonize the length of paired-end reads from different runs, sequences were trimmed to 64 nucleotides (read1) + 79 nucleotide (read2) paired reads.

The trimmed reads were then aligned to rRNA using bowtie2 (v.2.2.0)[62]. Unaligned reads were mapped to mouse genome mm10 using STAR (v.2.3.1)[63], alignments with soft clipping of ends were removed with SAMtools (v.1.0.0)[66], and reads with the same 5′ end were merged to represent a single 5′-monophosphorylated RNA species. For further analyses, only unambiguously mapping 5′-monophosphorylated RNA species were used. For 5′-monophosphorylated RNAs mapped in annotated transcripts, the nucleotide sequence of the corresponding transcript was used to find piRNAs potentially explaining the cleavage, and we used the genomic sequence for 5′-monophosphorylated RNAs mapped outside any annotated transcript. Searches for putative cleavage targets of $pi9$ and $pi17$ piRNAs in $pi9^{-/-}pi17^{-/-}$ primary spermatocytes (Fig. 3 and Supplementary Table 6) were performed with a threshold of ≥0.1 ppm for 5′-monophosphorylated putative cleavage products and the following piRNA–target pairing patterns were considered:

- for piRNAs at ≥1 ppm (10 molecules per primary spermatocyte), ≥20 nucleotides paired between g2 and g25;
- for piRNAs at ≥5 ppm (50 molecules per primary spermatocyte), contiguous pairing between g3 and g15;
- for piRNAs at ≥10 ppm (100 molecules per primary spermatocyte), contiguous pairing between g3 and g16;
- and for piRNAs at ≥50 ppm (500 molecules per primary spermatocyte), contiguous pairing between g4 and g17.

### Analysis of ribosome footprint sequencing data

The 3′ adapter (5′-TGGAATTCTCGGGTGCCAAGG-3′) was removed with fastx toolkit (v.0.0.14), PCR duplicates were eliminated as previously described[54] and rRNA matching reads were removed with bowtie (parameter -v 1; v.1.0.0)[58] against Mus musculus set in the SILVA rRNA database[59]. Unaligned reads were mapped to mouse genome mm10 using STAR (v.2.3.1)[63]. Ribosome occupancy was calculated using String-Tie (v.1.3.4)[64]. Differential expression analysis was performed using DESeq2 (v.1.18.1)[65]. Data from two biological replicates for each $pi9^{em1/em1}pi17^{em1/em1}$ and $pi9^{em2/em2}pi17^{em2/em2}$ were obtained and compared against the data from four biological replicates of C57BL/6 controls. Only significant (FDR < 0.01) changes in gene expression observed in both alleles for each pachytene piRNA mutation were considered (Supplementary Table 7). Identification of mRNAs with ELAVL1-binding motif was as previously described[22]. Data are presented in MS Excel 2013.

### Analysis of GRO-seq data

GRO-seq analysis was performed using piPipes for genomic alignment[61]. In brief, before starting piPipes, sequences were reformatted to extract UMIs[54]. The reformatted reads were then aligned to rRNA using bowtie2 (v.2.2.0)[62]. Unaligned reads were mapped to mouse genome mm10 using STAR (v.2.3.1)[63] and PCR duplicates were removed[54]. RNA PolII density was calculated using BEDTools genomecov (v.2.3.4)[67–71] as read coverage normalized by sequencing depth and gene length (parts per million per kb; ref. 64). To minimize any contribution from paused RNA PolII, the first 500 bp of genes were excluded from analyses.

### Reporting summary

Further information on research design is available in the Nature Portfolio Reporting Summary linked to this article.

## Data availability

Sequencing data are available from the National Center for Biotechnology Information Small Read Archive using accession number PRJNA1176701. Source data are provided with this paper.

## Code availability

The code used to analyse sequencing data was deposited into and can be accessed at GitHub (https://github.com/ildargv/Cecchini_et_al).

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

**Acknowledgements** We thank staff at the UMass FACS Core and NYU Center for Genomics and System Biology Genomics Core for help sorting mouse germ cells; members of the Zamore and Gainetdinov laboratories for discussions and critical comments on the manuscript; an anonymous reviewer for their suggestion of the term 'piRNA addiction'; the Zegar Family Foundation for their generous support; and staff at the NYU Center for Genomics and System Biology Genomics Core for their assistance and resources. This work was supported in part by NIGMS R35 GM136275 grant to P.D.Z., S10RR027897 to the UMass Electron Microscopy Facility, and 1S10 OD028576 to the UMass Flow Cytometry Core Facility. P.D.Z. is an investigator of the Howard Hughes Medical Institute.

**Author contributions** K.C., P.D.Z. and I.G. conceived the study. K.C. performed mouse fertility, sperm motility, testis and sperm morphology assays. M.Z., A.B. and N.A. performed and prepared sequencing libraries. J.V.-B. and S.B. performed in vitro cleavage experiments. I.G. performed sequencing data analyses. P.D.Z. and I.G. supervised the study. K.C., P.D.Z. and I.G. wrote the paper.

**Competing interests** The authors declare no competing interests.

**Additional information**
**Correspondence and requests for materials** should be addressed to Phillip D. Zamore or Ildar Gainetdinov.

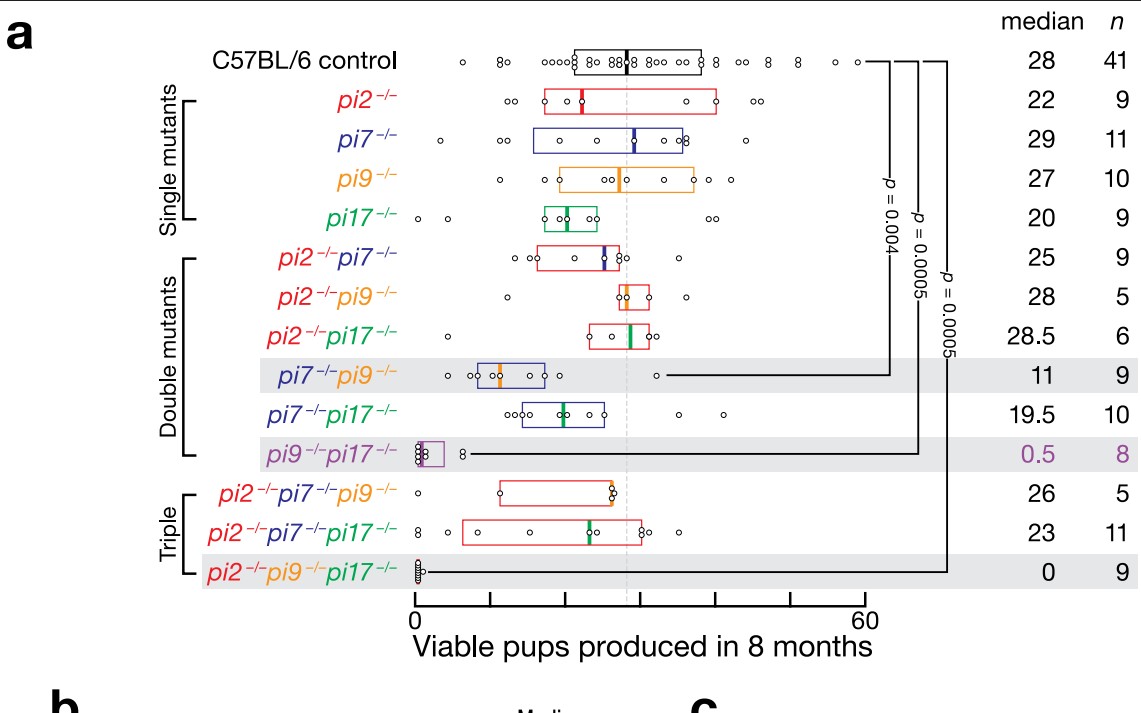

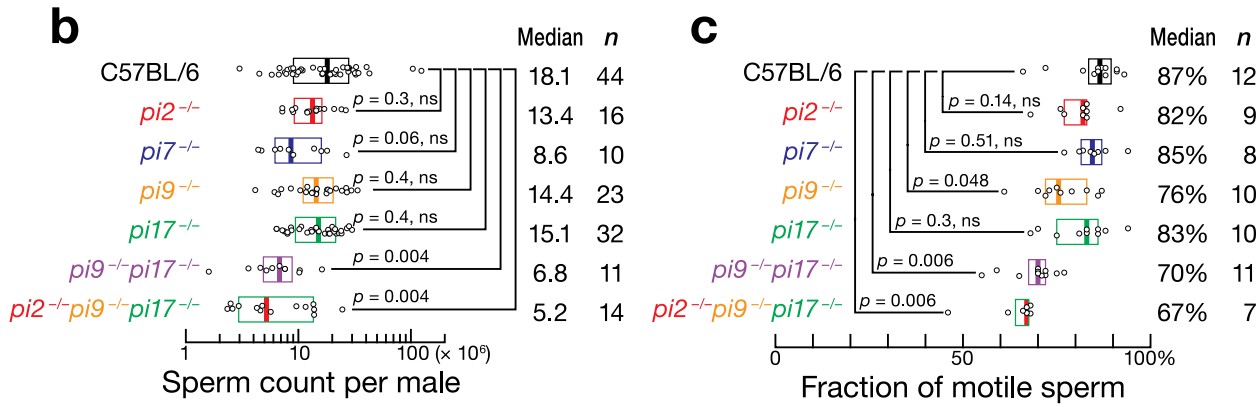

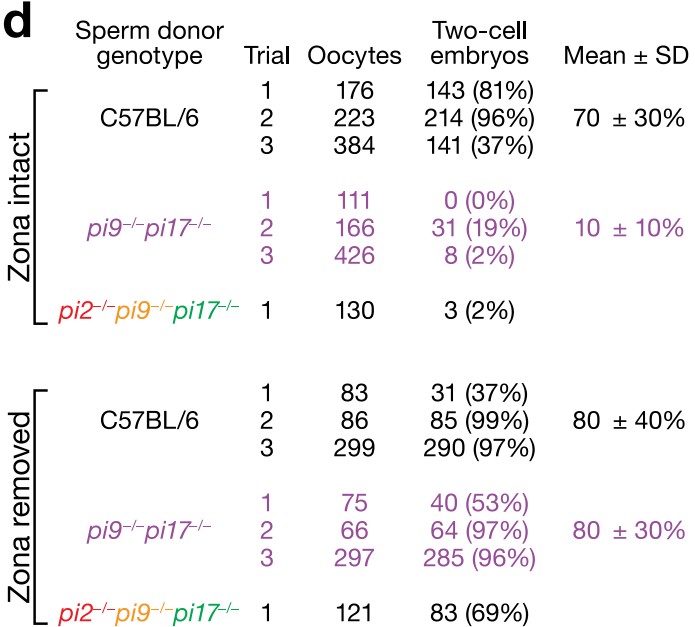

**Extended Data Fig. 1 | Fertility defects observed in double- and triple-mutant mice lacking pachytene piRNA loci. a**, Number of viable pups produced by the control (C57BL/6) and pachytene piRNA mutant males in successive matings with C57BL/6 females over 8 months. Median and IQR are shown. Kruskal-Wallis test (one-way ANOVA on ranks) $p$-value = $2 \times 10^{-7}$. Benjamini-Hochberg corrected $p$-values for *post hoc* pairwise two-tailed Mann-Whitney tests are shown. **b**, Total number of sperm in caudal epididymis of C57BL/6, $pi2^{-/-}$, $pi7^{-/-}$, $pi9^{-/-}$, $pi17^{-/-}$, and $pi9^{-/-}pi17^{-/-}$ males. Median and IQR are shown. Kruskal-Wallis test (one-way ANOVA on ranks) $p$-value = 0.00087. Benjamini-Hochberg corrected $p$-values for *post hoc* pairwise two-tailed Mann-Whitney tests are shown. **c**, Fraction of motile sperm from caudal epididymis of C57BL/6, $pi9^{-/-}$, $pi17^{-/-}$, and $pi9^{-/-}pi17^{-/-}$ males determined by CASAnova. Median and IQR are shown. Kruskal-Wallis test (one-way ANOVA on ranks) $p$-value = 0.00036. Benjamini-Hochberg corrected $p$-values for *post hoc* pairwise two-tailed Mann-Whitney tests are shown. **d**, C57BL/6 and $pi9^{-/-}pi17^{-/-}$ sperm function analysed by IVF using C57BL/6 oocytes with or without zona pellucida. Data are mean and SD.

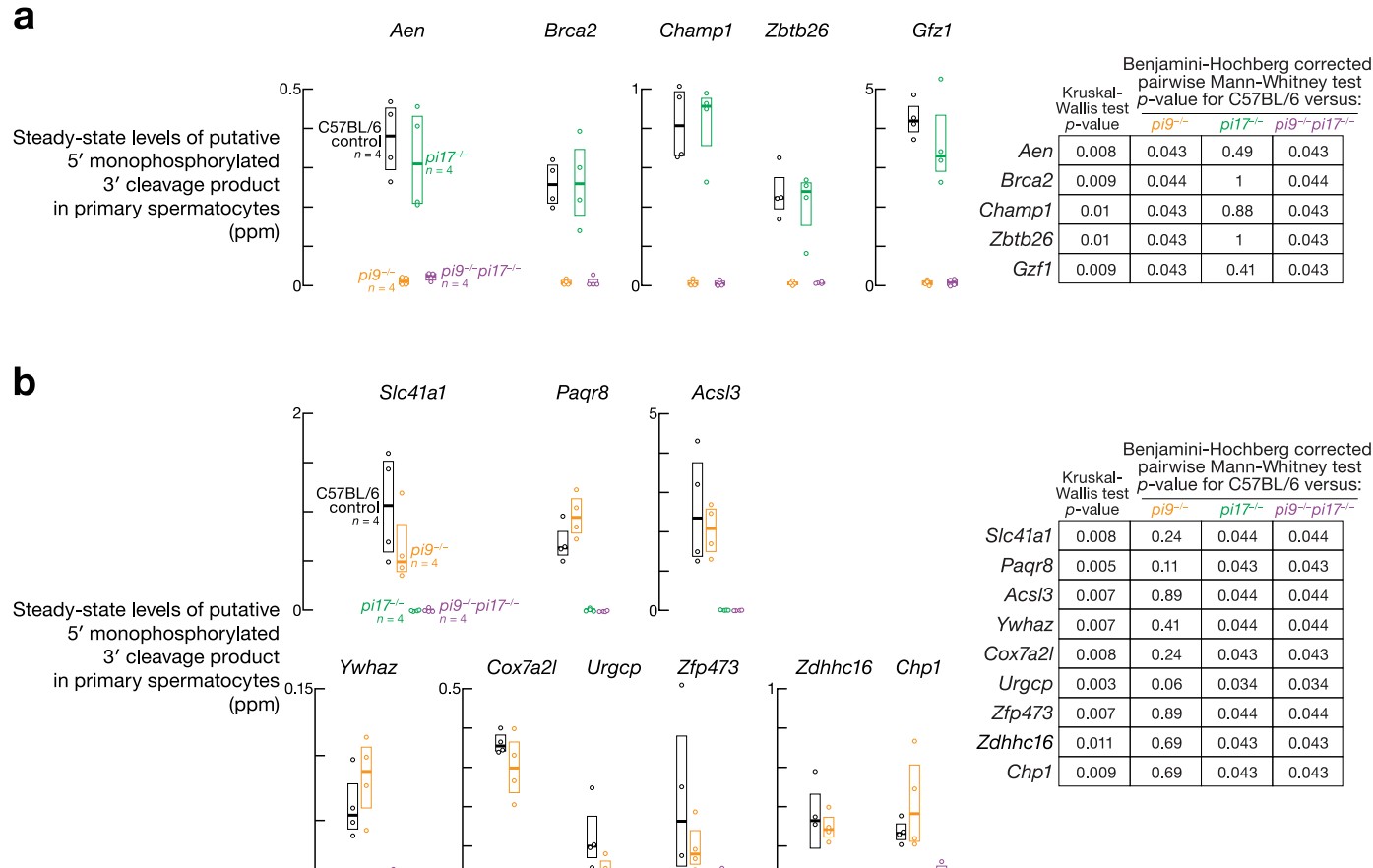

**a**

| | Kruskal-Wallis test *p*-value | Benjamini–Hochberg corrected, pairwise Mann-Whitney test *p*-value for C57BL/6 versus: | | |
|---|---|---|---|---|
| | | *pi9⁻/⁻* | *pi17⁻/⁻* | *pi9⁻/⁻pi17⁻/⁻* |
| *Aen* | 0.008 | 0.043 | 0.49 | 0.043 |
| *Brca2* | 0.009 | 0.044 | 1 | 0.044 |
| *Champ1* | 0.01 | 0.043 | 0.88 | 0.043 |
| *Zbtb26* | 0.01 | 0.043 | 1 | 0.043 |
| *Gzf1* | 0.009 | 0.043 | 0.41 | 0.043 |

**b**

| | Kruskal-Wallis test *p*-value | Benjamini–Hochberg corrected, pairwise Mann-Whitney test *p*-value for C57BL/6 versus: | | |
|---|---|---|---|---|
| | | *pi9⁻/⁻* | *pi17⁻/⁻* | *pi9⁻/⁻pi17⁻/⁻* |
| *Slc41a1* | 0.008 | 0.24 | 0.044 | 0.044 |
| *Paqr8* | 0.005 | 0.11 | 0.043 | 0.043 |
| *Acsl3* | 0.007 | 0.89 | 0.044 | 0.044 |
| *Ywhaz* | 0.007 | 0.41 | 0.044 | 0.044 |
| *Cox7a2l* | 0.008 | 0.24 | 0.043 | 0.043 |
| *Urgcp* | 0.003 | 0.06 | 0.034 | 0.034 |
| *Zfp473* | 0.007 | 0.89 | 0.044 | 0.044 |
| *Zdhhc16* | 0.011 | 0.69 | 0.043 | 0.043 |
| *Chp1* | 0.009 | 0.69 | 0.043 | 0.043 |

**Extended Data Fig. 2 | Steady-state levels of putative cleavage products generated by *pi9* and *pi17* piRNA-guided slicing. a**, At left, abundance in C57BL/6 (*n* = 4), *pi9⁻/⁻* (*n* = 4), *pi17⁻/⁻* (*n* = 4), and *pi9⁻/⁻pi17⁻/⁻* (*n* = 4) primary spermatocytes of putative 5′-monophosphate-bearing cleavage products derived from targets of *pi9* piRNAs (shown in Fig. 2a). Boxplots show median and interquartile range. At right, Kruskal-Wallis test *p*-values (one-way ANOVA on ranks) and Benjamini-Hochberg-corrected *p*-values for *post hoc* pairwise two-tailed Mann-Whitney tests are shown for each gene. **b**, At left, abundance in C57BL/6 (*n* = 4), *pi9⁻/⁻* (*n* = 4), *pi17⁻/⁻* (*n* = 4), and *pi9⁻/⁻pi17⁻/⁻* (*n* = 4) primary spermatocytes of putative 5′-monophosphate-bearing cleavage products derived from targets of *pi17* piRNAs (shown in Fig. 2a). Boxplots show median and interquartile range. At right, Kruskal-Wallis test *p*-values (one-way ANOVA on ranks) and Benjamini-Hochberg-corrected *p*-values for *post hoc* pairwise two-tailed Mann-Whitney tests are shown for each gene.

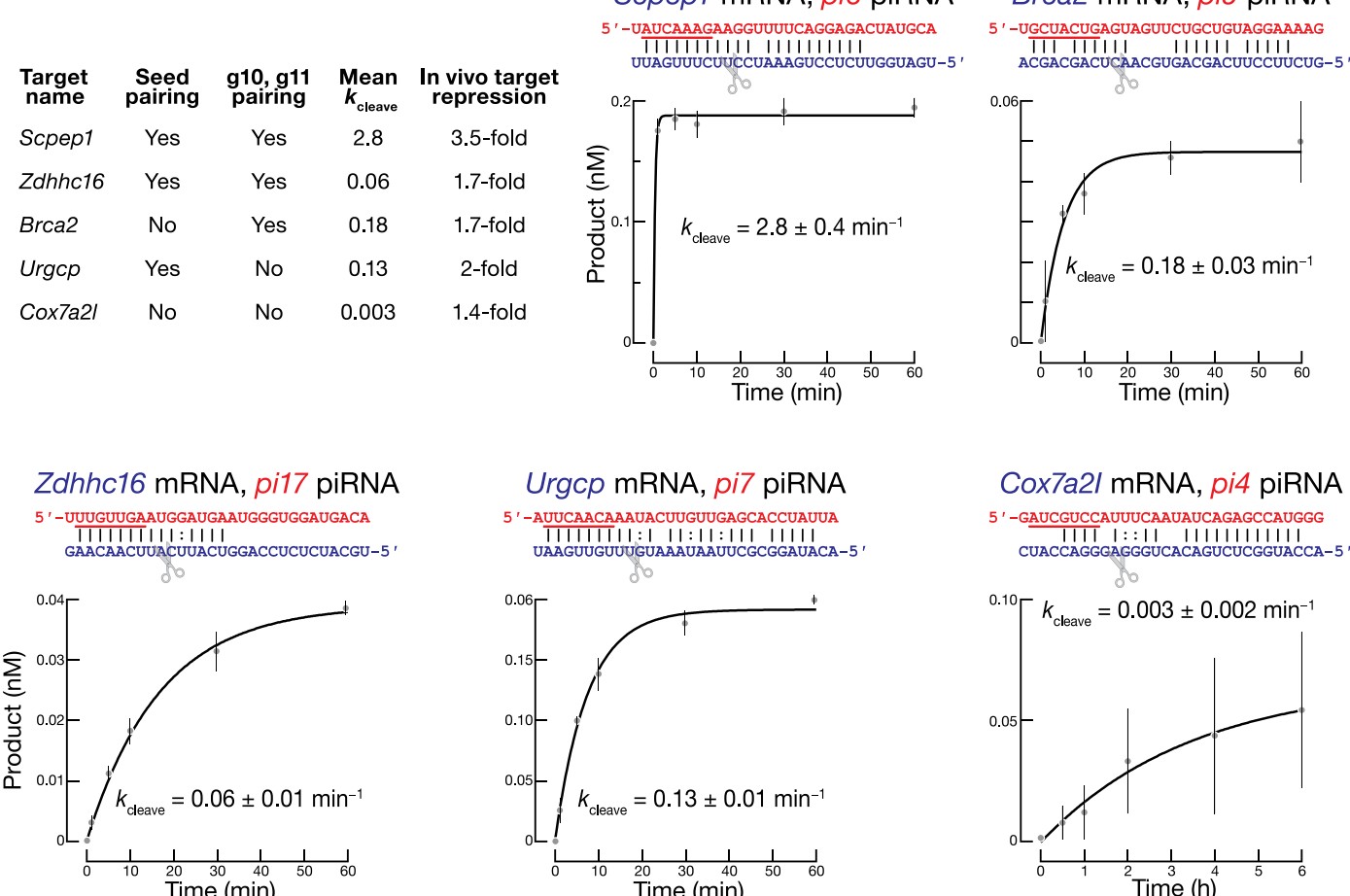

| Target name | Seed pairing | g10, g11 pairing | Mean $k_{cleave}$ | In vivo target repression |
|---|---|---|---|---|
| *Scpep1* | Yes | Yes | 2.8 | 3.5-fold |
| *Zdhhc16* | Yes | Yes | 0.06 | 1.7-fold |
| *Brca2* | No | Yes | 0.18 | 1.7-fold |
| *Urgcp* | Yes | No | 0.13 | 2-fold |
| *Cox7a2l* | No | No | 0.003 | 1.4-fold |

**Scpep1 mRNA, pi6 piRNA**

$k_{cleave} = 2.8 \pm 0.4$ min$^{-1}$

**Brca2 mRNA, pi9 piRNA**

$k_{cleave} = 0.18 \pm 0.03$ min$^{-1}$

**Zdhhc16 mRNA, pi17 piRNA**

$k_{cleave} = 0.06 \pm 0.01$ min$^{-1}$

**Urgcp mRNA, pi7 piRNA**

$k_{cleave} = 0.13 \pm 0.01$ min$^{-1}$

**Cox7a2l mRNA, pi4 piRNA**

$k_{cleave} = 0.003 \pm 0.002$ min$^{-1}$

**Extended Data Fig. 3 | MIWI cleaves synthetic *pi9* and *pi17* targets in vitro.** Product formed as a function of time by MIWI programmed with *pi9* and *pi17* piRNAs for three independent trials. Data are the means and SD of the three trials fit to the burst-and-steady-state equation. For gel source data, see Supplementary Fig. 1.

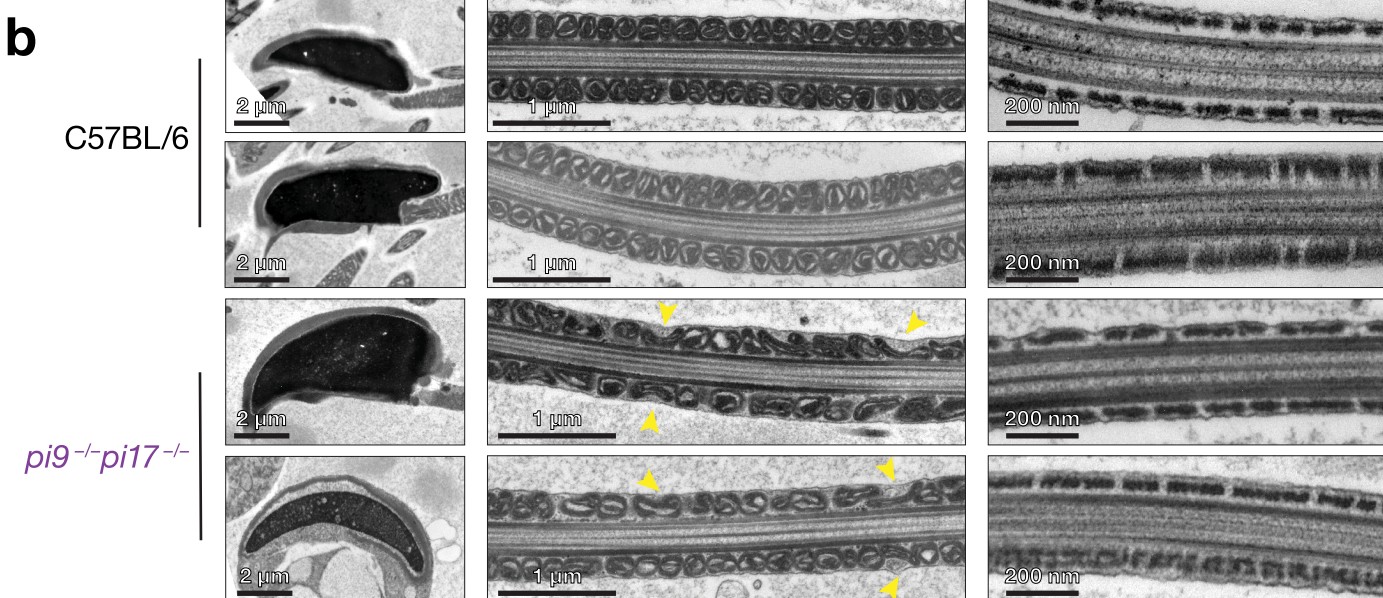

**Extended Data Fig. 4 | Testicular germ cell composition and sperm morphology in *pi9⁻ᐟ⁻*, *pi17⁻ᐟ⁻*, and *pi9⁻ᐟ⁻pi17⁻ᐟ⁻* males. a**, Representative (*n* = 3) hematoxylin and eosin staining of testis sections from 2–4-month-old C57BL/6, *pi9⁻ᐟ⁻*, *pi17⁻ᐟ⁻*, *pi2⁻ᐟ⁻pi17⁻ᐟ⁻*, *pi9⁻ᐟ⁻pi17⁻ᐟ⁻*, and *pi2⁻ᐟ⁻ pi9⁻ᐟ⁻ pi17⁻ᐟ⁻* males. **b**, Representative (*n* = 3) transmission electron micrographs of caudal sperm from C57BL/6 and *pi9⁻ᐟ⁻pi17⁻ᐟ⁻* males. Yellow arrowheads indicate deformed mitochondria.

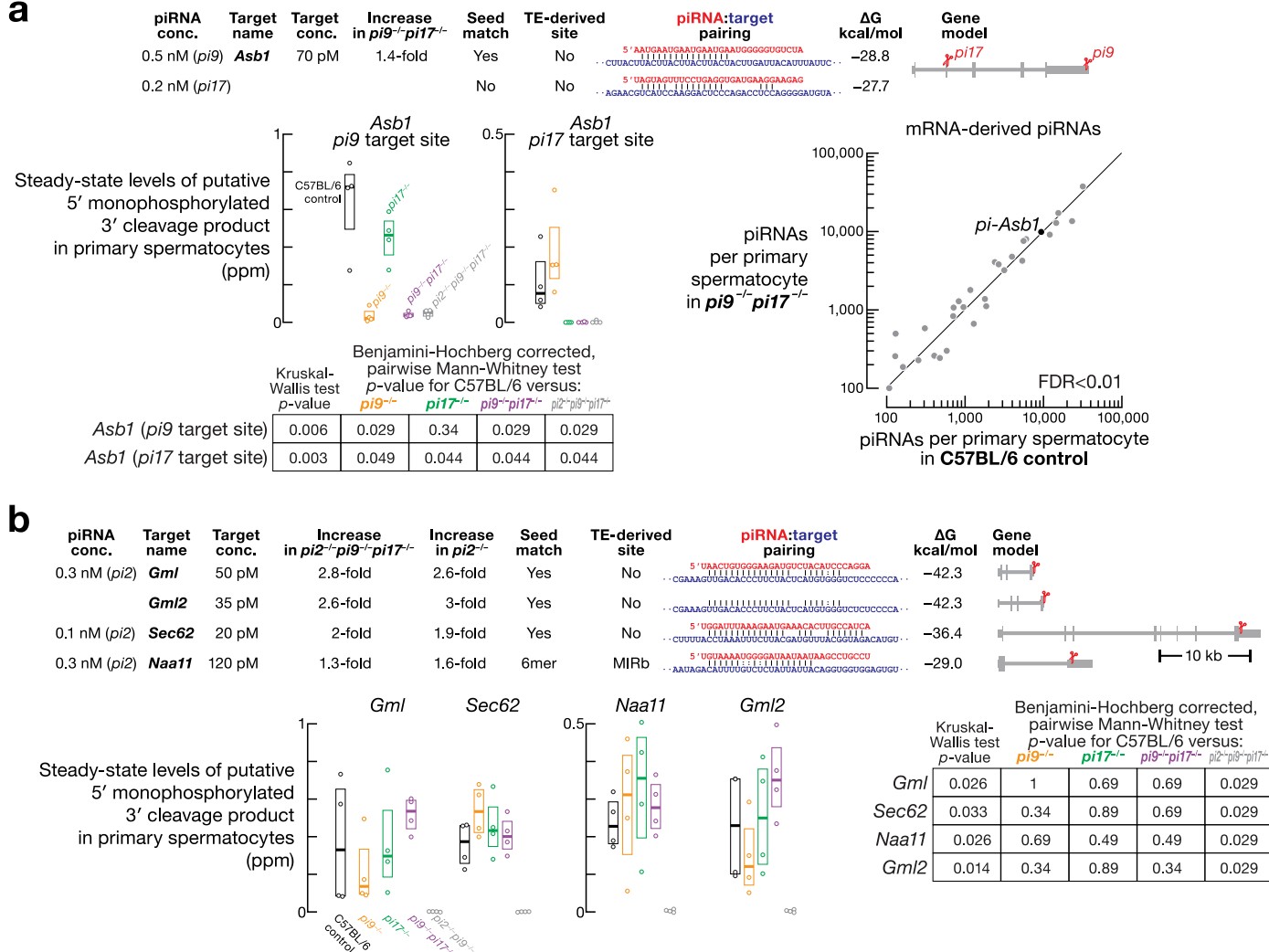

**Extended Data Fig. 5 | Direct targets of *pi2*, *pi9*, and *pi17* piRNAs. a**, At left, *Asb1* mRNA is cleaved by both a *pi9* and a *pi17* piRNA. Mean piRNA (*n* = 12) and target (*n* = 7) concentrations in C57BL/6 primary spermatocytes, extent of target increase in *pi9⁻/⁻ pi17⁻/⁻* primary spermatocytes, target site location in transcript, piRNA:target pairing pattern and binding energy (predicted Gibbs free energy, ΔG⁰) are shown. Presence of seed match (g2–g8) and whether cleavage site is found in a transposon fragment is indicated. Abundance in C57BL/6 (*n* = 4), *pi9⁻/⁻* (*n* = 4), *pi17⁻/⁻* (*n* = 4), *pi9⁻/⁻pi17⁻/⁻* (*n* = 4), and *pi2⁻/⁻pi9⁻/⁻ pi17⁻/⁻* (*n* = 4) primary spermatocytes of putative 5′-monophosphate-bearing cleavage products derived from *Asb1* target. Boxplots show median and interquartile range. At right, Kruskal-Wallis test *p*-values and Benjamini-Hochberg-corrected *p*-values for *post hoc* pairwise two-tailed Mann-Whitney tests are shown. At right, abundance of piRNAs derived from *Asb1* mRNA is unaltered in *pi9⁻/⁻ pi17⁻/⁻* primary spermatocytes. **b**, Direct targets of *pi2* piRNAs. Mean piRNA (*n* = 12) and target (*n* = 7) concentrations in C57BL/6 primary spermatocytes, extent of target increase in *pi2⁻/⁻* and *pi2⁻/⁻ pi9⁻/⁻ pi17⁻/⁻* primary spermatocytes, target site location in transcript, piRNA:target pairing pattern and binding energy (computationally predicted Gibbs free energy, ΔG⁰) are shown. Abundance in C57BL/6 (*n* = 4), *pi9⁻/⁻* (*n* = 4), *pi17⁻/⁻* (*n* = 4), *pi9⁻/⁻pi17⁻/⁻* (*n* = 4), and *pi2⁻/⁻pi9⁻/⁻pi17⁻/⁻* (*n* = 4) primary spermatocytes of putative 5′-monophosphate-bearing cleavage products derived from *pi2* targets. Boxplots show median and interquartile range. At right, Kruskal-Wallis test *p*-values (one-way ANOVA on ranks) and Benjamini-Hochberg-corrected *p*-values for *post hoc* pairwise two-tailed Mann-Whitney tests are shown for each gene.

| Genotype | Trial | Total Tubules examined | Tubules with TUNEL+ sperm | Mean ± SD |
|---|---|---|---|---|
|  | 1 | 121 | 3 (2.5%) |  |
| C57BL/6 | 2 | 245 | 47 (19.2%) | 10 ± 9% |
|  | 3 | 271 | 20 (7.4%) |  |
|  | 1 | 171 | 41 (24%) |  |
| *pi9*[−/−] | 2 | 106 | 16 (16%) | 19 ± 4% |
|  | 3 | 61 | 11 (18%) |  |
|  | 1 | 273 | 20 (7.3%) |  |
| *pi17*[−/−] | 2 | 274 | 15 (5.5%) | 5 ± 3% |
|  | 3 | 256 | 5 (2%) |  |
|  | 1 | 255 | 151 (59%) |  |
| *pi9*[−/−]*pi17*[−/−] | 2 | 255 | 77 (30%) | 40 ± 10% |
|  | 3 | 241 | 86 (36%) |  |
|  | 1 | 147 | 17 (12%) |  |
| *pi6*[−/−] | 2 | 63 | 5 (7.9%) | 20 ± 10% |
|  | 3 | 210 | 70 (33%) |  |

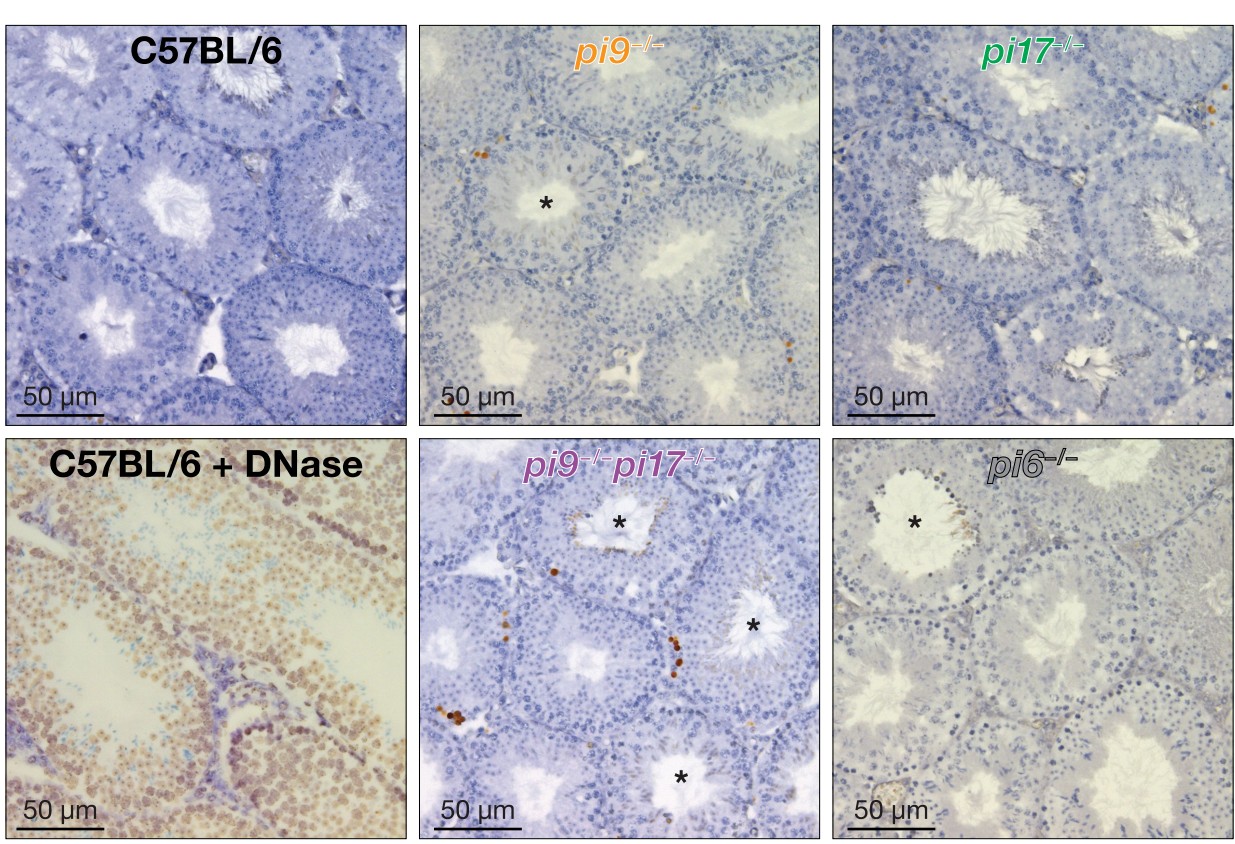

**Extended Data Fig. 6 | High incidence of double-stranded DNA breaks in *pi9*[−/−] *pi17*[−/−] testicular sperm.** Top, Fraction of seminiferous tubules with sperm staining positive for TUNEL (terminal deoxynucleotidyl transferase dUTP nick end labelling) of C57BL/6 and pachytene piRNA mutant testes. Data are mean and SD (*n* = 3). Bottom, representative images of seminiferous tubules of C57BL/6, C57BL/6 sections treated with DNase, and pachytene piRNA mutant testes. Brown, positive TUNEL reaction; blue, hematoxylin.

**a**

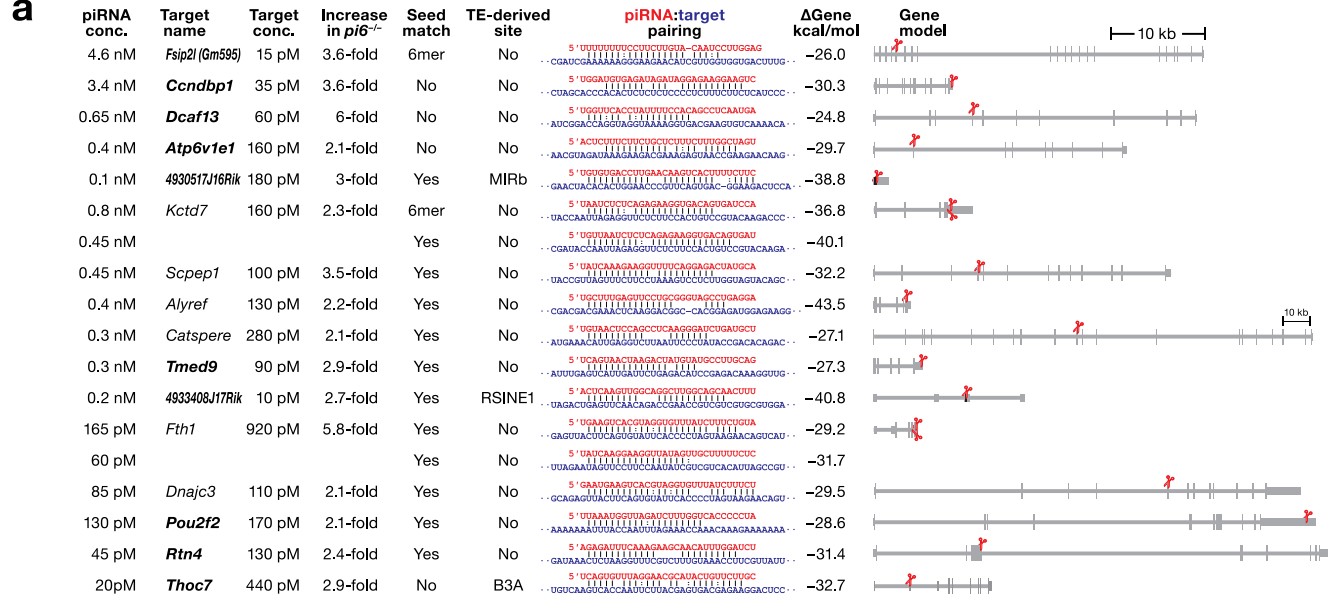

| piRNA conc. | Target name | Target conc. | Increase in *pi6⁻/⁻* | Seed match | TE-derived site | piRNA:target pairing | ΔGene kcal/mol | Gene model |
|---|---|---|---|---|---|---|---|---|
| 4.6 nM | *Fsip2l (Gm595)* | 15 pM | 3.6-fold | 6mer | No | | −26.0 | |
| 3.4 nM | **Ccndbp1** | 35 pM | 3.6-fold | No | No | | −30.3 | |
| 0.65 nM | **Dcaf13** | 60 pM | 6-fold | No | No | | −24.8 | |
| 0.4 nM | **Atp6v1e1** | 160 pM | 2.1-fold | No | No | | −29.7 | |
| 0.1 nM | *4930517J16Rik* | 180 pM | 3-fold | Yes | MIRb | | −38.8 | |
| 0.8 nM | *Kctd7* | 160 pM | 2.3-fold | 6mer | No | | −36.8 | |
| 0.45 nM | | | | Yes | No | | −40.1 | |
| 0.45 nM | *Scpep1* | 100 pM | 3.5-fold | Yes | No | | −32.2 | |
| 0.4 nM | *Alyref* | 130 pM | 2.2-fold | Yes | No | | −43.5 | |
| 0.3 nM | *Catspere* | 280 pM | 2.1-fold | Yes | No | | −27.1 | |
| 0.3 nM | **Tmed9** | 90 pM | 2.9-fold | Yes | No | | −27.3 | |
| 0.2 nM | *4933408J17Rik* | 10 pM | 2.7-fold | Yes | RSINE1 | | −40.8 | |
| 165 pM | *Fth1* | 920 pM | 5.8-fold | Yes | No | | −29.2 | |
| 60 pM | | | | Yes | No | | −31.7 | |
| 85 pM | *Dnajc3* | 110 pM | 2.1-fold | Yes | No | | −29.5 | |
| 130 pM | **Pou2f2** | 170 pM | 2.1-fold | Yes | No | | −28.6 | |
| 45 pM | **Rtn4** | 130 pM | 2.4-fold | Yes | No | | −31.4 | |
| 20 pM | **Thoc7** | 440 pM | 2.9-fold | No | B3A | | −32.7 | |

**b**

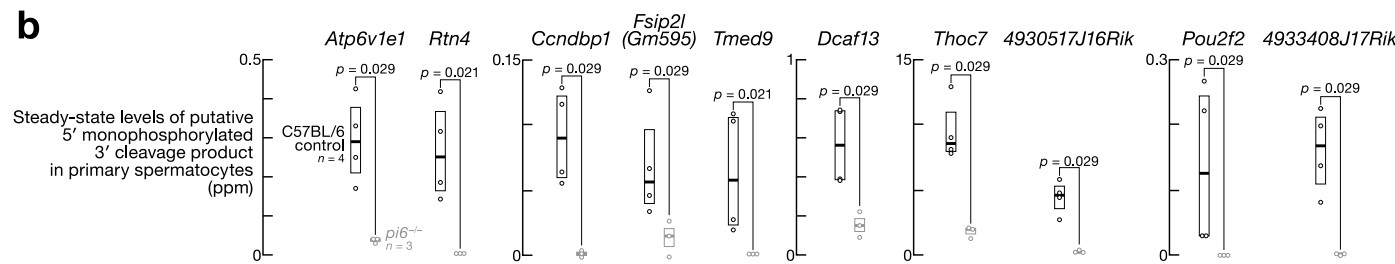

**Extended Data Fig. 7 | Direct targets of *pi6* piRNAs. a**, Median target increase in *pi6⁻/⁻* primary spermatocytes, secondary spermatocytes, and spermatids (ref. 17), mean piRNA (*n* = 12) and target (*n* = 7) concentrations in primary spermatocytes, target site location in transcript, piRNA:target pairing pattern and binding energy (computationally predicted Gibbs free energy, Δ*G*⁰) are shown. Presence of a seed match (g2–g8) and whether cleavage site is found in a transposon fragment is indicated. Cleavage of *Atp6v1e1* mRNA is guided by a *1-qC1.3-637* piRNA whose biogenesis is initiated by *pi6* piRNA-directed cleavage. Bold: targets identified in this study. **b**, Abundance in C57BL/6 (*n* = 4) and *pi6⁻/⁻* (*n* = 3) primary spermatocytes of putative 5′-monophosphate-bearing cleavage products derived from targets of *pi6* piRNAs (shown in Extended Data Fig. 4). Unpaired, two-tailed Mann-Whitney test *p*-values are shown. Boxplots show median and interquartile range.

**Extended Data Fig. 8 | Pachytene piRNAs do not use miRNA-like mechanism to regulate mRNA abundance. a**, Change in mean steady-state abundance in $pi9^{-/-} pi17^{-/-}$ ($n = 6$) vs C57BL/6 ($n = 3$) secondary spermatocytes, round spermatids, and elongating spermatids for mRNAs whose 3′UTRs pair to $pi9$ and $pi17$ piRNAs or to control piRNAs (i.e. piRNAs whose abundance does not change in $pi9^{-/-} pi17^{-/-}$ primary spermatocytes). Two-tailed KS test $p$-values are shown. **b**, Steady-state levels of $Grk4$ mRNA in C57BL/6 ($n = 3$), $pi9^{em1/em1} pi17^{em1/em1}$ ($n = 3$), and $pi9^{em2/em2} pi17^{em2/em2}$ ($n = 3$) primary spermatocytes, secondary spermatocytes, round spermatids, and elongating spermatids. Benjamini-Hochberg-corrected $p$-values for two-tailed Wald test calculated by DESeq2 are shown for each gene.

**a**

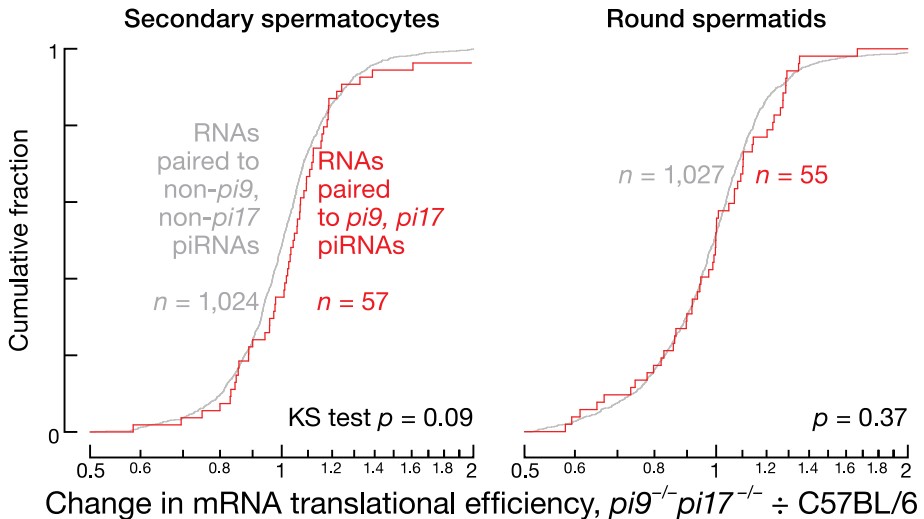

**mRNAs with ELAVL1-binding site and paired to g2–g8 + ≥12 bp to g9–g30**

**b**

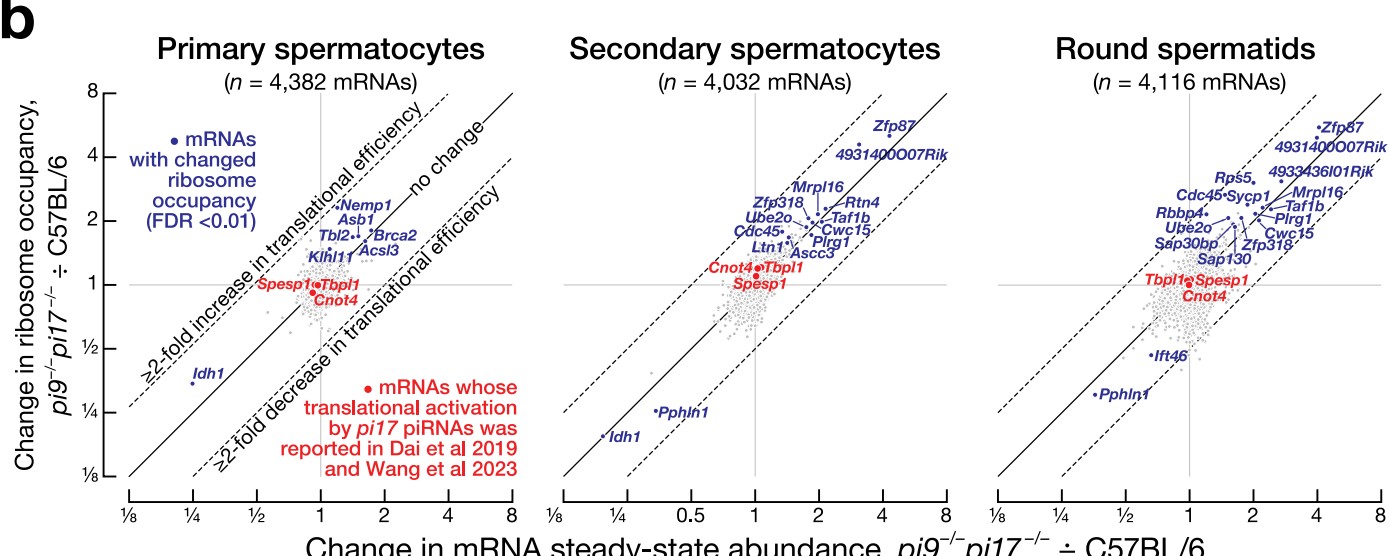

**Extended Data Fig. 9 | Pachytene piRNAs do not regulate mRNA translation. a**, Change in mean translational efficiency in FACS-purified $pi9^{-/-}pi17^{-/-}$ ($n = 4$) vs C57BL/6 ($n = 4$) secondary spermatocytes and round spermatids for mRNAs whose 3′UTRs pair to $pi9$ and $pi17$ piRNAs or to control piRNAs (i.e. piRNAs whose abundance does not change in $pi9^{-/-}pi17^{-/-}$ germ cells). Two-tailed Kolmogorov-Smirnov (KS) test $p$-values are shown. **b**, Change in mean steady-state abundance of mRNAs and their ribosome occupancy in FACS-purified $pi9^{-/-}pi17^{-/-}$ ($n = 4$) vs C57BL/6 ($n = 4$) primary spermatocytes, secondary spermatocytes and round spermatids. Data are for all mRNAs with ≥10 TPM ribosome occupancy in each cell type. mRNAs with significantly changed ribosome occupancy (FDR < 0.01) are shown in blue and were identified using Benjamini-Hochberg-corrected $p$-values for two-tailed Wald test calculated by DESeq2 (see also Methods).

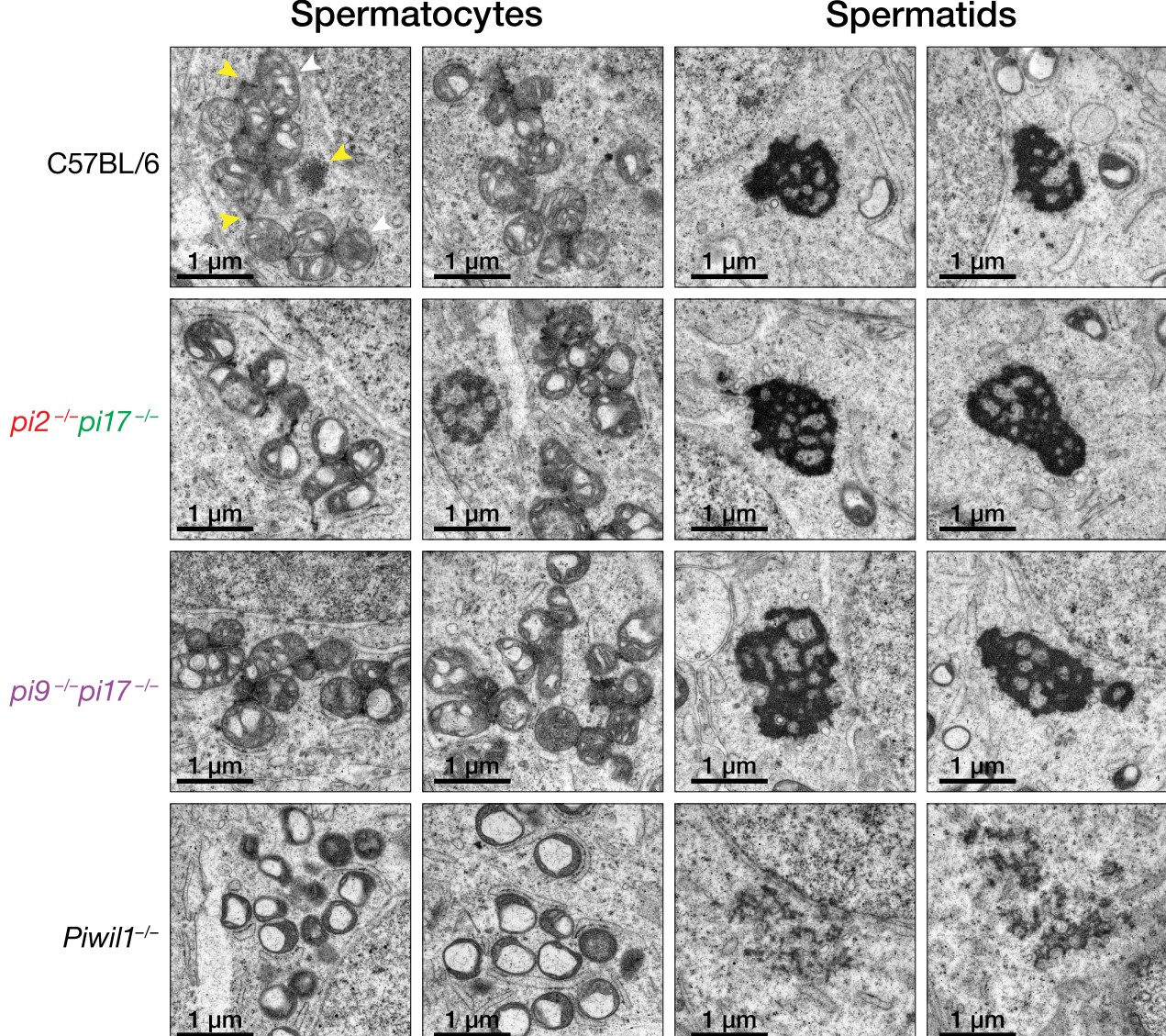

**Extended Data Fig. 10 | Pachytene piRNAs are dispensable for germ granule formation.** Representative ($n$ = 3) transmission electron micrographs of spermatocytes and round spermatids from C57BL/6, *Piwil1/Miwi*$^{-/-}$, and pachytene piRNA mutant testes. In primary spermatocytes, white arrowheads indicate mitochondria, and yellow arrowheads show intermitochondrial cement. In round spermatids, the electron-dense structures in the centre of micrographs are chromatoid bodies.

# Reporting Summary

## Statistics

For all statistical analyses, confirm that the following items are present in the figure legend, table legend, main text, or Methods section.

| n/a | Confirmed | |
|---|---|---|
| ☐ | ☒ | The exact sample size (*n*) for each experimental group/condition, given as a discrete number and unit of measurement |
| ☐ | ☒ | A statement on whether measurements were taken from distinct samples or whether the same sample was measured repeatedly |
| ☐ | ☒ | The statistical test(s) used AND whether they are one- or two-sided<br>*Only common tests should be described solely by name; describe more complex techniques in the Methods section.* |
| ☒ | ☐ | A description of all covariates tested |
| ☐ | ☒ | A description of any assumptions or corrections, such as tests of normality and adjustment for multiple comparisons |
| ☐ | ☒ | A full description of the statistical parameters including central tendency (e.g. means) or other basic estimates (e.g. regression coefficient) AND variation (e.g. standard deviation) or associated estimates of uncertainty (e.g. confidence intervals) |
| ☐ | ☒ | For null hypothesis testing, the test statistic (e.g. *F*, *t*, *r*) with confidence intervals, effect sizes, degrees of freedom and *P* value noted<br>*Give P values as exact values whenever suitable.* |
| ☒ | ☐ | For Bayesian analysis, information on the choice of priors and Markov chain Monte Carlo settings |
| ☒ | ☐ | For hierarchical and complex designs, identification of the appropriate level for tests and full reporting of outcomes |
| ☒ | ☐ | Estimates of effect sizes (e.g. Cohen's *d*, Pearson's *r*), indicating how they were calculated |

*Our web collection on statistics for biologists contains articles on many of the points above.*

## Software and code

Policy information about availability of computer code

| Data collection | BD FACSAria II Cell Sorter (644832; March 2009); Illumina NextSeq 550; Illumina NovaSeq 6000; ElemBio AVITI |
|---|---|
| Data analysis | The Code used to analyze sequencing data was deposited at GitHub and can be accessed at https://github.com/ildargv/Cecchini_et_al; other used software include bedtools (v2.3.4); fastx toolkit (v0.0.14); bowtie2 (v2.2.0); STAR (v2.3.1); StringTie (v1.3.4); DESeq2 (v1.18.1); bowtie (v1.0.0); SAMtools (v1.0.0); Microsoft Excel 2013; CRISPR design tool (v.RUO22-1364_001; https://www.idtdna.com/site/order/designtool/index/CRISPR_SEQUENCE); Tailor (v1.1; https://github.com/jhhung/Tailor); piPipes (v1.5.0; https://github.com/bowhan/piPipes) |

For manuscripts utilizing custom algorithms or software that are central to the research but not yet described in published literature, software must be made available to editors and reviewers. We strongly encourage code deposition in a community repository (e.g. GitHub). See the Nature Portfolio guidelines for submitting code & software for further information.

## Data

Policy information about availability of data

All manuscripts must include a data availability statement. This statement should provide the following information, where applicable:

- Accession codes, unique identifiers, or web links for publicly available datasets
- A description of any restrictions on data availability
- For clinical datasets or third party data, please ensure that the statement adheres to our policy

Sequencing data are available from the National Center for Biotechnology Information Small Read Archive using accession number PRJNA1176701 (https://

www.ncbi.nlm.nih.gov/bioproject/?term=PRJNA1176701); mouse genome sequence and annotation (build mm10/GRCm38.92) were downloaded from https://ftp.ensembl.org/pub/release-92/fasta/mus_musculus/dna/ and https://ftp.ensembl.org/pub/release-92/gtf/mus_musculus/; transposon consensus sequences were obtained from Repbase (v27.02; https://www.girinst.org/repbase/); Panther database can accessed at https://pantherdb.org/

## Research involving human participants, their data, or biological material

Policy information about studies with <u>human participants or human data</u>. See also policy information about <u>sex, gender (identity/presentation), and sexual orientation</u> and <u>race, ethnicity and racism</u>.

| | |
|---|---|
| Reporting on sex and gender | N/A |
| Reporting on race, ethnicity, or other socially relevant groupings | N/A |
| Population characteristics | N/A |
| Recruitment | N/A |
| Ethics oversight | N/A |

Note that full information on the approval of the study protocol must also be provided in the manuscript.

# Field-specific reporting

Please select the one below that is the best fit for your research. If you are not sure, read the appropriate sections before making your selection.

☒ Life sciences        ☐ Behavioural & social sciences        ☐ Ecological, evolutionary & environmental sciences

For a reference copy of the document with all sections, see <u>nature.com/documents/nr-reporting-summary-flat.pdf</u>

# Life sciences study design

All studies must disclose on these points even when the disclosure is negative.

| | |
|---|---|
| Sample size | No statistical method was used to determine the sample size. For biological experiments, sample size was n > 3 to ensure reproducibility, i.e., for effect sizes of >2-fold, Relative Standard Deviation was <50% for >90% of data. For biological samples, the maximum possible sample size (n = 4–12) was used for each type of data, ensuring that variability arising from all accountable sources was incorporated in the analyses (animal, day of data collection, reagent lots). For biochemical experiments, sample size was n = 3 to ensure reproducibility, i.e., for effect sizes of >2-fold, Relative Standard Deviation was <50% for >90% of data. |
| Data exclusions | No data were excluded from the analyses. |
| Replication | All data were collected during independent trials conducted on separate days. When using several types of data for analyses, all possible permutations of samples were analyzed (e.g., 4 control × 4 mutant data sets produced 16 permutations). All attempts at replication were successful. |
| Randomization | This study did not involve treatment or exposure of animals to any agent. Instead, the goal of this work was to compare untreated wild-type mice and untreated mutant mice lacking piRNAs from four genomic loci: all wild-type animals were compared to all mutant mice. Therefore, randomization is not relevant to this study. |
| Blinding | Blinding is not relevant to our study, because during analyses wild-type control and mutant data sets are easily identified. Blinding was not performed during data acquisition and/or analysis, because it is impossible to perform blinding as any experimenter analyzing data can readily distinguish between the mutant from wild-type control datasets from reads covering piRNA loci. |

# Reporting for specific materials, systems and methods

We require information from authors about some types of materials, experimental systems and methods used in many studies. Here, indicate whether each material, system or method listed is relevant to your study. If you are not sure if a list item applies to your research, read the appropriate section before selecting a response.

## Materials & experimental systems

| n/a | Involved in the study |
|---|---|
| ☐ | ☒ Antibodies |
| ☐ | ☒ Eukaryotic cell lines |
| ☒ | ☐ Palaeontology and archaeology |
| ☐ | ☒ Animals and other organisms |
| ☒ | ☐ Clinical data |
| ☒ | ☐ Dual use research of concern |
| ☒ | ☐ Plants |

## Methods

| n/a | Involved in the study |
|---|---|
| ☒ | ☐ ChIP-seq |
| ☐ | ☒ Flow cytometry |
| ☒ | ☐ MRI-based neuroimaging |

# Antibodies

| | |
|---|---|
| Antibodies used | Anti-SCP3 antibody (Abcam, ab15093); Anti-phospho-Histone H2A.X (Ser139) antibody, clone JBW301 (Millipore, 05-636, clone JBW301); Donkey anti-Mouse IgG (H+L) Highly Cross-Adsorbed Secondary Antibody, Alexa Fluor 594 (ThermoFisher, A-21203); Donkey anti-Rabbit IgG (H+L) Highly Cross-Adsorbed Secondary Antibody, Alexa Fluor 488 (ThermoFisher, A-21206); Anti-BrdU mouse biotin-conjugated antibody (Clone PRB-1, MilliporeSigma, MAB3262BMI) |
| Validation | Anti-SCP3 antibody (https://www.abcam.com/products/primary-antibodies/scp3-antibody-ab15093.pdf); Anti-phospho-Histone H2A.X (Ser139) antibody, clone JBW301 (https://www.emdmillipore.com/US/en/product/Anti-phospho-Histone-H2A.X-Ser139-Antibody-clone-JBW301,MM_NF-05-636#anchor_COA); Anti-BrdU mouse biotin-conjugated antibody (https://www.sigmaaldrich.com/deepweb/assets/sigmaaldrich/product/documents/228/850/mab3262b.pdf) |

# Eukaryotic cell lines

Policy information about cell lines and Sex and Gender in Research

| | |
|---|---|
| Cell line source(s) | HEK293T cells (lab stock) were obtained from ATCC. |
| Authentication | The cell lines were not authenticated; the cell lines were only to produce recombinant proteins. |
| Mycoplasma contamination | Not tested. |
| Commonly misidentified lines (See ICLAC register) | No commonly misidentified cell lines were used in the study. |

# Animals and other research organisms

Policy information about studies involving animals; ARRIVE guidelines recommended for reporting animal research, and Sex and Gender in Research

| | |
|---|---|
| Laboratory animals | 2–6-month old C57BL/6 wild-type and mutant adult male mice including MIWI mutants (Jackson Laboratory, MGI: 2182488) and piRNA locus mutants reported in this manuscript (Supplementary Table 1). |
| Wild animals | The study did not involve wild animals. |
| Reporting on sex | Only males have testes. |
| Field-collected samples | No field-collected samples were used in the study. |
| Ethics oversight | (1) PI on IACUC protocol: Phillip D. Zamore<br>(2) Name of IACUC: UMass Medical School Institutional Animal Care and Use Committee<br>(3) IACUC Docket: A2222-17, "Investigation of mechanisms of small RNA function in vivo" |

Note that full information on the approval of the study protocol must also be provided in the manuscript.

# Plants

Seed stocks

*Report on the source of all seed stocks or other plant material used. If applicable, state the seed stock centre and catalogue number. If plant specimens were collected from the field, describe the collection location, date and sampling procedures.*

Novel plant genotypes

*Describe the methods by which all novel plant genotypes were produced. This includes those generated by transgenic approaches, gene editing, chemical/radiation-based mutagenesis and hybridization. For transgenic lines, describe the transformation method, the number of independent lines analyzed and the generation upon which experiments were performed. For gene-edited lines, describe the editor used, the endogenous sequence targeted for editing, the targeting guide RNA sequence (if applicable) and how the editor was applied.*

Authentication

*Describe any authentication procedures for each seed stock used or novel genotype generated. Describe any experiments used to assess the effect of a mutation and, where applicable, how potential secondary effects (e.g. second site T-DNA insertions, mosiacism, off-target gene editing) were examined.*

# Flow Cytometry

## Plots

Confirm that:

☒ The axis labels state the marker and fluorochrome used (e.g. CD4-FITC).

☒ The axis scales are clearly visible. Include numbers along axes only for bottom left plot of group (a 'group' is an analysis of identical markers).

☒ All plots are contour plots with outliers or pseudocolor plots.

☒ A numerical value for number of cells or percentage (with statistics) is provided.

## Methodology

Sample preparation

Testes of 2–6-month-old mice were isolated, decapsulated, and incubated for 15 min at 33°C in 1× Gey's Balanced Salt Solution (GBSS, Sigma, G9779) containing 0.4 mg/ml collagenase type 4 (Worthington, LS004188) rotating at 150 rpm. Seminiferous tubules were then washed twice with 1× GBSS and incubated for 15 min at 33°C in 1× GBSS with 0.5 mg/ml Trypsin and 1 μg/ml DNase I, rotating at 150 rpm. Next, tubules were homogenized by pipetting through a glass Pasteur pipette for 3 min at 4°C. Fetal bovine serum (FBS; 7.5% f.c., v/v) was added to inactivate trypsin, and the cell suspension was then strained through a pre-wetted 70 μm cell strainer (ThermoFisher, 22363548); cells were collected by centrifugation at 300 × g for 10 min. The supernatant was removed, cells resuspended in 1× GBSS containing 5% (v/v) FBS, 1 μg/ml DNase I, and 5 μg/ml Hoechst 33342 (ThermoFisher, 62249) and rotated at 150 rpm for 45 min at 33ºC. Propidium iodide (0.2 μg/ml, f.c.; ThermoFisher, P3566) was added, and cells strained through a pre-wetted 40 μm cell strainer (ThermoFisher, 22363547).

Instrument

FACSAria II Cell Sorter (BD Biosciences; UMass Medical School FACS Core)

Software

BD FACSDiva (v9.0)

Cell population abundance

Spermatogonia: ~100,000 cells/animal; ~95–100% pure with ≤ 5% pre-leptotene spermatocytes;
Primary spermatocytes: ~1,000,000 cells/animal; ~10–15% leptotene/zygotene spermatocytes, ~45–50% pachytene spermatocytes, ~35–40% diplotene spermatocytes;
Secondary spermatocytes: ~1,000,000 cells/animal; ~100%;
Round spermatids: ~1,500,000 cells/animal; ~95–100%, ≤ 5% elongated spermatids.

Gating strategy

The gating strategy used to sort mouse primary germ cells is detailed in Supplementary Figure 5. Briefly, propidium iodide was used to label dead cells (top left panel in Supplementary Figure 5), forward and side scatter were used to isolate single cells (two top middle panels in Supplementary Figure 5), Hoechst 33342 emission in 450/50 and 670/50 bandpass filters was used to separate spermatogonia, spermatocytes, and spermatids (bottom left panel in Supplementary Figure 5). Forward scatter was then used to isolate round spermatids form the mixed population of round and elongated spermatids top right panel in Supplementary Figure 5). The percentages for each subpopulation are shown in the bottom right panel in Supplementary Figure 5.

☒ Tick this box to confirm that a figure exemplifying the gating strategy is provided in the Supplementary Information.

