## [Peer Review file · Nature]

Cleavage of mRNAs by a minority of pachytene piRNAs improves sperm fitness

Corresponding Author: Professor Phillip Zamore

Version 1:

Reviewer comments:

Referee #1

(Remarks to the Author)

Pachytene piRNAs represent the most abundant type of small non-coding RNAs in the mammalian testis. However, the biological function of this subtype of piRNAs and the specific mechanism by which they contribute to male fertility still remains to be elucidated. From the six largest pachytene piRNA loci in mice (pi2, pi6, pi7, pi9, pi17, and pi18), that produce about 40% of all pachytene piRNAs in mouse spermatocytes, only knockout of piRNAs derived from pi6 and pi18 have recently been described to cause male infertility.

To investigate the impact of additional piRNA-encoding loci and decipher the mechanism involved in pachytene piRNA-mediated function, Cecchini and co-workers generated double- and triple-piRNA loci mutant mice of pi2, pi7, pi9, and pi17 piRNA loci. None of the single mutant, but two double mutant combinations (pi7/pi9 and pi9/pi17), showed reduced fertility, while one triple mutant (pi2/pi9/pi17) was completely sterile. Functional sperm assays confirmed that pi2, pi9, and pi17 are essential for mature sperm production, with single mutants showing reduced sperm motility and capacitation. With the aim to identify the mechanisms contributing to the observed reproductive phenotype, the authors performed transcriptome analysis of pi9, pi17, and pi9/pi17 primary spermatocytes to determine the impact of these loci on gene expression. They found that while these mutations eliminated ~13.5% of pachytene piRNAs, only 23 transcripts were significantly altered. 14 of the 17 upregulated mRNAs were direct targets of pi9 or pi17 piRNAs, with RNA sequencing confirming their loss of cleavage in mutants. Several of these upregulated mRNAs encode proteins implicated in the DNA damage response, cell proliferation or apoptosis. In line, the testes of pi9/pi17 males showed increased double-stranded DNA breaks. The authors conclude that pi9 and pi17-mediated mRNA cleavage is essential for normal spermatogenesis and genome stability in developing sperm and that PIWI protein-mediated endonucleolytic cleavage is the primary mechanism by which pachytene piRNAs regulate gene expression in spermatogenesis.

However, since only 17 of 112 direct target mRNAs were upregulated in pi9/pi17 mutants, the authors suggest that piRNA-directed cleavage is widespread but finely tuned and plays a subtle role in gene regulation during spermatogenesis. Using global run-on sequencing, the authors show that transcripts cleaved by pachytene piRNAs have in general a high transcription rate and thus cleavage by pachytene piRNAs has only a limited impact on steady-state transcript levels, and pachytene piRNAs that alter transcript abundance are rare. In addition to analyzing the impact of abolished piRNA levels of specific piRNA loci on mRNA transcript levels, the authors focused on other suggested models for pachytene piRNA-mediated function such as an miRNA-like mechanism, regulation of mRNA translation, or impact on chromatoid body/IMC architecture. None of the mechanisms investigated was associated with abolished expression of pachytene piRNAs.

This is an interesting study using diverse knockout mouse models and different RNA sequencing to not only unravel the impact of different mouse pachytene piRNA loci on male fertility but also to decipher the biological mechanism contributing to the pachytene piRNA mediated function. However, we believe that some of the conclusions drawn by the authors, especially regarding the association between the identified upregulated mRNAs and the reproductive phenotype observed in the respective mouse models (decreased sperm fitness), are based on too few data and suggest further analysis to

support them.

1. In the fertility screening of the different piRNA loci the authors found a significant impact only for the combination of pi9/pi17 and pi2/pi9/pi17. How did the reproductive phenotype of the double mutant differ from that of the triple mutant? Why was the pi2/pi9/pi17 strain not included in the analysis of “sperm hyperactivated” or “fraction of progressive sperm”? Did these mice not produce any sperm at all and were they azoospermic? Did the authors observe any differences in the testicular phenotype of the different mice strains (amount of spermatogonia, spermatocytes, round spermatids, elongated spermatids)? With respect to the observed decreased sperm motility, was this linked to any changes in sperm morphology?

2. In the RNA-seq analysis, only pi9 and pi17 single mutant lines and pi9/pi17 double mutants but no pi2/pi9/pi17 triple mutants were included. Since only the double/triple mutants demonstrated impaired fertility, it would be interesting to see whether identical mRNAs were upregulated than in the single mutant strains. Accordingly, in Fig. 2, at least data from the double mutant should be included.

3. The effects observed at the transcriptional level are rather small. The authors highlight that among the 23 dysregulated transcripts in pi9 and pi17 KO primary spermatocytes, 17 mRNAs and 5 lncRNAs increased 1.4– to 4.4-fold and only one mRNA decreased 2.5-fold. However, among the 14 mRNAs that are targets of pi9 or pi17 piRNAs, 13 demonstrate a maximal increase of 1.4- to 2.4-fold and only 1 mRNA increased 4.4-fold. We believe that a comprehensive analysis of not only the pi9 and pi17 single mutants but also the double and triple mutants in comparison to data from infertile piRNA single loci ko mice such as pi6 or pi18 will be needed before conclusions on the cellular effect of these minimal changes (increased apoptosis, etc.) can be drawn. In line with this, the testicular sections could also be analyzed for further marker proteins such as γ H2AX or SYCP3 to identify the impact on progression of meiosis.

4. To better understand the different experiments and in which way they contribute to the main hypothesis of the manuscript that only changes in the transcriptional level of a few mRNAs are responsible for the observed decreased sperm fitness in pachytene piRNA locus mouse mutants, it would be beneficial to present a schematic figure summarizing the content and the key findings.

Referee #2

(Remarks to the Author)

This is a very intriguing study that argues that the majority of mouse pachytene piRNAs expressed from large intergenic piRNA cluster loci are not exhibiting an obvious target regulation event. However, a small minority of individual piRNAs could be essential for spermatogenesis by exhibiting detectable target silencing events. The authors argue that these few piRNAs are behind the evolutionary force to conserve and express a staggering repertoire of pachytene piRNAs to allow proper sperm formation. This study focuses on four mouse mutants that have deleted the promoters from the piRNA cluster loci of pi2, pi6, pi9 and pi17. Individual piRNA cluster mutants males are still able to sire litters, so the authors generated several double and triple mutants combinations. The combinations of the pi9 mutant with pi7 and pi8 with pi17 mutants shows the strongest loss of male fertility and sperm dysfunction, while combinations of the pi2, pi7 and pi17 mutations together that remove 19% of piRNAs are still nearly as fertile as wild type. This genetic data is both amazing and baffling that some piRNA cluster loci have compensation from the remaining piRNAs able to support normal spermatogenesis, while the loss of pi9 and pi17 mutated piRNA clusters individually and together exhibit the strongest infertility defects.

Are the transcriptome analysis of differential expression between pi9 and pi17 individual and double and triple mutants compared to the C57BL/6 controls also showing that no transposon RNAs and long-noncoding RNAs are upregulated in mutants like in pi2 or pi6 mutants? If I missed this analysis, could this be extracted from all the existing transcriptome and ribosome occupancy data that could also be subjected to mapping of reads to transposons and other long-noncoding RNAs. Do the new loosened complementarity rules for pachytene piRNA targeting increase the potential for more transposon transcripts to be targetable for cleavage?

The ribosome occupancy profile experiment with these piRNA mutants is a very important negative result to refute the controversial claim made by the Liu group that pachytene piRNAs may activate mRNA translation. I was impressed by multiple replicates of this RFP-seq profiling of mutants and wt control spermatocytes and spermatids, but the results are not clear how replicates were compared for reproducibility and RFP-seq library quality control (QC). To back up this negative result that there is no translational activation, authors should show this is not due to incomplete RFP-seq, so I would like to see some QC like polysome profiles and the RFP coverage plots on Cnot4, Tbp1, and Spesp1 with the periodic peaks over codon triplets. Similar QC results were in the Liu group's paper.

The authors should comment if they have biochemical data from their monumental study in Arif et al 2022 to validate in vitro cleavage by MIWI piRNA complexes to bolster the portrayed piRNA:target pairings for Urgcp, Cox7a2l and Zdhhc16 of Fig. 2b where there is interruption of the piRNA seed-sequence pairing. Would these piRNAs cleave a target in their MIWI in vitro cleavage assay?

I am not convinced by the authors' claim for why many putative piRNA-cleaved targets are not changing abundance in the piRNA mutants simply because they have a higher transcription rate compared to other non-piRNA target transcripts. Although Drosophila Piwi/piRNA complexes can impact a target transcript's transcription rate, mouse MIWI/piRNA cleavage should be independent of target transcription rate, and this argument of high transcription rate “buffering” out transcriptional change between WT and piRNA cluster mutant can only be substantiated if the piRNA cleavage activity is shown to be very weak at reducing steady state target levels. If piRNA cleavage activity is presumed strong like the impressive MIWI cleavage

activity assays shown in Arif et al, then regardless of target transcription rate the RNAseq measurements for WT and mutants are steady-state measurements that should show target up-regulation in the mutants. Could piRNA-target cleavage rates show wide variation and perhaps actually be weak for many piRNAs but only strong for the specific piRNAs highlighted in Fig. 2? Is there cleavage kinetics from the Arif et al assay that could substantiate variation and weak cleavage by some piRNAs?

This study is very significant, and as usual the Zamore lab presents high quality data and rigorous analysis. The crux of this paper is the very provocative interpretation that just a small set of individual piRNAs may have an outsized impact of target regulation while the rest of the piRNAs in the cluster may be useless bystanders. But I feel that a necessary genetic validation of this interpretation that is missing, and this would be for the authors generate additional mouse mutants that just delete the individual key piRNA sequences in the pi9 and pi17 piRNA loci for the piRNA sequences shown in Figure 2, and then see if those deletion mutants of a single piRNA sequence would have the same major spermatogenesis phenotypes as the deletion of the piRNA cluster promoter that removes all the other piRNAs from the locus.

I realized this additional experiment would extend the scope of this study, and to this point the Editor should arbitrate on this point I raise. The most feasible test could be to generate a mutant or mutants that delete just the region around the two individual pi9 piRNAs in Fig. 2A., leaving all the other pi9 cluster piRNAs intact, and see these single or smaller fraction piRNA-loss mutants phenocopy the pi9-promoter mutant? If the editor does not request this result for this study, then I recommend that the authors add this point to the paper discussion as a limitation in this study that individual piRNA mutants are not yet available to validate this provocative conclusion.

Minor point: The title says ["Mammalian" pachytene piRNAs persist...] but this study only has primary analysis of mouse piRNAs, no analysis in other mammals, so maybe change title to just "Mouse" pachytene piRNAs?

Referee #3

(Remarks to the Author)

In this manuscript, Cecchini et al. characterize the effects of deleting four pachytene piRNA clusters—either separately or in various combinations—on the expression of protein-coding genes. They found that only a small subset (1%) of pachytene piRNAs can target host genes. Among the putative piRNA targets, only a small fraction shows altered expression following the loss of clusters. The authors attribute the lack of transcriptional changes to the high expression of most targeted genes. Interestingly, those genes that do show differential expression tend to be enriched in genes whose derepression impedes spermatogenesis. This suggests that piRNAs are necessary to keep the expression of these genes low in order to maintain fitness. The authors argue that this fitness advantage, derived from regulating a small number of genes, ensures the survival of all pachytene piRNAs, the majority of which are considered selfish elements. They also propose that natural selection eliminates piRNAs that would cause detrimental changes in gene expression, thereby limiting the number of genes affected by piRNA regulation. Finally, they find that genes regulated by piRNAs are disproportionately targeted at TE-derived sequences within transcripts recognized by piRNAs complementary to or derived from transposon sequences. This suggests an important role for transposons in the evolution of host gene regulation. Their findings provide strong evidence for gene regulatory function of pachytene piRNAs, and support for their retention in the genome.

Experiments are well designed, with necessary controls and appropriate statistical analysis to support the claims. The text is well written and clear, although some of the figures might benefit from simplifications to more clearly convey the message.

Cecchini et al. also carefully investigate the mechanism by which piRNAs influence gene expression. Previous studies have suggested that, in addition to transcript cleavage, piRNAs also impact translation efficiency and induce deadenylation of target transcripts. However, the authors find no evidence to support these claims.

Unfortunately, the piRNA field has been impacted by high-profile publications that claim broad functions for pachytene piRNAs in gene regulation and propose diverse, unexpected mechanisms of action—some of which contradict well-established knowledge about piRNAs. The authors of this manuscript provide a much-needed service to the small RNA field and the broader RNA and gene regulation community by examining two controversial topics through rigorously planned and executed experiments and proper statistical analysis:

1. Disproving claims about the generalized gene regulatory function of pachytene piRNAs.
2. Demonstrating that piRNAs regulate gene expression solely through endonucleolytic cleavage of complementary targets, rather than inducing deadenylation or regulating translation.

Although the general findings of this work that identify true pachytene piRNA targets that explain the mutant phenotype, and the detailed dissection of mechanism of action provide ample novelty, even without additional novelty this work should be strongly considered for publication in Nature to set the record about piRNA targets and mechanism straight.

Below, I summarize some comments and suggestions for modifications and additional analyses that might further enhance the manuscript.

Comments and Suggestions:

- Statistically significant effects on fertility are only observed when different cluster mutations are combined, implying functional redundancy between clusters. It would be interesting to determine whether these redundancies in targeting are observable at the sequence level. Do multiple clusters produce piRNAs with sufficient complementarity to a given gene such that the loss of one cluster does not significantly reduce piRNA levels to the point of altering gene expression, but the loss of several clusters does? Alternatively, are multiple genes targeted independently by different clusters, with the additive effect of derepressing several genes necessary to observe the decreased fertility phenotype?

A potential follow-up, albeit a time-consuming one, could be to test whether changes in expression of the identified deregulated genes (e.g., those involved in DSB repair) account for the observed phenotype. Would the relatively mild 1.5x-2x overexpression of target transcripts regulated by piRNAs be sufficient to induce a fitness loss in sperm or an increase in DSBs? Would this effect depend on the simultaneous overexpression of multiple genes?

- Related to the previous question: Effective targeting (i.e., reduction in transcript levels) could be accomplished by a single highly expressed piRNA, as seen in the examples in Fig. 2. Alternatively, it could result from many independent but lower-concentration piRNAs acting along the transcript. These could, in theory, come from multiple clusters, even if individual targeting sequences are unique to specific clusters. Could "weak but parallel" targeting of certain genes by piRNAs from several clusters explain the lack of broader changes in gene expression observed when only 1–3 clusters are deleted? Theoretically, such parallel targeting by many clusters could add robustness, allowing for some level of sequence drift without disrupting overall gene regulation by piRNAs.

- Do expression levels of transposable elements (TEs) also change in the cluster mutants? Although pachytene piRNAs contain few TE sequences, some still regulate host genes. Could derepression of specific transposons contribute to the observed fertility effects?

- Are genes regulated by TE-derived piRNAs also regulated by pre-pachytene piRNAs? One might predict that pre-pachytene piRNA sequences could have higher concentrations and thus repress these genes more strongly at earlier developmental stages.

- The almost identical target sequence in four genes regulated by pi9 suggests co-regulation of these genes. Is the URR1B sequence overrepresented in 3' UTRs in general, or is the presence of this URR1B site in these genes the result of natural selection to co-regulate them? How does the expression of these four genes in mouse spermatocytes compare to that in other, closely or distantly related species?

- The authors argue that natural selection eliminates deleterious targeting, and that the genes that piRNAs effectively target provide benefits that help maintain pachytene piRNA production. Alternatively, one could view this as a form of "piRNA addiction." It is possible that the original expression levels of these genes were low, and their promoters "adjusted" to compensate for piRNA-mediated silencing. Without piRNAs, their expression might now be too high, creating a dependency on piRNA silencing. For genes where higher expression is not deleterious, the host could have adjusted by increasing gene expression, which could explain why most targeted genes are highly expressed and unaffected. Instead of natural selection eliminating the piRNA, the target gene's expression level could have been increased. The authors could test whether orthologs in closely and distantly related species are expressed at similarly low levels when complementary piRNAs are absent. The results could further refine the understanding of the evolutionary importance of piRNAs in host gene regulation and the co-evolution of piRNA targets.

Minor Comments:

- Fig. 3 is difficult to interpret, as it contains many text blocks that are not intuitive to navigate. For example, the text blocks that describe "n=16 permutations..." could be simplified by replacing this with asterisks (*) and defining them in the figure legends. Additionally, the bars showing the stats in Figures 3c and 3d are hard to interpret. Could these be simplified to represent all 16 permutations with one IQR and median? In particular, in Fig. 3d, the evolutionary significance would be clearer if more attention were drawn to the difference between piRNAs that cause changes in gene expression versus those that do not, with the genomic features shown below, including those with little (e.g., all pachytene piRNAs) or high (CDS) conservation.

- The connection between the lack of change in germ cell granule formation and the paper's narrative is not entirely clear. I assume this result is meant to show that overall piRNA processing/cleavage functions are not affected by gross structural changes, and that the observed sterility phenotype is primarily due to the loss of transcript repression. This section would benefit from a sentence explicitly linking this result to the paper's conclusions.

- How is "piRNAs removed" in panel 1a defined? Does it refer to reads or sequences? Does "removed sequence" mean completely absent in the mutant library, or is there a threshold for fold decrease or maximum read count?

- Do the authors believe that the decreased *Idh1* levels in the pi9 mutant (Fig. 2a) are secondary?

- In the discussion, I suggest modifying the sentence "only several hundred are sufficiently abundant and complementary to a transcript to direct its cleavage," as this implies that abundance affects cleavage. Abundance actually only impacts detectable changes in transcript levels due to cleavage.

Version 2:

Reviewer comments:

Referee #1

(Remarks to the Author)

In the revised version, the authors have addressed most of our comments, yet the following point remains unclear and would benefit from rephrasing:

1. The authors included transcriptome analyses in spermatocytes of single pi9^{-/-}, single pi17^{-/-}, double pi9^{-/-}/pi17^{-/-} and triple mutants (pi9^{-/-}-pi17^{-/-} and pi2^{-/-}) in Figure 2c, demonstrating that the 14 pi9 and pi17 targets are de-repressed to the same extent in single, double, and triple mutants. However, in the manuscript text on page 7 they are not referring to the triple mutant but only to the double mutant and write: "To identify the molecular cause of pi9^{-/-} pi17^{-/-} fertility defects, we compared the transcriptomes of pi9^{-/-} , pi17^{-/-} , and pi9^{-/-} pi17^{-/-} primary spermatocytes. The abundance of pi17 targets increased to the same extent in pi17^{-/-} and pi9^{-/-} pi17^{-/-} primary spermatocytes (Fig. 2c). pi9 targets were also similarly (±50%) derepressed in pi9^{-/-} and pi9^{-/-} pi17^{-/-} males (Fig. 2c)." We suggest to explain all data of Figure 2c in the respective section. The result from the triple mutant is only mentioned on page 9, when the authors explain that targets co-regulated by piRNAs from different loci are rare.

2. The triple mutant is not mentioned in Figure legend 1B/C.

3. A figure legend for Figure 5 is missing.

(Remarks on code availability)

Referee #2

(Remarks to the Author)

I have completed my assessment of the revision and rebuttal letter. I am satisfied with the author response in the rebuttal letter, but have two more suggestions on the new data and updated tables provided in the revision. These small suggestions once addressed by the reviewers would complete my review of this study, happy to see it get accepted for publication.

1) There is new analysis in this revision regarding the one common mRNA being targeted by piRNA complementarity to pi9 and pi17, and that Asb1 mRNA is only de-repressed when both pi9 and pi17 were both removed. Since this is a new finding being reported in this revision, I would like the authors to comment in the final manuscript that when Asb1 mRNA is upregulated 1.4 fold in the double pi9&pi17 mutant, are the 3'UTR Asb1 piRNAs also increased in a greater capacity compared to other genic 3'UTR piRNA clusters? Just looking at genic piRNA cluster and not other intergenic clusters should serve as a control for the overall change in piRNA sequences being lost in the double pi9&pi17 mutant?

2) Some of the new Supplemental tables numbers have too many digits beyond significance that make reading the tables difficult (yes, I know I can download into Excel and change the formatting myself, but authors should make it easier for readers to follow these tables).

3) The Github repository is missing a Readme file to explain the code and input files.

(Remarks on code availability)

The Github repository is missing a Readme file to explain the code and input files.

Referee #3

(Remarks to the Author)

As already described in detail in my first review, I find the work significant, rigorously executed with appropriate statistical analyses and well described. The conclusions are robust (apart from a few comments below), the references appropriate and the key results for the most part unchanged compared to the previous submission. Overall, I find that the authors answered the concerns of the reviewers to my satisfaction.

I would like to push back on the authors' argument for why they believe that promoter strength "adjustment" is not contributing to a "piRNA addiction" model. In the rebuttal, the authors argue that such promoter strength adjustment "requires that the cleavage-induced decrease in the target level does not reduce sperm fitness, i.e., the piRNA-directed cleavage is neutral. Otherwise, the piRNA or the target site would be purged by negative selection even before the promoter is adjusted."

I disagree and would like to provide two counterarguments:

First, their own analysis in response to another concern showed that the transcript levels varied ~2x between closely related species for the 14 identified pi9 and pi17 targets, despite all target-piRNA pairs being conserved. The fold changes in gene expression in the cluster mutants compared to wild type were 1.4-4.4x with a median of 2x. The similar level of variance in target expression between the subspecies to the effect of piRNAs implies that there is often plenty of tolerance for variable expression, so negative selection pressure might not be very strong for 2x changes in gene expression.

Second and more important, while the authors are correct that negative selection would purge any piRNAs or target sites that cause very severe fitness changes, such as complete sterility, this does not necessarily hold true for milder fitness reduction on a population level. In the case of moderate fitness cost, deleterious piRNAs could either be lost through negative selection (as suggested by the authors) or compensatory mechanism, such as hyperactive promoters, could be positively selected to restore expression. Importantly, the promoter “adjustment” that may occur is on the population scale, meaning they are not necessarily changing in response to piRNA; rather, through the random walk of evolution, hyperactive promoter variants may arise and be present in the population in a subset of individuals. While the higher transcript level might be disadvantageous under normal circumstances, these individuals would be positively selected when pervasive deleterious piRNAs emerge. This way, deleterious piRNAs could persist, while enhanced target promoter expression becomes the new steady-state level in subsequent generations/populations. In the dataset in this paper, these targets may appear as genes that are highly transcribed and thus unaffected by small amounts of piRNA... but they may have originally been expressed as low level genes that were subjected to piRNA repression, resulting in a shift at the population level towards enhanced expression of the targeted gene to escape the repressive effect, but becoming “addicted” to the piRNA to counter the deleterious effect of even higher gene expression. Based on the presented data, one cannot assume that high expression genes were highly expressed before they became targets.

As a side note, this counterargument also implies that the model presented in Fig. 5 with four different types of pachytene piRNAs might be an over-simplification of the evolutionary history of these targets and to me it does not provide a lot of aid to understand the key findings. That said, I found the text sufficiently clear without any figure.

Minor comments:

1) The following section is difficult to understand:

“The steady-state abundance of Asb1 mRNA was unchanged in pi9^{-/-} and pi17^{-/-} single mutants yet increased 1.4-fold in pi9^{-/-} pi17^{-/-} primary spermatocytes. These data suggest that cleavage by a single pi9 or pi17 piRNA is insufficient to alter Asb1 steady-state levels; Asb1 derepression was only detectable when the pi9 and pi17 piRNAs were both removed (Extended Data Fig. 5a).”

The first and 3rd sentence essentially say the same thing and thus the 3rd sentence is superfluous. However, the logic of the 2nd sentence is confusing, and I believe incorrect. In wild type (piRNAs from both clusters) and single mutants (piRNAs from only one cluster) transcript levels are the same. This I interpret as piRNAs from a single cluster are sufficient to get the transcript to the lower level that is seen in wild type. Only when both clusters are mutated (i.e., there is no mapping piRNA left) is there an increase in transcript level. Thus, the two clusters act redundantly in repressing Asb1 and activity of either is sufficient to induce the silencing of Asb1 to “WT” levels.

2) In the section titled “Pachytene piRNAs cleave hundreds of transcripts but alter the abundance of just a minority of targets” the authors introduce/define the different rules that need to be satisfied to be considered a target. However, they refer to piRNA targets in the previous sections. Were those piRNA targets identified based on sequence complementarity or also based on these strict criteria? If the latter, then this should be defined earlier in the results. For example, in the section before these target rules are introduced the authors write: “Among the 26 RNAs derepressed in pi2^{-/-} pi9^{-/-} pi17^{-/-} but not pi9^{-/-} pi17^{-/-}, we identified four mRNAs cleaved exclusively by pi2 piRNAs but could not find any transcripts repressed by piRNAs from pi2 and pi9 or pi17 (Extended Data Fig. 5b).”

Are the targets “cleaved exclusively by pi2 piRNAs” defined based solely on transcript differential expression in knockouts or piRNA complementarity alone; or is it based on the cleavage criteria defined in the following section?

3) Figure legends are missing for Fig. 5 (model). While we appreciate the acknowledgement of the “piRNA addiction” phrase, it is not completely clear from the text and the model what it refers to. My interpretation was more towards the “adjustment of transcript level” that then forces the need to retain piRNAs against those genes, while the authors seem to use it to explain why due to the need of a few piRNAs that regulate genes, all the other piRNAs are also retained, keeping the pachytene piRNAs present. Both are valid (and in some ways functionally overlapping) interpretations, but I would suggest explaining the term clearly if it is introduced.

4) In Fig. 2c, I suggest adding the “pi2 targets” explanation for the pink dots to the legend, or, similar to the other categories, on the side, rather than directly on the plot.

5) I find the sentence on p. 16 “The change in translational efficiency between pi9^{-/-}pi17^{-/-} and C57BL/6 was indistinguishable comparing the putative pi9 and pi17 targets and the putative targets of non-pi9 and non-pi17 piRNAs in primary spermatocytes (two-tailed KS test p-value = 0.64), secondary spermatocytes (p = 0.09), and round spermatids (p = 0.37; Fig. 4b and Extended Data Figs. 12a and 12b)” very difficult to understand and would suggest rephrasing.

(Remarks on code availability)

Version 3:

Reviewer comments:

Referee #3

(Remarks to the Author)

Responses to Reviewers' Critiques

We thank the Reviewers for their useful comments that have substantially improved our work. The revised manuscript includes additional biochemical data as well as phenotypic and molecular characterization of the double and triple mutant males. In particular, we are grateful for the suggestion to investigate the molecular basis of the apparent genetic redundancy among piRNA-producing loci. Our analyses of data from $pi9^{-/-}pi17^{-/-}$ and $pi2^{-/-}pi9^{-/-}pi17^{-/-}$ males showed that different piRNA-producing loci rarely have common targets. Instead, targets of different piRNA loci often act in the same molecular pathways and their simultaneous derepression likely causes the fertility defects in $pi9^{-/-}pi17^{-/-}$ and $pi2^{-/-}pi9^{-/-}pi17^{-/-}$ mutants. Analyses suggested by the Reviewers also helped better explain why cleavage by most piRNAs is inconsequential for target steady-state abundance. Both low efficacy of piRNA-guided cleavage and high target transcription rate preclude most pachytene piRNAs from altering the steady-state abundance of their targets. Below are our detailed responses to the reviewers' comments.

Referee #1

Pachytene piRNAs represent the most abundant type of small non-coding RNAs in the mammalian testis. However, the biological function of this subtype of piRNAs and the specific mechanism by which they contribute to male fertility still remains to be elucidated. From the six largest pachytene piRNA loci in mice ($pi2$, $pi6$, $pi7$, $pi9$, $pi17$, and $pi18$), that produce about 40% of all pachytene piRNAs in mouse spermatocytes, only knockout of piRNAs derived from $pi6$ and $pi18$ have recently been described to cause male infertility.

To investigate the impact of additional piRNA encoding loci and decipher the mechanism involved in pachytene piRNA mediated function, Cecchini and co-workers generated double- and triple-piRNA loci mutant mice of piRNA loci $pi2$, $pi7$, $pi9$, and $pi17$. None of the single mutant but two double mutant combinations ($pi7/pi9$ and $pi9/pi17$) showed reduced fertility, while one triple mutant ($pi2/pi9/pi17$) was completely sterile. Functional sperm assays confirmed that $pi2$, $pi9$, and $pi17$ are essential for mature sperm production, with single mutants showing reduced sperm motility and capacitation. With the aim to identify the mechanisms contributing to the observed reproductive phenotype, the authors performed transcriptome analysis of $pi9$, $pi17$, and $pi9/pi17$ primary spermatocytes to determine the impact of these loci on gene expression and found that while these mutations eliminated ~13.5% of pachytene piRNAs, only 23 transcripts were significantly altered. 14 of the 17 upregulated mRNAs were direct targets of $pi9$ or $pi17$ piRNAs, with RNA sequencing confirming their loss of cleavage in mutants. Several of these upregulated mRNAs encode proteins implicated in the DNA damage response, cell proliferation or apoptosis. In line, the testes of $pi9/pi17$ males showed increased double-stranded DNA breaks. The authors conclude that $pi9$ and $pi17$ -mediated mRNA cleavage is essential for normal spermatogenesis and genome stability in developing sperm and that PIWI-

protein-mediated endonucleolytic cleavage is the primary mechanism by which pachytene piRNAs regulate gene expression in spermatogenesis.

However, since only 17 of 112 direct target mRNAs were upregulated in *pi9/pi17* mutants the authors suggest that piRNA-directed cleavage is widespread but finely tuned and plays a subtle role in gene regulation during spermatogenesis. Using global run-on sequencing, the authors show that transcripts cleaved by pachytene piRNAs have in general a high transcription rate and thus cleavage by pachytene piRNAs has only a limited impact on steady-state transcript levels, and pachytene piRNAs that alter transcript abundance are rare. In addition to analyzing the impact of abolished piRNA levels of specific piRNA loci on mRNA transcript levels, the authors focused on other suggested models for pachytene piRNA mediated function such as miRNA-like mechanism, regulation of mRNA translation, or impact on chromatoid body/IMC architecture. None of the mechanisms investigated was associated with abolished expression of pachytene piRNAs.

This is an interesting study using diverse knockout mouse models and different RNA sequencing to not only unravel the impact of different mouse pachytene piRNA loci on male fertility but also to decipher the biological mechanism contributing to the pachytene piRNA mediated function.

However, we believe that some of the conclusions drawn by the authors, especially regarding the association between the identified upregulated mRNAs and the reproductive phenotype observed in the respective mouse models (decreased sperm fitness), are based on too few data and suggest further analysis to support them.

1. In the fertility screening of the different piRNA loci the authors found a significant impact only for the combination of *pi9/pi17* and *pi2/pi9/pi17*. How did the reproductive phenotype of the double mutant differ from that of the triple mutant? Why was the *pi2/pi9/pi17* strain not included in the analysis of “sperm hyperactivated” or “fraction of progressive sperm”? Did these mice not produce any sperm at all and were azoospermic? Did the authors observe any differences in the testicular phenotype of the different mice strains (amount of spermatogonia, spermatocytes, round spermatids, elongated spermatids? With respect to the observed decreased sperm motility, was this linked to any changes in sperm morphology?

Both pi9^{-/-}pi17^{-/-} and pi2^{-/-}pi9^{-/-}pi17^{-/-} males produce sperm. Fig. 1 and Extended Data Fig. 1 now contain new data for pi2^{-/-}pi9^{-/-}pi17^{-/-}, including fractions of motile, progressive, and hyperactivated sperm, embryo counts, and the percentage of successfully fertilized oocytes in vitro (IVF). Additional trials of fertility tests for pi2^{-/-}pi9^{-/-}pi17^{-/-} males included in the revised manuscript demonstrate that these mutants are capable of siring pups in vivo, albeit at a much lower frequency than pi9^{-/-}pi17^{-/-} males (Fig. 1 and Extended Data Fig. 1). We therefore updated the text to state that both pi2^{-/-}pi9^{-/-}pi17^{-/-} and pi9^{-/-}pi17^{-/-} males produce fewer litters than the control mice, with the triple mutant exhibiting more severe fertility defects.

The revised manuscript adds H&E-stained testis sections showing no difference in testicular germ cell composition among C57BL/6, $pi9^{-/-}$, $pi17^{-/-}$, $pi9^{-/-}pi17^{-/-}$, and $pi2^{-/-}pi9^{-/-}pi17^{-/-}$ males (Extended Data Fig. 4a). New transmission electron microscopy images also demonstrate no changes in the sperm gross morphology but reveal deformed mitochondria in the midpiece of $pi9^{-/-}pi17^{-/-}$ caudal sperm (Extended Data Fig. 4b). Our molecular analyses suggest that the combined derepression of the 14 $pi9$ and $pi17$ target mRNAs explain the fertility defects observed in $pi9^{-/-}pi17^{-/-}$. Similarly, the fertility defects in $pi2^{-/-}pi9^{-/-}pi17^{-/-}$ males can be explained by the combined derepression of the 14 $pi9$ and $pi17$ target mRNAs plus potential contributions from three $pi2$ targets with putative roles in regulation of apoptosis (*Gml*, *Gml2*; PMID: 10717525) and translation of ER-targeted proteins (*Sec62*).

2. In the RNAseq analysis, only $pi9$ and $pi17$ single mutant lines and $pi9/pi17$ double mutants but no $pi2/pi9/pi17$ triple mutants were included. Since only the double/triple mutants demonstrated impaired fertility, it would be interesting to see whether identical mRNAs were upregulated than in the single mutant strains. Accordingly, in Figure 2, at least data from the double mutant should be included.

The reviewer's comment encouraged us to investigate the molecular basis for the apparent genetic redundancy of $pi9$ and $pi17$ loci. Analyses of $pi9^{-/-}pi17^{-/-}$ and $pi2^{-/-}pi9^{-/-}pi17^{-/-}$ data are now included in the revised text and figures (Fig. 2c and Extended Data Fig. 5). We find that the 14 $pi9$ and $pi17$ targets are derepressed to the same extent in single, double, and triple mutants (Fig. 2c).

Remarkably, our data also showed that the simultaneous derepression of the 14 $pi9$ and $pi17$ targets in $pi9^{-/-}pi17^{-/-}$ is accompanied by changes in the abundance of 323 transcripts that are not direct targets of $pi9$ and $pi17$ piRNAs. Consistent with the elevated incidence of dsDNA breaks in $pi9^{-/-}pi17^{-/-}$ testicular sperm, Gene Ontology enrichment analyses showed that many of these 323 RNAs are implicated in regulating cell cycle progression and chromosome segregation. In $pi2^{-/-}pi9^{-/-}pi17^{-/-}$ primary spermatocytes, in addition to the 323 RNAs, the levels of 59 more transcripts were altered. Among the 26 RNAs derepressed in $pi2^{-/-}pi9^{-/-}pi17^{-/-}$ but not $pi9^{-/-}pi17^{-/-}$, we identified four mRNAs cleaved exclusively by $pi2$ piRNAs but could not find any transcripts repressed by piRNAs from $pi2$ and $pi9$ or $pi17$.

Among the transcripts derepressed in $pi9^{-/-}pi17^{-/-}$ and $pi2^{-/-}pi9^{-/-}pi17^{-/-}$ males, we identified a single mRNA that is co-regulated by a $pi9$ and a $pi17$ piRNA (Extended Data Fig. 4). These results suggest that targets redundantly regulated by $pi2$, $pi9$, and $pi17$ piRNAs are rare and unlikely to explain the fertility defects of $pi9^{-/-}pi17^{-/-}$ and $pi2^{-/-}pi9^{-/-}pi17^{-/-}$ males. Instead, we find that the 14 $pi9$ and $pi17$ targets are distinct but act in the same or overlapping molecular pathways, such as cell proliferation, DNA damage response, or apoptosis. We thus propose that the combined derepression of the 14 $pi9$ and $pi17$ target mRNAs underlies the infertility of $pi9^{-/-}pi17^{-/-}$. Similarly, the fertility

defects in pi2^{-/-}pi9^{-/-}pi17^{-/-} males can be explained by the combined derepression of the 14 pi9 and pi17 target mRNAs and likely the three pi2 targets with putative roles in regulation of apoptosis and translation of ER-targeted proteins.

3. The effects observed at the transcriptional level are rather small. The authors highlight that among the 23 dysregulated transcripts in *pi9* and *pi17* ko primary spermatocytes, 17 mRNAs and 5 lncRNAs increased 1.4–4.4-fold and only one mRNA decreased 2.5-fold. However, among the 14 mRNAs that are targets of *pi9* or *pi17* piRNAs, 13 demonstrate a maximal increase of 1.4–2.4 and only 1 mRNA increased 4.4-fold. We believe that a comprehensive analysis of not only the *pi9* and *pi17* single mutants but also the double and triple mutants in comparison to data from infertile piRNA single loci ko mice such as *pi6* or *pi18* will be needed before conclusions on the cellular effect of these minimal changes (increased apoptosis etc.) can be drawn. In line with this, the testicular sections could also be analyzed for further marker proteins such as γH2AX or SYCP3 to identify the impact on progression of meiosis.

Our data indeed show that the total number of pi9 and pi17 mRNA targets is low and the extent of piRNA-directed repression is modest in primary spermatocytes. These findings are however similar to those for pi6^{-/-}, which is subfertile (Wu et al., 2020), and pi18^{-/-}, which is completely sterile (Choi et al., 2021):

pi9: five mRNAs repressed by 1.5–2.4-fold (median: 1.8-fold); pi17: nine mRNAs repressed by 1.4–4.4-fold (median: 1.9-fold); pi6: eight mRNAs repressed by 2.1–3.6-fold (median: 2.8-fold); pi18: six mRNAs repressed by 1.2–1.9-fold (median: 1.4-fold).

H&E-stained sections of testis and analyses of RNA-seq data for pi9^{-/-}pi17^{-/-} and pi2^{-/-}pi9^{-/-}pi17^{-/-} mutants have been added to the revised manuscript (see our responses to the previous comments). Briefly, our data do not support redundant coregulation of mRNAs by pi2, pi9, and pi17 piRNAs. Instead, we find that pi9 and pi17 targets act in the same pathways and their simultaneous derepression in pi9^{-/-}pi17^{-/-} and pi2^{-/-}pi9^{-/-}pi17^{-/-} males likely explains infertility of these mutants.

4. To better understand the different experiments and in which way they contribute to the main hypothesis of the manuscript that only changes in the transcriptional level of a few mRNAs are responsible for the observed decreased sperm fitness in pachytene piRNA locus mouse mutants, it would be beneficial to present a schematic Figure summarizing the content and the key findings.

We added Fig. 5 summarizing our findings and the proposed model for pachytene piRNA function.

Referee #2

This is a very intriguing study that argues that the majority of mouse pachytene piRNAs expressed from large intergenic piRNA cluster loci are not exhibiting an obvious target regulation event. However, a small minority of individual piRNAs could be essential for spermatogenesis by exhibiting detectable target silencing events. The authors argue that these few piRNAs are behind the evolutionary force to conserve and express a staggering repertoire of pachytene piRNAs to enable proper sperm formation. This study focuses on four mouse mutants that have deleted the promoters from the piRNA cluster loci of *pi2*, *pi6*, *pi9* and *pi17*. Individual piRNA cluster mutants males are still able to sire litters, so the authors generated several double and triple mutants combinations. The combinations of the *pi9* mutant with *pi7* and *pi8* with *pi17* mutants shows the strongest loss of male fertility and sperm dysfunction, while combinations of the *pi2*, *pi7* and *pi17* mutations together that remove 19% of piRNAs are still nearly as fertile as wild type. This genetic data is both amazing and baffling that some piRNA cluster loci have compensation from the remaining piRNAs able to support normal spermatogenesis, while the loss of *pi9* and *pi17* mutated piRNA clusters individually and together exhibit the strongest infertility defects.

Are the transcriptome analysis of differential expression between *pi9* and *pi17* individual and double and triple mutants compared to the C57BL/6 controls also showing that no transposon RNAs and long-noncoding RNAs are upregulated in mutants like in *pi2* or *pi6* mutants? If I missed this analysis, could this be extracted from all the existing transcriptome and ribosome occupancy data that could also be subjected to mapping of reads to transposons and other long-noncoding RNAs. Do the new loosened complementarity rules for pachytene piRNA targeting increase the potential for more transposons transcripts to be targetable for cleavage?

The revised manuscript includes analyses showing that active transposons with intact open reading frames remain repressed in $pi2^{-/-}$, $pi9^{-/-}$, $pi17^{-/-}$, $pi9^{-/-}pi17^{-/-}$, and $pi2^{-/-}pi9^{-/-}pi17^{-/-}$ males (Supplementary Table 5). The steady-state levels of six long noncoding RNAs change in $pi6^{-/-}$, $pi9^{-/-}$, and $pi17^{-/-}$ primary spermatocytes (Wu et al, 2020; Supplementary Table 3) but no molecular or biological function has been reported for any of these lncRNAs.

The ribosome occupancy profile experiment with these piRNA mutants is a very important negative result to refute the controversial claim made by the Liu group that pachytene piRNAs may activate mRNA translation. I was impressed by multiple replicates of this RFP-seq profiling of mutants and wt control spermatocytes and spermatids, but the results are not clear how replicates were compared for reproducibility and RFP-seq library quality control (QC). To back up this negative result that there is no translational activation, authors should show this is not due to incomplete RFP-seq, so I would like to see some QC like polysome profiles and the RFP coverage plots on *Cnot4*, *Tbpl1*, and *Spesp1* with the periodic peaks over codon triplets. Similar QC results were in the Liu group's paper.

Ribosome footprint profiling (RFP) was performed as described previously by the Ingolia lab (PMID: 28579404). The detailed description of data analyses has been included in the revised Methods section. As a quality control metric, we calculated the 5'-to-5' distance for RFP sequencing reads; the data show prominent density peaks at 3-nt intervals as expected for ribosomes translocating from triplet codon to triplet codon (Extended Data Fig. 10). As suggested, we added RFP-seq and RNA-seq coverage plots for Cnot4, Tbp11, and Spesp1 mRNAs demonstrating that the periodic peaks of 5' ends of RFP reads are present in ORFs but not UTRs (Supplementary Figs. 1–3).

The authors should comment if they have biochemical data from their monumental study in Arif et al 2022 to validate in vitro cleavage by MIWI piRNA complexes to bolster the portrayed piRNA:target pairings for *Urgcp*, *Cox7a2l* and *Zdhhc16* of Fig. 2b where there is interruption of the piRNA seed-sequence pairing. Would these piRNAs cleave a target in their MIWI in vitro cleavage assay?

We thank the reviewer for their recommendation to use cleavage assays to test slicing of targets lacking perfect pairing to piRNA seed. The new data helped us refine our piRNA targeting analyses. We loaded piRNAs targeting Brca2, Urgcp, Cox7a2l, or Zdhhc16 into purified, recombinant MIWI and performed cleavage assays in the presence of 100 nM recombinant GTSF1 using model RNAs containing the corresponding target site. Because the concentration of each piRNA is greater than its target in vivo, we used 0.5 nM piRISC and 0.1 nM target RNA. We readily detected target cleavage for Brca2, Urgcp, and Cox7a2l (Extended Data Fig. 3) but not Zdhhc16. Consequently, we reanalyzed our data using more relaxed targeting rules, requiring fewer total paired nucleotides while preferring longer contiguous complementarity. The new approach identified an extremely abundant pi17 piRNA (1,800 molecules/primary spermatocyte) pairing with its nucleotides g2–g16 to the Zdhhc16 target site with a single mismatch at g12 (Fig. 2b). The newly identified piRNA directed efficient slicing of the model Zdhhc16 target in vitro (Extended Data Fig. 3). We note that the more relaxed rules failed to identify additional targets of pi2, pi9, or pi17 piRNAs.

I am not convinced by the authors' claim for why many putative piRNA-cleaved targets are not changing abundance in the piRNA mutants simply because they have a higher transcription rate compared to other non-piRNA target transcripts. Although *Drosophila* Piwi/piRNA complexes can impact a target transcript's transcription rate, mouse MIWI/piRNA cleavage should be independent of target transcription rate, and this argument of high transcription rate "buffering" out transcriptional change between WT and piRNA cluster mutant can only be substantiated if the piRNA cleavage activity is shown to be very weak at reducing steady state target levels. If piRNA cleavage activity is presumed strong like the impressive MIWI cleavage activity assays shown in Arif et al, then regardless of target transcription rate the RNAseq measurements for WT and mutants are steady-state measurements that should show target up-regulation in the

mutants. Could piRNA-target cleavage rates show wide variation and perhaps actually be weak for many piRNAs but only strong for the specific piRNAs highlighted in Fig. 2? Is there cleavage kinetics from the Arif et al assay that could substantiate variation and weak cleavage by some piRNAs?

The reviewer's suggestion prompted us to analyze the in vivo slicing efficacy of piRNAs with detectable 3' cleavage products. As hypothesized by the reviewer, our data show that the efficiency of slicing by most pachytene piRNAs is low (Fig. 3b). The in vivo cleavage efficiency for each target transcript can be compared by determining the fraction of target cleaved. We estimated this as the ratio of the abundance of 3' cleavage products (detected by sequencing 5' monophosphorylated RNA) to the abundance of uncleaved targets (from poly(A)⁺ RNA-seq). The estimated fraction of target cleaved was ~3.5-fold higher for piRNAs that alter steady-state target levels by ≥ 1.25 -fold, compared to piRNAs with little effect on steady-state target abundance (change in mutant vs. control ≤ 1.25 -fold; Fig. 3b). Moreover, piRNA concentration strongly influences cleavage rate, and we find that piRNAs that do not change target levels are two-fold less abundant in vivo than those that alter target steady-state abundance (Fig. 3b).

This study is very significant, and as usual the Zamore lab presents high quality data and rigorous analysis. The crux of this paper is the very provocative interpretation that just a small set of individual piRNAs may have an outsized impact of target regulation while the rest of the piRNAs in the cluster may be useless bystanders. But I feel that a necessary genetic validation of this interpretation that is missing, and this would be for the authors generate additional mouse mutants that just delete the individual key piRNA sequences in the *pi9* and *pi17* piRNA loci for the piRNA sequences shown in Figure 2, and then see if those deletion mutants of a single piRNA sequence would have the same major spermatogenesis phenotypes as the deletion of the piRNA cluster promoter that removes all the other piRNAs from the locus. I realized this additional experiment would extend the scope of this study, and to this point the Editor should arbitrate on this point I raise. The most feasible test could be to generate a mutant or mutants that delete just the region around the two individual pi9 piRNAs in Fig. 2A., leaving all the other pi9 cluster piRNAs intact, and see these single or smaller fraction piRNA loss mutants phenocopy the pi9-promoter mutant? If the editor does not request this result for this study, then I recommend that the authors add this point to the paper discussion as a limitation in this study that individual piRNA mutants are not yet available to validate this provocative conclusion.

The follow-up genetic experiments suggested by the reviewer are an excellent direction to further test the proposed model. We are grateful to the editors for deeming these beyond the scope of the present study. The Discussion section of the revised manuscript now states: "Future experiments, such as deletion of individual piRNA target sites or piRNA sequences should help test this "piRNA addiction" model."

Minor point: The title says [“Mammalian” pachytene piRNAs persist...] but this study only has primary analysis of mouse piRNAs, no analysis in other mammals, so maybe change title to just “Mouse” pachytene piRNAs?

We agree that the proposed edit makes the title more accurate.

Referee #3

In this manuscript, Cecchini et al. characterize the effects of deleting four pachytene piRNA clusters—either separately or in various combinations—on the expression of protein-coding genes. They found that only a small subset (1%) of pachytene piRNAs can target host genes. Among the putative piRNA targets, only a small fraction shows altered expression following the loss of clusters. The authors attribute the lack of transcriptional changes to the high expression of most targeted genes. Interestingly, those genes that do show differential expression tend to be enriched in genes whose derepression impedes spermatogenesis. This suggests that piRNAs are necessary to keep the expression of these genes low in order to maintain fitness. The authors argue that this fitness advantage, derived from regulating a small number of genes, ensures the survival of all pachytene piRNAs, the majority of which are considered selfish elements. They also propose that natural selection eliminates piRNAs that would cause detrimental changes in gene expression, thereby limiting the number of genes affected by piRNA regulation. Finally, they find that genes regulated by piRNAs are disproportionately targeted at TE-derived sequences within transcripts recognized by piRNAs complementary to or derived from transposon sequences. This suggests an important role for transposons in the evolution of host gene regulation. Their findings provide strong evidence for gene regulatory function of pachytene piRNAs, and support for their retention in the genome.

Experiments are well designed, with necessary controls and appropriate statistical analysis to support the claims. The text is well written and clear, although some of the figures might benefit from simplifications to more clearly convey the message.

Cecchini et al. also carefully investigate the mechanism by which piRNAs influence gene expression. Previous studies have suggested that, in addition to transcript cleavage, piRNAs also impact translation efficiency and induce deadenylation of target transcripts. However, the authors find no evidence to support these claims.

Unfortunately, the piRNA field has been impacted by high-profile publications that claim broad functions for pachytene piRNAs in gene regulation and propose diverse, unexpected mechanisms of action—some of which contradict well-established knowledge about piRNAs. The authors of this manuscript provide a much-needed service to the small RNA field and the broader RNA and gene regulation community by examining two controversial topics through rigorously planned and executed experiments and proper statistical analysis:

1. Disproving claims about the generalized gene regulatory function of pachytene piRNAs.
2. Demonstrating that piRNAs regulate gene expression solely through endonucleolytic cleavage of complementary targets, rather than inducing deadenylation or regulating translation.

Although the general findings of this work that identify true pachytene piRNA targets that explain the mutant phenotype, and the detailed dissection of mechanism of action provide ample novelty, even without additional novelty this work should be strongly

considered for publication in Nature to set the record about piRNA targets and mechanism straight.

Below, I summarize some comments and suggestions for modifications and additional analyses that might further enhance the manuscript.

Comments and Suggestions:

- Statistically significant effects on fertility are only observed when different cluster mutations are combined, implying functional redundancy between clusters. It would be interesting to determine whether these redundancies in targeting are observable at the sequence level. Do multiple clusters produce piRNAs with sufficient complementarity to a given gene such that the loss of one cluster does not significantly reduce piRNA levels to the point of altering gene expression, but the loss of several clusters does? Alternatively, are multiple genes targeted independently by different clusters, with the additive effect of derepressing several genes necessary to observe the decreased fertility phenotype?

We thank the reviewer for the excellent suggestion to investigate the molecular basis of the apparent genetic redundancy of pachytene piRNA-producing loci. The new results included in the revised manuscript lend support to the reviewer's second explanation (Figs. 2c and Extended Data Fig. 5). Briefly, analyses of new phenotypic and molecular data from pi9^{-/-}pi17^{-/-} and pi2^{-/-}pi9^{-/-}pi17^{-/-} males suggest that redundant co-regulation of targets by piRNAs from multiple loci is extremely rare. In contrast, our data show that pi9 and pi17 targets are distinct but act in overlapping pathways, including regulation of DNA damage repair, cell proliferation, and apoptosis. We therefore propose that the combined derepression of pi9 and pi17 targets likely underlies the fertility defects of pi9^{-/-}pi17^{-/-} and pi2^{-/-}pi9^{-/-}pi17^{-/-} males.

- A potential follow-up, albeit a time-consuming one, could be to test whether changes in expression of the identified deregulated genes (e.g., those involved in DSB repair) account for the observed phenotype. Would the relatively mild 1.5x-2x overexpression of target transcripts regulated by piRNAs be sufficient to induce a fitness loss in sperm or an increase in DSBs? Would this effect depend on the simultaneous overexpression of multiple genes?

The genetic experiments proposed by the reviewer are a great potential way to test the "death-by-a-thousand-cuts" hypothesis but are outside of the scope of this work. We are again grateful to the editors for not requiring additional mouse models.

- Related to the previous question: Effective targeting (i.e., reduction in transcript levels) could be accomplished by a single highly expressed piRNA, as seen in the examples in Fig. 2. Alternatively, it could result from many independent but lower-concentration piRNAs acting along the transcript. These could, in theory, come from multiple clusters, even if individual targeting sequences are unique to specific clusters. Could "weak but parallel" targeting of certain genes by piRNAs from several clusters explain the lack of broader changes in gene expression observed when only 1–3 clusters are deleted? Theoretically, such parallel targeting by many clusters could add robustness, allowing for some level of sequence drift without disrupting overall gene regulation by piRNAs.

Data from pi9^{-/-}pi17^{-/-} and pi2^{-/-}pi9^{-/-}pi17^{-/-} males suggest that targets co-regulated by several piRNA-producing loci are extremely rare (Figs. 2c and Extended Data Fig. 5). Our analyses of mutants lacking pi2, pi9, and pi17 piRNAs identified only one transcript co-regulated by a pi9 and a pi17 piRNA (Extended Data Fig. 5a). These results support the idea that the fertility defects of pi9^{-/-}pi17^{-/-} and pi2^{-/-}pi9^{-/-}pi17^{-/-} males are unlikely to be a consequence of derepression of transcripts regulated redundantly by pi2, pi9, and pi17 piRNAs.

- Do expression levels of transposable elements (TEs) also change in the cluster mutants? Although pachytene piRNAs contain few TE sequences, some still regulate host genes. Could derepression of specific transposons contribute to the observed fertility effects?

The revised manuscript includes analyses of changes in the abundance of transcripts derived from transposons. Our data show that, as previously reported for pi6^{-/-} and pi18^{-/-} mutants, active transposons with intact open reading frames remain repressed in pi2^{-/-}, pi9^{-/-}, pi17^{-/-}, pi9^{-/-}pi17^{-/-}, and pi2^{-/-}pi9^{-/-}pi17^{-/-} males (Supplementary Table 5).

- Are genes regulated by TE-derived piRNAs also regulated by pre-pachytene piRNAs? One might predict that pre-pachytene piRNA sequences could have higher concentrations and thus repress these genes more strongly at earlier developmental stages.

This is an interesting question. To test the reviewer's prediction, we searched for transposon-derived pre-pachytene piRNAs with sufficient complementarity to guide cleavage of pi9 and pi17 targets bearing transposon-derived sequences in their UTRs. We found no such piRNAs. A potential explanation is that many pre-pachytene piRNAs derive from evolutionarily young, transpositionally active transposon families, whereas mRNA UTRs are depleted of such insertions.

- The almost identical target sequence in four genes regulated by pi9 suggests co-regulation of these genes. Is the URR1B sequence overrepresented in 3' UTRs in general, or is the presence of this URR1B site in these genes the result of natural selection to co-regulate them? How does the expression of these four genes in mouse spermatocytes compare to that in other, closely or distantly related species?

We find that the frequency of URR1B insertions is similar in intergenic regions (0.45% of all intergenic TEs) and in 3' UTRs (0.5% of all sense TE insertions and 0.49% of all antisense TE insertions in 3' UTRs), arguing against the idea that URR1B preferentially transposed into 3' UTR sequences or was selected for retention in 3' UTRs. We discuss the differences in expression levels of pi9 targets in other species in our response to the next comment.

- The authors argue that natural selection eliminates deleterious targeting, and that the genes that piRNAs effectively target provide benefits that help maintain pachytene piRNA production. Alternatively, one could view this as a form of "piRNA addiction." It is possible that the original expression levels of these genes were low, and their promoters "adjusted" to compensate for piRNA-mediated silencing. Without piRNAs, their expression might now be too high, creating a dependency on piRNA silencing. For genes where higher expression is not deleterious, the host could have adjusted by increasing gene expression, which could explain why most

targeted genes are highly expressed and unaffected. Instead of natural selection eliminating the piRNA, the target gene's expression level could have been increased. The authors could test whether orthologs in closely and distantly related species are expressed at similarly low levels when complementary piRNAs are absent. The results could further refine the understanding of the evolutionary importance of piRNAs in host gene regulation and the co-evolution of piRNA targets.

The reviewer's hypothesis that piRNA cleavage could occur first and may later be followed by the compensatory promoter strength "adjustment" requires that the cleavage-induced decrease in the target level does not reduce sperm fitness, i.e., the piRNA-directed cleavage is neutral. Otherwise, the piRNA or the target site would be purged by negative selection even before the promoter is adjusted. Investigating how such complex gene regulatory networks arise is an important question in evolutionary studies but is beyond the scope of our work.

*Nonetheless, to begin to test the reviewer's hypothesis, we FACS-purified primary spermatocytes from the PWK/PhJ and CAST/EiJ mouse strains and sequenced their piRNAs, mRNAs, and 3' cleavage products. These two strains are derived from the two closest subspecies of *Mus musculus domesticus*: *M. m. musculus* and *M. m. castaneus*. These three subspecies diverged ~0.5 MYA. All 14 piRNA:target pairs and cleavage products identified in this work for pi9 and pi17 in *M. m. domesticus* were present in *M. m. musculus* and *M. m. castaneus*. However, we observed ~2-fold variation in target abundance among the three subspecies, likely because of differences in promoter sequences and potential post-transcriptional regulation distinct from piRNA-directed cleavage. These preliminary data suggest that testing the reviewer's hypothesis by comparing target abundance in different species is unlikely to be feasible because of additional confounding regulatory mechanisms that vary among even closely related species.*

Minor Comments:

- Fig. 3 is difficult to interpret, as it contains many text blocks that are not intuitive to navigate. For example, the text blocks that describe "n=16 permutations..." could be simplified by replacing this with asterisks (*) and defining them in the figure legends. Additionally, the bars showing the stats in Figures 3c and 3d are hard to interpret. Could these be simplified to represent all 16 permutations with one IQR and median? In particular, in Fig. 3d, the evolutionary significance would be clearer if more attention were drawn to the difference between piRNAs that cause changes in gene expression versus those that do not, with the genomic features shown below, including those with little (e.g., all pachytene piRNAs) or high (CDS) conservation.

We updated Fig. 3 to improve its clarity.

- The connection between the lack of change in germ cell granule formation and the paper's narrative is not entirely clear. I assume this result is meant to show that overall piRNA processing/cleavage functions are not affected by gross structural changes, and that the observed sterility phenotype is primarily due to the loss of transcript repression. This section would benefit from a sentence explicitly linking this result to the paper's conclusions.

Pachytene piRNAs are the major RNA component of germ cell granules and were hypothesized to be important for granule integrity. We updated the text to

say: "In contrast, we find that loss of ~13% of pachytene piRNAs in pi2^{-/-}pi17^{-/-} and pi9^{-/-}pi17^{-/-} males has no detectable effect on germ cell granule morphology."

- How is "piRNAs removed" in panel 1a defined? Does it refer to reads or sequences? Does "removed sequence" mean completely absent in the mutant library, or is there a threshold for fold decrease or maximum read count?

Panel 1a shows the number of piRNA species completely absent from each mutant. We describe how piRNA species are defined in the Methods section: "Because piRNA 3' trimming by PNLDC1 results in heterogeneous 3' ends, sequencing reads were next grouped by their 5', 25-nt prefix. For further analyses, we kept only prefix groups that met two criteria. First, the prefix group total abundance was ≥ 1 ppm, i.e., ≥ 10 piRNAs/mouse primary spermatocyte. Assuming either a Poisson or a Negative Binomial distribution for piRNA concentration in different cells, this threshold ensures that $\geq 99.99\%$ of primary spermatocytes contained at least one molecule of the piRNA 25-nt prefix. Second, the total abundance of the prefix group was required to be ≥ 1 ppm in all 12 replicates of the C57BL/6 control samples (Supplementary Table 2). piRNAs were considered undetectable in pi6^{-/-}, pi9^{-/-}, pi17^{-/-}, pi9^{-/-}pi17^{-/-} or pi2^{-/-}pi9^{-/-}pi17^{-/-} primary spermatocytes if their mean abundance in mutants was ≤ 0.1 ppm."

- Do the authors believe that the decreased Idh1 levels in the pi9 mutant (Fig. 2a) are secondary?

Such an interpretation would be the simplest explanation for the change in Idh1 mRNA abundance.

- In the discussion, I suggest modifying the sentence "only several hundred are sufficiently abundant and complementary to a transcript to direct its cleavage," as this implies that abundance affects cleavage. Abundance actually only impacts detectable changes in transcript levels due to cleavage.

We agree. The sentence was changed to "...only several hundred (~1%) are sufficiently complementary to a transcript to direct its cleavage."

Responses to Reviewers' Critiques on the Revised Version of Cecchini et al.

In addition to the edits requested by the reviewers, we updated the main text on page 5 to highlight how we calculated changes in transcript abundance for $pi9^{-/-}$, $pi17^{-/-}$, and $pi9^{-/-}pi17^{-/-}$ mutants was calculated. This information was previously only in the Methods.

“To exclude Cas9-induced off-target changes in gene expression, we used two different pairs of sgRNAs to generate two independent alleles for each locus and considered only molecular changes detected in both alleles. We report the smaller change in transcript abundance between the two alleles.”

Referee #1

In the revised version, the authors have addressed most of our comments, yet the following point remains unclear and would benefit from rephrasing:

1. The authors included transcriptome analyses in spermatocytes of single $pi9^{-/-}$, single $pi17^{-/-}$, double $pi9^{-/-}pi17^{-/-}$ and triple mutants ($pi9^{-/-}pi17^{-/-}$ and $pi2^{-/-}$) in Figure 2c, demonstrating that the 14 $pi9$ and $pi17$ targets are de-repressed to the same extent in single, double, and triple mutants. However, in the manuscript text on page 7 they are not referring to the triple mutant but only to the double mutant and write: "To identify the molecular cause of $pi9^{-/-}pi17^{-/-}$ fertility defects, we compared the transcriptomes of $pi9^{-/-}$, $pi17^{-/-}$, and $pi9^{-/-}pi17^{-/-}$ primary spermatocytes. The abundance of $pi17$ targets increased to the same extent in $pi17^{-/-}$ and $pi9^{-/-}pi17^{-/-}$ primary spermatocytes (Fig. 2c). $pi9$ targets were also similarly ($\pm 50\%$) derepressed in $pi9^{-/-}$ and $pi9^{-/-}pi17^{-/-}$ males (Fig. 2c)." We suggest to explain all data of Figure 2c in the respective section. The result from the triple mutant is only mentioned on page 9, when the authors explain that targets co-regulated by piRNAs from different loci are rare.

We have changed the text on page 7 to include a discussion of the triple mutant:

“Unlike $pi9^{-/-}$ and $pi17^{-/-}$, $pi9^{-/-}pi17^{-/-}$ and $pi2^{-/-}pi9^{-/-}pi17^{-/-}$ males were infertile (Fig. 1a). The median number of embryos carried by females at day 14.5 after mating with $pi9^{-/-}pi17^{-/-}$ or $pi2^{-/-}pi9^{-/-}pi17^{-/-}$ males was 0, compared to 8 embryos for females mated with control C57BL/6 males (Fig. 1d). $pi9^{-/-}pi17^{-/-}$ and $pi2^{-/-}pi9^{-/-}pi17^{-/-}$ males exhibited no change in testicular germ cell composition or sperm gross morphology (Extended Data Fig. 4a). However, $pi9^{-/-}pi17^{-/-}$ and $pi2^{-/-}pi9^{-/-}pi17^{-/-}$ sperm were less plentiful, showed impaired motility, and were unable to penetrate the oocyte zona pellucida (Figs. 1b and 1c, Extended Data Figs. 1b, 1c, 1d). $pi9^{-/-}pi17^{-/-}$ caudal sperm also contained deformed midpiece mitochondria (Extended Data Fig. 4b).

“To identify the molecular cause of $pi9^{-/-}pi17^{-/-}$ and $pi2^{-/-}pi9^{-/-}pi17^{-/-}$ fertility defects, we compared the transcriptomes of $pi9^{-/-}$, $pi17^{-/-}$, $pi9^{-/-}pi17^{-/-}$, and $pi2^{-/-}pi9^{-/-}pi17^{-/-}$ primary spermatocytes. The abundance of $pi17$ targets increased to the same extent in $pi17^{-/-}$, $pi9^{-/-}pi17^{-/-}$, and $pi2^{-/-}pi9^{-/-}pi17^{-/-}$ primary spermatocytes (Fig. 2b). $pi9$ targets were also

derepressed to similar ($\pm 50\%$) extents in $pi9^{-/-}$, $pi9^{-/-}pi17^{-/-}$, and $pi2^{-/-}pi9^{-/-}pi17^{-/-}$ males (Fig. 2b).”

2. *The triple mutant is not mentioned in Figure legend 1B/C.*

We have revised the figure legend to include the triple mutant $pi2^{-/-}pi9^{-/-}pi17^{-/-}$.

3. *A figure legend for Figure 5 is missing.*

Apologies! We have now included a legend for Figure 5 (Fig. 4c in the revised version).

Referee #2

I have completed my assessment of the revision and rebuttal letter. I am satisfied with the author response in the rebuttal letter, but have two more suggestions on the new data and updated tables provided in the revision. These small suggestions once addressed by the reviewers would complete my review of this study, happy to see it get accepted for publication.

1) *There is new analysis in this revision regarding the one common mRNA being targeted by piRNA complementarity to pi9 and pi17, and that Asb1 mRNA is only derepressed when both pi9 and pi17 were both removed. Since this is a new finding being reported in this revision, I would like the authors to comment in the final manuscript that when Asb1 mRNA is upregulated 1.4 fold in the double pi9&pi17 mutant, are the 3'UTR Asb1 piRNAs also increased in a greater capacity compared to other genic 3'UTR piRNA clusters? Just looking at genic piRNA cluster and not other intergenic clusters should serve as a control for the overall change in piRNA sequences being lost in the double pi9&pi17 mutant?*

As suggested by the reviewer, we added the new data to Extended Data Fig. 5a. Our analyses show no significant change in abundance of piRNAs derived from the *Asb1* mRNA, potentially due to the loss of the *pi9* and *pi17* piRNAs that initiate the production of responder piRNAs from *Asb1*.

2) *Some of the new Supplemental tables numbers have too many digits beyond significance that make reading the tables difficult (yes, I know I can download into Excel and change the formatting myself, but authors should make it easier for readers to follow these tables).*

We totally understand this request! The tables have been reformatted to include fewer digits.

3) *The GitHub repository is missing a Readme file to explain the code and input files.*

We have added the Readme file to the GitHub site.

Reviewer #3

As already described in detail in my first review, I find the work significant, rigorously executed with appropriate statistical analyses and well described. The conclusions are robust (apart from a few comments below), the references appropriate and the key results for the most part unchanged compared to the previous submission. Overall, I find that the authors answered the concerns of the reviewers to my satisfaction.

I would like to push back on the authors argument for why they believe that promoter strength “adjustment” is not contributing to a “piRNA addiction” model. In the rebuttal, the authors argue that such promoter strength adjustment “requires that the cleavage-induced decrease in the target level does not reduce sperm fitness, i.e., the piRNA-directed cleavage is neutral. Otherwise, the piRNA or the target site would be purged by negative selection even before the promoter is adjusted.”

I disagree and would like to provide two counterarguments:

First, their own analysis in response to another concern showed that the transcript levels varied ~2x between closely related species for the 14 identified pi9 and pi17 targets, despite all target-piRNA pairs being conserved. The fold changes in gene expression in the cluster mutants compared to wild type were 1.4-4.4x with a median of 2x. The similar level of variance in target expression between the subspecies to the effect of piRNAs implies that there is often plenty of tolerance for variable expression, so negative selection pressure might not be very strong for 2x changes in gene expression.

Second and more important, while the authors are correct that negative selection would purge any piRNAs or target sites that cause very severe fitness changes, such as complete sterility, this does not necessarily hold true for milder fitness reduction on a population level. In the case of moderate fitness cost, deleterious piRNAs could either be lost through negative selection (as suggested by the authors) or compensatory mechanism, such as hyperactive promoters, could be positively selected to restore expression. Importantly, the promoter “adjustment” that may occur is on the population scale, meaning they are not necessarily changing in response to piRNA; rather, through the random walk of evolution, hyperactive promoter variants may arise and be present in the population in a subset of individuals. While the higher transcript level might be disadvantageous under normal circumstances, these individuals would be positively selected when pervasive deleterious piRNAs emerge. This way, deleterious piRNAs could persist, while enhanced target promoter expression becomes the new steady-state level in subsequent generations/populations. In the dataset in this paper, these targets may appear as genes that are highly transcribed and thus unaffected by small amounts of piRNA... but they may have originally been expressed as low level genes that were subjected to piRNA repression, resulting in a shift at the population level towards enhanced expression of the targeted gene to escape the repressive effect, but becoming “addicted” to the piRNA to counter the deleterious effect of even higher gene expression. Based on the presented data, one cannot assume that high expression genes were highly expressed before they became targets.

We agree that in a wild-type population, the inter-animal level of gene expression can vary substantially. Over an evolutionary timescale, small differences in fitness can provide a significant disadvantage, but the less fit allele will nonetheless persist via heterozygous animals. Because we were asked to substantially shorten the manuscript, we moved to supplementary discussion our speculation on evolutionary mechanisms that could have produced the observed high transcription rates associated with piRNAs that cleave but do not change the abundance of their targets. The supplementary discussion now includes the reviewer's alternative explanation for why most piRNA cleavage targets are highly transcribed.

As a die note, this counterargument also implies that the model presented in figure 5 with four different types of pachytene piRNAs might be an over-simplification of the evolutionary history of these targets and to me it does not provide a lot of aid to understand the key findings. That said, I found the text sufficiently clear without any figure.

We have found that in seminars, the figure has been quite useful in helping an audience understand the fundamental hypothesis, which, like all models, is an oversimplification. Because the model figure was specifically requested by Reviewer 1, we would prefer to keep it.

Minor comments:

1) The following section is difficult to understand:

“The steady-state abundance of Asb1 mRNA was unchanged in pi9^{-/-} and pi17^{-/-} single mutants yet increased 1.4-fold in pi9^{-/-} pi17^{-/-} primary spermatocytes. These data suggest that cleavage by a single pi9 or pi17 piRNA is insufficient to alter Asb1 steady-state levels; Asb1 derepression was only detectable when the pi9 and pi17 piRNAs were both removed (Extended Data Fig. 5a).”

The first and 3rd sentence essentially say the same thing and thus the 3rd sentence is superfluous. However, the logic of the 2nd sentence is confusing, and I believe incorrect. In wild type (piRNAs from both clusters) and single mutants (piRNAs from only one cluster) transcript levels are the same. This I interpret as piRNAs from a single cluster are sufficient to get the transcript to the lower level that is seen in wild type. Only when both clusters are mutated (i.e., there is no mapping piRNA left) is there an increase in transcript level. Thus, the two clusters act redundantly in repressing Asb1 and activity of either is sufficient to induce the silencing of Asb1 to “WT” levels.

We agree and have revised the text to read:

“The steady-state abundance of Asb1 mRNA was unchanged in pi9^{-/-} and pi17^{-/-} single mutants but increased 1.4-fold in pi9^{-/-}pi17^{-/-} primary spermatocytes. Because Asb1 derepression was only detectable when pi9 and pi17 piRNAs were both removed, we conclude that pi9 and pi17 loci act redundantly to repress Asb1 (Extended Data Fig. 5a).

2) In the section titled “Pachytene piRNAs cleave hundreds of transcripts but alter the abundance of just a minority of targets” the authors introduce/ define the different rules that need to be satisfied to be considered a target. However, they refer to piRNA targets in the previous sections. Were those piRNA targets identified based on sequence complementarity or also based on these strict criteria? If the latter, then this should be defined earlier in the results.

The latter, more stringent criteria were used throughout the manuscript. All targets of *pi2*, *pi9*, *pi17* piRNAs had to meet the same two requirements: (1) the presence of a piRNA of sufficient intracellular abundance and target complementarity to have directed cleavage *and* the absence of any piRNA from outside of *pi2*, *pi9* or *pi17* that could have directed target cleavage; (2) the putative 3' cleavage product must have been detected in the C57BL/6 controls and decreased >8-fold in mutant primary spermatocytes. These rules are now defined on both pages 6 and 10–11. We note that we discuss cleavage targets whose abundance is changed by piRNAs—*regulated* targets—on page 6 and describe *all* piRNA cleavage targets on pages 10–11 in the section “Pachytene piRNAs rarely alter target abundance.”

For example, in the section before these target rules are introduced the authors write: “Among the 26 RNAs derepressed in *pi2*^{-/-} *pi9*^{-/-} *pi17*^{-/-} but not *pi9*^{-/-} *pi17*^{-/-}, we identified four mRNAs cleaved exclusively by *pi2* piRNAs but could not find any transcripts repressed by piRNAs from *pi2* and *pi9* or *pi17* (Extended Data Fig. 5b).” Are the targets “cleaved exclusively by *pi2* piRNAs” defined based solely on transcript differential expression in knockouts or piRNA complementarity alone; or is it based on the cleavage criteria defined in the following section?

As before, the latter, more stringent rules were applied throughout the manuscript. The text was updated as suggested.

3) Figure legends are missing for figure 5 (model). While we appreciate the acknowledgement of the “piRNA addiction” phrase, it is not completely clear from the text and the model what it refers to. My interpretation was more towards the “adjustment of transcript level” that then forces the need to retain piRNAs against those genes, while the authors seem to use it to explain why due to the need of a few piRNAs that regulate genes, all the other piRNAs are also retained, keeping the pachytene piRNAs present. Both are valid (and in some ways functionally overlapping) interpretations, but I would suggest explaining the term clearly if it is introduced.

We mean the latter. We now describe our interpretation of the “piRNA addiction” model in the Supplementary Discussion section. We have also included a legend for Figure 5 (Fig. 4c in the revised version).

4) In Fig2c, I suggest adding the “*pi2* targets” explanation for the pink dots to the legend, or, similar to the other categories, on the side, rather than directly on the plot.

We reorganized the figure as suggested.

5) I find the sentence on page 16 “The change in translational efficiency between $pi9^{-/-}pi17^{-/-}$ and C57BL/6 was indistinguishable comparing the putative $pi9$ and $pi17$ targets and the putative targets of non- $pi9$ and non- $pi17$ piRNAs in primary spermatocytes (two-tailed KS test p -value = 0.64), secondary spermatocytes (p = 0.09), and round spermatids (p = 0.37; Fig. 4b and Extended Data Figs. 12a and 12b)” very difficult to understand and would suggest rephrasing.

To improve clarity, we rewrote the sentence to read:

“We identified mRNAs whose 3' UTRs contain both an ELAVL1-binding motif and a piRNA target site complementary to (1) the piRNA seed (nucleotides g2–g8) and (2) an additional ≥ 12 nucleotides within the piRNA region g9–g30 (ref.^{25,26}). We divided these putative targets into two types: those predicted to be regulated by (1) piRNAs from $pi9$ or $pi17$ and (2) piRNAs from neither $pi9$ nor $pi17$. For each target type, we calculated the change in translational efficiency in $pi9^{-/-}pi17^{-/-}$ mutants compared to control. The change in translational efficiency was indistinguishable for the two target types in primary spermatocytes (two-tailed KS test p -value = 0.64), secondary spermatocytes (p = 0.09), and round spermatids (p = 0.37; Fig. 4b and Extended Data Figs. 12a and 12b). We conclude that piRNAs from $pi9$ or $pi17$ do not activate translation in collaboration with ELAVL1.”